# Obligatory and accessory respiratory muscle structure, function and control in early and advanced disease in the *mdx* mouse model of Duchenne muscular dystrophy

Aoife D. Slyne [ID], David P. Burns, Karina Wöller, Amandine May, Roisin Dowd, Sarah E. Drummond, Grzegorz Jasionek and Ken D. O'Halloran [ID]

*Department of Physiology, University College Cork, Cork, Ireland*

Handling Editors: Harold Schultz & Kevin Murach

The peer review history is available in the Supporting Information section of this article (https://doi.org/10.1113/JP288709#support-information-section).

**Abstract figure legend** We examined obligatory (diaphragm, external intercostal and parasternal) and accessory (sternomastoid, cleidomastoid, scalene and trapezius) muscle form, function and control in early (4 months) and advanced (16 months) dystrophic disease. Peak inspiratory pressure is preserved in 4-month-old *mdx* mice but is decreased in 16-month-old *mdx* mice compared to age-matched wild-type control mice. Progressive respiratory muscle remodelling with extensive fibrosis in the diaphragm muscle underpins the emergence of impairments in respiratory system performance in *mdx* mice. Interestingly, ventilation is remarkably well compensated in *mdx* mice at 16 months of age, revealing that logistically convenient mouse models of muscular dystrophy are required to study ventilatory insufficiency of end-stage disease.

The Journal of Physiology

**Abstract** Peak inspiratory pressure-generating capacity is preserved in the *mdx* mouse model of Duchenne muscular dystrophy in early disease, despite profound diaphragm muscle weakness and reduced electrical activation, revealing adequate compensation by extra-diaphragmatic muscles. Respiratory system compensation is lost as disease progresses, with the emergence of reduced peak inspiratory pressure-generating capacity in advanced disease. We hypothesised that extra-diaphragmatic inspiratory muscles compensate for diaphragm dysfunction in early dystrophic disease, supporting the maintenance of peak respiratory performance in *mdx* mice. We reasoned that extra-diaphragmatic muscle dysfunction would emerge with progressive disease, leading to the loss of peak inspiratory pressure-generating capacity in advanced dystrophic disease. We measured ventilation, inspiratory pressure, and obligatory (diaphragm, intercostal and parasternal) and accessory (sternomastoid, cleidomastoid, scalene and trapezius) respiratory muscle form, function and EMG activity in early (4 months) and advanced (16 months) dystrophic disease. Despite obligatory and accessory muscle dysfunction, including structural remodelling, weakness and reduced EMG activity, peak inspiratory pressure-generating capacity and ventilation are preserved in early disease. Obligatory and accessory muscle dysfunction progressively declines with advanced disease, with the emergence of reduced peak inspiratory pressure-generating capacity. However, although there was evidence of progressive accessory muscle dysfunction, more profound remodelling was seen in the diaphragm muscle comparing early and advanced dystrophic disease. In conclusion, in early dystrophic disease, peak inspiratory performance is compensated. A progressive decline in diaphragm and extra-diaphragmatic muscles contributes to respiratory system compromise in advanced disease. Further loss of compensation afforded by extra-diaphragmatic muscles probably contributes to end-stage respiratory failure.

(Received 9 February 2025; accepted after revision 16 May 2025; first published online 16 June 2025)

**Corresponding author** K. D. O'Halloran: Department of Physiology, University College Cork, Western Gateway Building, Western Road, Cork, T12 XF62, Ireland. Email: k.ohalloran@ucc.ie

## Key points

- We characterised obligatory and accessory respiratory muscle form, function and control in early and advanced disease in the *mdx* mouse model of Duchenne muscular dystrophy.
- Profound diaphragm muscle remodelling, immune cell infiltration, elevated cytokine concentrations and dysfunction present in early disease, but peak inspiratory performance is fully compensated. The burden of breathing is shared across many muscles, revealed as remodelling, elevated cytokine concentrations, weakness and impaired control in several obligatory and accessory muscles.
- Peak inspiratory performance declines in advanced disease with evidence of progressive remodelling in the diaphragm muscle with extensive fibrosis and further decline in the form, function and control of accessory muscles of breathing.
- Diaphragm remodelling with profound fibrosis, more so than progressive accessory muscle remodelling (although evident), is the striking phenotype at 16 months of age when the decline in peak inspiratory performance appears.
- The progressive decline to end-stage disease ($\sim$20–22 months of age in *mdx* mice) probably relates to continued profound loss of diaphragm contractile function and loss of compensatory support provided by extra-diaphragmatic muscles. Logistically convenient models of rapid, progressive muscular dystrophy are required to facilitate the study of end-stage disease.

## Introduction

Duchenne muscular dystrophy (DMD) is a life-limiting X-linked disorder, characterised by neuromuscular dysfunction secondary to dystrophin deficiency (Ervasti, 2007; Hoffman et al., 1987). Dystrophin is a structural protein expressed in muscle, playing a

key role in stabilisation of the sarcolemma during muscle contraction, protecting the muscle from contraction-induced injury (Petrof et al., 1993). The absence of dystrophin in DMD leads to severe and progressive skeletal muscle remodelling and dysfunction (Blake et al., 2002), which extends to the respiratory muscles, with deleterious consequences for respiratory system performance (Mankodi et al., 2017). There is a progressive decline in respiratory muscle and pulmonary function with disease progression in DMD (Khirani et al., 2014; Pennati et al., 2020; Phillips et al., 2001). Premature death ultimately occurs in DMD because of ventilatory insufficiency leading to cardiorespiratory failure (LoMauro et al., 2015). There is currently no cure for this devastating disease.

The respiratory control system is relatively under-studied in DMD (Mhandire et al., 2022). Profound diaphragm muscle weakness is a feature of DMD and has been well documented in patients and animal models of DMD (Delaney & O'Halloran, 2024; LoMauro et al., 2018; Mhandire et al., 2022; Pennati et al., 2020). A relative paucity of knowledge exists regarding the integrative respiratory control system in DMD, particularly in advanced dystrophic disease. Studies characterising the respiratory control system in early dystrophic disease in the *mdx* mouse model of DMD have revealed that, despite profound diaphragm muscle structural remodelling and weakness (Burns, Ali, et al., 2017; Burns, Roy, et al., 2017; Burns et al., 2018; Burns, Drummond, et al., 2019; Burns, Murphy, et al., 2019; Coirault et al., 1999; Coirault et al., 2003; O'Halloran et al., 2023; Stedman et al., 1991), as well as impaired electrical activation (Burns, Murphy, et al., 2019; O'Halloran et al., 2023), peak inspiratory pressure-generating capacity is preserved, revealing a remarkable capacity for compensation of respiratory system performance in the face of substantial impairments to the primary inspiratory muscle, suggesting a role for extra-diaphragmatic muscles (Burns, Murphy, et al., 2019; O'Halloran et al., 2023). Indeed, lesioning of accessory inspiratory and abdominal muscles withdraws accessory muscle compensation of peak respiratory system performance in early disease (O'Halloran et al., 2023).

Peak obligatory (diaphragm, external intercostal and parasternal) respiratory muscle EMG activities are reduced in *mdx* mice, evident from early in disease, before the temporal decline in peak respiratory system performance (Burns, Murphy, et al., 2019; O'Halloran et al., 2023). Peak EMG activities of some accessory respiratory muscles are lower (sternomastoid), whereas others are preserved (cleidomastoid and scalene) or greater (trapezius) in *mdx* mice in early disease (O'Halloran et al., 2023). Respiratory system compensation is lost as disease progresses, with the emergence of reduced peak inspiratory pressure-generating capacity in advanced disease (16-month-old *mdx* mice) (O'Halloran et al., 2023). We reasoned that progressive accessory muscle dysfunction in advanced DMD underpins the temporal decline in respiratory performance culminating in respiratory failure in end-stage disease. We sought to examine accessory respiratory muscle form, function and EMG activity in early dystrophic disease, when peak respiratory system performance is preserved in *mdx* mice, and in advanced dystrophic disease, when peak respiratory system performance declines in *mdx* mice.

We hypothesised that extra-diaphragmatic inspiratory muscles compensate for diaphragm dysfunction in early dystrophic disease, supporting the maintenance of peak respiratory performance in *mdx* mice. We reasoned that this compensation is eroded as disease progresses, leading to the loss of peak inspiratory pressure-generating capacity in advanced dystrophic disease. Therefore, we hypothesised that accessory muscle form, function and control would undergo progressive impairment with substantive dysfunction in advanced disease.

## Methods

### Ethical approval

Procedures on live animals were performed under project authorisations (AE19130/P117 and AE19130/P157) from the Health Products Regulatory Authority in accordance with Irish and European law (directive 2010/63/EU) following approval by University College Cork's animal research ethics committee (AEEC 2019/013 and AEEC 2021/019). Experiments were carried out in accordance with University College Cork's Animal Welfare Body guidelines and conform to the principles and regulations described by Grundy (2015) and O'Halloran (2024).

**Aoife D. Slyne** has a primary research interest in the control of breathing in neuromuscular disorders, including Duchenne muscular dystrophy. Aoife was awarded an Eli Lilly doctoral research scholarship to pursue postgraduate training at University College Cork, Ireland. The studies described in this article were performed as part of her doctoral training. The long-term goal of Aoife's work is to aid in the development of novel therapeutic strategies to protect and improve breathing in neuromuscular disease.

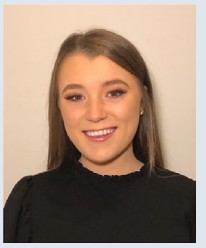

All procedures in live animals were performed under individual authorisation by the national regulatory authority by ADS (AE19130/I408), DPB (AE19130/I040) and KDO'H (AE19130/I126).

### Experimental animals

Male and female wild-type (C57BL/10ScSnJ) and *mdx* (C57BL/10ScSn-Dmd$^{mdx}$/J) mice were purchased from The Jackson Laboratory (Bar Harbor, ME, USA) and bred at University College Cork's specific pathogen-free animal housing facility. Animals were housed in individually ventilated cages in temperature- and humidity-controlled rooms, operating under a 12:12 h light/dark photocycle with food and water available *ad libitum*. Colonies were established and maintained in our facility. Studies were performed in male mice: 4-month-old wild-type ($n = 49$) and *mdx* ($n = 56$) and 16-month-old wild-type ($n = 30$) and *mdx* ($n = 29$) were studied for a comprehensive assessment of respiratory performance, including measurement of breathing and metabolism, *in vivo* respiratory muscle EMG activities, thoracic inspiratory pressure, ventilation and *ex vivo* muscle function. Respiratory muscle tissue was harvested for structural and molecular analyses using histology, immunohistochemistry and cytokine assays.

### Respiratory recordings

Respiratory flow measurements were recorded in unrestrained, unanaesthetised mice using whole-body plethysmography. Four- and 16-month-old wild-type ($n = 12$–15 per group) and *mdx* ($n = 12$–15 per group) mice were introduced into plethysmograph chambers (Model PLY4211; volume = 600 mL; Buxco Research Systems, Wilmington, NC, USA) with room air passing through each chamber. Mice were allowed a minimum of 60 min to acclimatise to the chamber environment until sufficiently settled. Following exploration and grooming behaviours, mice settled and were studied during quiet rest.

**Experimental protocol.** Following acclimatisation to the chambers, during a confirmed period of rest, a 10 min baseline recording was performed in normoxia ($F_{IO_2} = 0.21$). This was followed by a 10 min hypoxic-hypercapnic challenge ($F_{IO_2} = 0.10$ and $F_{ICO_2} = 0.06$), to stimulate both central and peripheral chemoreceptors, allowing maximum chemoactivated breathing to be examined. Respiratory parameters including respiratory frequency ($f_R$), tidal volume ($V_T$), minute ventilation ($\dot{V}_I$), inspiratory time ($T_i$), expiratory time ($T_e$), peak inspiratory flow (PIF) and peak expiratory flow (PEF) were recorded on a breath-by-breath basis for

analysis offline. Mice were subsequently anaesthetised with 5% isoflurane in air and killed by anaesthetic (pentobarbital) overdose (single bolus of 70 mg in 350 μL equivalent to 1.4–2.9 g kg$^{-1}$ I.P.). Muscles were harvested for *ex vivo* functional, structural and molecular analyses.

**Data analysis.** Ventilation during normoxia was determined as an average of the 10 min baseline period. Ventilation during hypoxic-hypercapnia was determined during steady-state conditions in the last 5 min of the challenge. $V_T$ and $\dot{V}_I$ were normalised for body mass (g).

### Metabolism measurements

O$_2$ consumption ($\dot{V}_{O_2}$) and CO$_2$ production ($\dot{V}_{CO_2}$) were measured in 4- and 16-month-old wild-type ($n = 9$–15 per group) and *mdx* ($n = 10$–15 per group) mice during the whole-body plethysmography protocol. Airflow through the chamber was maintained at 1.4 L min$^{-1}$. Fractional concentrations of O$_2$ and CO$_2$ were measured in the air entering and exiting the plethysmography chambers (O$_2$ and CO$_2$ analyser; ADInstruments, Colorado Springs, CO, USA), similar to that previously described by Burns, Roy, et al. (2017).

**Data analysis.** Calculation of $\dot{V}_{O_2}$ and $\dot{V}_{CO_2}$ was performed as previously described by Haouzi et al. (2009). During normoxia, $\dot{V}_{O_2}$ and $\dot{V}_{CO_2}$ were calculated per minute for a 10 min period and averaged. During the hypoxic-hypercapnic challenge, $\dot{V}_{O_2}$ and $\dot{V}_{CO_2}$ were calculated per minute during steady-state conditions in the last 5 min of the challenge and averaged. $\dot{V}_{O_2}$ and $\dot{V}_{CO_2}$ were normalised for body mass (g).

### Respiratory muscle EMG, oesophageal pressure and ventilatory recordings

**Anaesthetic protocol.** Anaesthesia was induced in 4- and 16-month-old wild-type ($n = 15$ per group) and *mdx* ($n = 14$–15 per group) mice with 5% isoflurane in 60% O$_2$ (balance N$_2$) in an anaesthetic induction chamber. Mice were subsequently placed in the supine position and received 2% isoflurane in 60% O$_2$ (balance N$_2$) via nose cone. Mice were gradually weaned from isoflurane and transferred to urethane (1.7 g kg$^{-1}$ I.P. in total, given as three injections) over a 25 min period. The above anaesthetic protocol was sufficient to maintain an acceptable surgical plane of anaesthesia, determined by the absence of tachypnoea and an absent pedal withdrawal reflex and cardiac and respiratory frequency response to noxious pinch. Supplemental anaesthetic was administered over the course of the experimental protocol as required to ensure an adequate depth of anaesthesia, determined by the absence of palpebral and withdrawal

reflexes and spontaneous whisking, and stability of cardio-respiratory recordings. The final cumulative dose of urethane administered was 1.7–2.3 g kg$^{-1}$ I.P. Body temperature was maintained at 37°C via a rectal probe and thermostatically controlled heating blanket (Harvard Apparatus, Holliston, MA, USA) for the duration of the experiment. A pulse oximeter clip (MouseOx; Starr Life Sciences Corporation, Oakmount, PA, USA) was placed on a shaved thigh for the measurement of peripheral capillary $O_2$ saturation ($S_{pO_2}$).

**Surgical preparation.** Following anaesthesia, a mid-cervical tracheotomy was performed. All animals were maintained with a bias flow of supplemental $O_2$ ($F_{IO_2} = 0.60$) under baseline conditions. Additional $O_2$ was administered as required. A pressure-tip catheter (Mikro-Tip; Millar Inc., Houston, TX, USA) was positioned in the thoracic oesophagus through the mouth, allowing oesophageal pressure to be measured as an index of intrapleural sub-atmospheric pressure generated by the respiratory musculature during inspiration. The catheter was advanced into the stomach to record positive pressure swings during inspiration and then withdrawn into the lower oesophagus where stable phasic sub-atmospheric pressure deflections during inspiration were observed. For the measurement of ventilation, a pneumotachometer was connected to the tracheal cannula for continuous recording of respiratory airflow. For the measurement of respiratory muscle EMG activities, concentric needle monopolar recording electrodes (26G; Natus Manufacturing Ltd., Galway, Ireland) were inserted into the middle costal region of the diaphragm on the right-hand side, second to fourth rostro-ventral external intercostal space, second to third parasternal intercostal space, superficial upper left trapezius, mid-belly of the sternomastoid, mid-belly of the cleidomastoid and mid-belly of the scalene for the continuous measurement of EMG activity. EMG signals were amplified (5000×), filtered (500 Hz low cut-off to 5000 Hz high cut-off) and integrated (50 ms time constant; Neurolog system; Digitimer Ltd, Welwyn Garden City, UK). All signals were passed through an analogue-to-digital converter (Powerlab r8/30; ADInstruments, Colorado Springs, CO, USA) and were acquired using Labchart 8 (ADInstruments) at a sampling frequency of 20 kHz (EMG) or 1 kHz (other parameters).

**Experimental protocol.** Following instrumentation, spontaneously breathing mice were allowed to stabilise before initial baseline parameters were recorded. Mice were then challenged with hypoxic-hypercapnia ($F_{IO_2} = 0.15$ and $F_{ICO_2} = 0.06$) for 2 min (4-month-old mice) or 90 s (16-month-old mice) to examine the

effects of chemostimulation on respiratory muscle EMG activities, inspiratory pressure and ventilation. Mice were allowed to stabilise post-challenge for 10 min before they underwent bilateral vagotomy by sectioning of the vagi at the mid-cervical level. Mice were allowed to stabilise post-vagotomy for a minimum of 10 min, before post-vagotomy baseline parameters were recorded. Mice were subsequently challenged with hypoxic-hypercapnia ($F_{IO_2} = 0.15$ and $F_{ICO_2} = 0.06$) for 2 min to examine the effects of chemostimulation on respiratory muscle EMG, inspiratory pressure and ventilation following vagotomy. It is established that chemostimulation in spontaneously breathing mice elicits a significantly greater phrenic motor response following vagotomy, in comparison to vagi intact (Kline et al., 2002). Maximal inspiratory pressure and respiratory muscle EMG activities were examined during sustained tracheal occlusion. Peak inspiratory pressures and respiratory EMG activities (which coincide with the nadir pressure achieved during sustained tracheal occlusion) were identified ahead of task failure, that is, the inability to sustain peak inspiratory pressures. Death was confirmed by cervical dislocation.

**Data analysis.** The amplitudes of integrated inspiratory respiratory muscle EMG activities, peak inspiratory sub-atmospheric oesophageal pressure and tidal volume were analysed and averaged for 1 min of steady-state baseline conditions immediately preceding the onset of each chemostimulation challenge. The amplitudes of integrated inspiratory respiratory muscle EMG activities, peak inspiratory sub-atmospheric oesophageal pressure and tidal volume were analysed and averaged for the final 30 s of each chemostimulation challenge. To determine maximal EMG activities, the amplitudes of integrated inspiratory respiratory muscle EMG activities were analysed during a 30 s plateau during sustained tracheal occlusion. To determine peak pressure-generating capacity, sub-atmospheric oesophageal pressure was analysed for the five successive maximal sustained efforts (maximal response) of the sustained airway occlusion challenge. Data for respiratory muscle EMG activities and oesophageal pressure are reported in absolute units (EMG, mV·s; oesophageal presssure, cmH₂O). During baseline and chemoactivated breathing, peak inspiratory and expiratory flows were measured and inspiratory tidal volume was derived from the integral of tracheal airflow measurements. Tidal volume and minute ventilation are normalised for body mass (g).

### *Ex vivo* respiratory muscle function analysis

Our experimental set-up and approach allowed for the assessment of two muscles in each animal. Four cohorts of mice underwent *ex vivo* muscle function

analysis. In cohort 1, diaphragm and scalene muscle function were examined in 4-month-old wild-type ($n = 9$–12) and *mdx* ($n = 8$–15) mice. In cohort 2, parasternal and cleidomastoid muscle function was examined in 4-month-old wild-type ($n = 9$–11) and *mdx* ($n = 8$–11) mice. In cohort 3, sternomastoid muscle function was examined in 4-month-old wild-type ($n = 11$) and *mdx* ($n = 15$) mice. In cohort 4, diaphragm and scalene muscle function were examined in 16-month-old wild-type ($n = 11$–15) and *mdx* ($n = 13$–14) mice. A limited availability of 16-month-old *mdx* mice necessitated studies of fewer muscles. We elected to study diaphragm as the major muscle of breathing and scalene given that it is an important accessory muscle of breathing *per se* and given that it showed normal forces at 4 months of age in *mdx* mice, allowing us to examine our hypothesis that accessory muscle dysfunction (emergence and/or worsening) contributes to respiratory compromise in advanced disease. Muscles were examined *ex vivo* under isometric and isotonic conditions. Mice were anaesthetised by 5% isoflurane in air and killed by anaesthetic (pentobarbital) overdose as described above. Muscles were immediately excised and placed in a tissue bath at room temperature containing continuously gassed (95% $O_2$/5% $CO_2$) Krebs solution (in mM: NaCl, 120; KCl, 5; $Ca^{2+}$ gluconate, 2.5; $MgSO_4$, 1.2; $NaH_2PO_4$, 1.2; $NaHCO_3$, 25; and glucose, 11.5) and *D*-tubocurarine (25 μM) prior to functional analysis. Diaphragm muscle was harvested with rib and central tendon attached.

Parasternal, sternomastoid, cleidomastoid and scalene muscles were prepared with muscle fibres arranged longitudinally and suspended vertically between two platinum plate electrodes. One end was fixed to an immobile hook and the other end attached to a dual–mode lever transducer system (Aurora Scientific Inc, Aurora, ON, Canada) by non-elastic string. A section of diaphragm muscle with longitudinally arranged muscle fibres was prepared for functional assessment. Each diaphragm muscle strip was suspended vertically between two platinum plate electrodes. Using non-elastic string, the rib was sutured to an immobile hook and the central tendon was attached to a dual-mode force transducer. Muscle preparations were studied in a water jacketed tissue bath at 35°C containing Krebs solution continuously aerated with 95% $O_2$/5% $CO_2$. The contralateral muscles were harvested and either mounted at resting length on a cube of liver and embedded in optimal cutting temperature (OCT) embedding medium (VWR International, Dublin, Ireland) for cryoprotection and then frozen in isopentane (Sigma-Aldrich, Wicklow, Ireland) cooled on dry ice for structural analysis or snap frozen in liquid nitrogen for subsequent molecular analysis. Muscles were stored at –80°C, until undergoing structural or molecular analyses.

**Experimental protocol.** Muscle preparations were allowed a 10-min equilibration period. The optimum length ($L_o$) of the muscle was determined by adjusting the length of the muscle preparations, using a micro-positioner, between intermittent twitch contractions (Burns, Ali, et al., 2017). The muscle length which revealed maximal isometric twitch force for a single isometric twitch stimulation (supramaximal stimulation, 1 ms duration) was considered $L_o$. Muscle preparations were maintained at $L_o$ for the duration of the protocol.

Three isometric twitch contractions were recorded. Peak isometric twitch force, time to peak force (TTP; contraction time) and half-relaxation time ($\frac{1}{2}$RT; time for peak force to decay by 50%) were determined. An isometric tetanic contraction was elicited by stimulating muscle strips with supramaximal voltage at 100 Hz for 300 ms duration. The force–frequency relationship was examined by stimulating the muscle sequentially at 25, 50, 75, 100, 125 and 150 Hz (300 ms train duration). Contractions were interspersed by a 1 min interval. Following the isometric protocol, isotonic properties were examined using the load clamp technique. Muscle preparations were maximally stimulated at 150 Hz (300 ms) and allowed to shorten against a fixed percentage of maximal force produced at 150 Hz (0% and 50%). An isotonic contraction was elicited in preparations at 0% load to examine maximum unloaded muscle shortening and velocity of shortening (Burns, Ali, et al., 2017). An isotonic contraction was elicited in preparations at 50% load to examine muscle work and power. An interval of 1 min was allowed between each isotonic contraction.

**Data analysis.** Muscle force was normalised for muscle cross-sectional area (CSA) and expressed as specific force (N cm$^{-2}$). The CSA of each strip was calculated by dividing muscle mass (g) by the product of muscle $L_o$ (cm) and muscle density (assumed to be 1.06 g cm$^{-3}$) (Close, 1972). TTP and $\frac{1}{2}$RT were measured as indices of isometric twitch kinetics and were expressed in ms. For isotonic contractions, total muscle shortening was considered the total distance shortened during muscle contraction at 0% load. Total muscle shortening ($S_{max}$) was determined in absolute units (cm) and was normalised to $L_o$ and expressed as $L/L_o$. Muscle shortening velocity ($V_{max}$) was determined as the distance shortened during the initial 30 ms of an unloaded contraction (Burns, Ali, et al., 2017). $V_{max}$ was determined in absolute units (cm s$^{-1}$) and was normalised to $L_o$ and expressed as $L_o/s$. Work was calculated as the product of force (N cm$^{-2}$) and distance (total muscle shortening; $L/L_o$) at 50% load. Power was calculated as the product of force (N cm$^{-2}$) and velocity of shortening ($L_o/s$) at 50% load. Technical difficulties arose with inconsistent shortening at 50% load in parasternal and accessory muscles. The tests

were discontinued and data for these parameters are not reported.

## Muscle immunofluorescence and histology

**Tissue preparation.** A portion of diaphragm, intercostal, parasternal, sternomastoid, cleidomastoid, scalene, and trapezius muscles were excised from 4- and 16-month-old wild-type ($n = 6$ per group) and *mdx* ($n = 6$ per group) mice and mounted at resting length on cubes of liver. Muscle samples were embedded in OCT for cryoprotection and then frozen in isopentane (Sigma-Aldrich) cooled on dry ice. Samples were stored at $-80°C$ for subsequent structural analysis. Serial transverse muscle sections (10 μm) were cut using a cryostat (Leica CM3050; Leica Microsystems, Nussloch, Germany) at $-20°C$ and four sections per animal per muscle per region were mounted serially across polylysine-coated glass slides (VWR International) allowing for a distribution of tissue on a given slide. Two slides per animal from two distinct regions of muscle containing a minimum of four sections per slide were processed for 4-month-old wild-type and *mdx* histology and immunofluorescence. Given that no difference was observed between regions and data were pooled, only one region of muscle was examined for 16-month-old animals. Therefore, a total of eight tissue sections per animal were examined in 4-month-old wild-type and *mdx* muscle and a total of four tissue sections per animal were examined in 16-month-old wild-type and *mdx* muscle.

**Histological analysis.** To examine putative inflammatory cell infiltration of muscle fibres, tissue sections were stained with haematoxylin and eosin (H&E) using an autostainer (Leica ST5010 Autostainer XL; Leica Microsystems). For collagen staining, picro-sirius red (Leica Biosystems, Wetzlar, Germany) staining was completed. Slides were mounted using DPX mounting medium (Sigma-Aldrich), air-dried and visualised on a bright field microscope (Olympus BX51; Olympus, Tokyo, Japan) at ×10 (picro-sirius red) and ×20 (H&E) magnification.

**Data analysis.** Muscle histology was scored using ImageJ (NIH, Bethesda, MD, USA). Putative inflammatory cell infiltration (the presence of cells in the extracellular matrix) was scored and expressed as a percentage of the total area of muscle. A square test frame (400 × 400 μm) with inclusion and exclusion boundaries, was placed randomly over each image. The number of myofibres displaying central nucleation was expressed as a percentage of the total number of myofibres per image. The average total number of fibres counted per region for 4-month-old wild-type muscles were: 277 ± 64, mean ± SD (diaphragm), 160 ± 70 (intercostal), 81 ± 10 (parasternal), 89 ± 49 (sternomastoid), 92 ± 13 (cleidomastoid), 117 ± 37 (scalene) and 111 ± 7 (trapezius). The average total number of fibres counted per region for 4-month-old *mdx* muscles were: 356 ± 115, mean ± SD (diaphragm), 309 ± 46 (intercostal), 104 ± 27 (parasternal), 139 ± 64 (sternomastoid), 120 ± 22 (cleidomastoid), 138 ± 41 (scalene), and 144 ± 51 (trapezius). The average total number of fibres counted per region for 16-month-old wild-type muscles were: 242 ± 30, mean ± SD (diaphragm), 181 ± 34 (intercostal), 91 ± 13 (parasternal), 108 ± 32 (sternomastoid), 91 ± 18 (cleidomastoid), 128 ± 19 (scalene) and 134 ± 33 (trapezius). The average total number of fibres counted per region for 16-month-old *mdx* muscles were: 228 ± 25, mean ± SD (diaphragm), 280 ± 41 (intercostal), 202 ± 43 (parasternal), 213 ± 50 (sternomastoid), 192 ± 69 (cleidomastoid), 241 ± 20 (scalene) and 188 ± 38 (trapezius). For slides stained with picro-sirius red, the microscope lighting exposure was standardised during imaging. Images were analysed using a colour balance threshold and the area of collagen was expressed as a percentage of the total area of muscle. A square test frame (600 × 600 μm) with inclusion and exclusion boundaries was placed randomly over each image. Data generated from multiple images was averaged per animal before computing group means.

**Laminin immunofluorescence.** Slides were immersed in phosphate-buffered saline (PBS) (0.01 м) containing 1% bovine serum albumin (BSA) for 15 min. After 3 × 5 min PBS washes, slides were immersed in PBS containing 5% normal goat serum (Sigma-Aldrich) for 1 h. Slides then underwent a further 3 × 5 min PBS washes prior to application of the primary antibody (rabbit anti-laminin; dilution 1:100; L9393; Sigma-Aldrich), diluted in PBS and 1% BSA. Slides were incubated overnight at 4°C in a humidity chamber. After the incubation period, slides were washed with PBS for 3 × 5 min before the secondary antibody (FITC-conjugated goat anti-rabbit; dilution 1:100; F9887; Sigma-Aldrich), diluted in PBS and 1% BSA, was applied. Slides were incubated for 1 h in the dark at room temperature. Slides were washed with PBS for 3 × 5 min and cover-slipped with poly-vinyl alcohol mounting medium with DABCO anti-fade (Sigma-Aldrich) before observation with a fluorescence microscope (Olympus BX51). Muscle sections were viewed at 10× magnification and images captured using an Olympus BX51 microscope and a DP71 camera (Olympus).

**Data analysis.** For measurements, a square test frame (600 × 600 μm) with inclusion and exclusion boundaries,

was placed randomly over each image. To determine the size distribution of muscle fibres within muscles, the individual fibre boundaries were determined using ImageJ. From this, minimum Feret's minimal diameters were determined. Data generated from multiple images were averaged per animal before computing group means. The average total number of fibres counted per region for 4-month-old wild-type muscles were: 691 ± 244, mean ± SD (diaphragm), 652 ± 265 (intercostal), 351 ± 86 (parasternal), 275 ± 92 (sternomastoid), 291 ± 40 (cleidomastoid), 333 ± 74 (scalene) and 345 ± 123 (trapezius). The average total number of fibres counted per region for 4-month-old *mdx* muscles were: 1280 ± 150, mean ± SD (diaphragm), 1554 ± 418 (intercostal), 442 ± 101 (parasternal), 524 ± 155 (sternomastoid), 306 ± 86 (cleidomastoid), 533 ± 85 (scalene) and 442 ± 186 (trapezius). The average total number of fibres counted per region for 16-month-old wild-type muscles were: 733 ± 149, mean ± SD (diaphragm), 874 ± 80 (intercostal), 397 ± 66 (parasternal), 446 ± 138 (sternomastoid), 427 ± 51 (cleidomastoid), 522 ± 103 (scalene) and 470 ± 82 (trapezius). The average total number of fibres counted per region for 16-month-old *mdx* muscles were: 744 ± 106, mean ± SD (diaphragm), 1116 ± 295 (intercostal), 711 ± 213 (parasternal), 757 ± 170 (sternomastoid), 618 ± 219 (cleidomastoid), 750 ± 83 (scalene) and 777 ± 122 (trapezius).

### Molecular analysis

**Tissue preparation.** Diaphragm, intercostal, parasternal, sternomastoid, scalene and trapezius muscle samples of 4- and 16-month-old wild-type and *mdx* mice ($n = 10$ per group) were removed from storage at −80°C, weighed and homogenised on ice, in ice-cold modified radioimmunoprecipitation assay (1X RIPA) buffer, consisting of 10X RIPA, deionised $H_2O$, 200 mM sodium fluoride (NaF), 100 mM phenylmethylsulfonylfluoride, 1X protease cocktail inhibitor and 200 mM sodium orthovanadate (Sigma-Aldrich) at a 5% w/v ratio using a general laboratory bead homogeniser (4 × 10 s bursts; Fisherbrand; Fisher Scientific, Dublin, Ireland). Homogenates were allowed 20 min of lyse time on ice with intermittent vortexing. Homogenates were centrifuged in a U-320R centrifuge (15,366 g at 4°C for 20 min) to separate insoluble cellular fractions from protein homogenates. The clear protein-containing supernatant was removed and the protein concentration of each sample was quantified using a bicinchoninic acid (BCA) assay (Pierce Biotechnology, Thermo Scientific, Dublin, Ireland). Homogenates were stored at –80°C until further use.

**Cytokine multiplex assay.** A multiplex cytokine assay (K15255D; Meso Scale Discovery, Rockville, MD, USA) was used to examine cytokine concentrations in diaphragm, intercostal, parasternal, sternomastoid, scalene and trapezius muscles of 4- and 16-month-old wild-type and *mdx* mice ($n = 10$ per group). The assay was performed in accordance with the manufacturer's instructions, with an extended incubation time (overnight at 4°C) to improve the detection of cytokines in muscle homogenates (Burns et al., 2018; Burns, Drummond, et al., 2019). Following incubation, the plate was read on a Quickplex SQ 120 imager (Meso Scale Discovery). Values are expressed as pg mg$^{-1}$ total protein homogenate, previously determined by BCA assay.

### Statistical analysis

Values are expressed as the mean ± SD in tables and as box and whisker plots (median, 25–75th percentile, minimum and maximum values and scatter plot) or bar graphs (mean, SD and scatter plot) in graphs. The Kolmogorov–Smirnov test was used to test for normality of the distribution in all data sets with visual inspection of Q-Q plots. Data were statistically analysed using Prism, version 10 (Graphpad Software Inc., San Diego, CA, USA). Additionally, mean group values are expressed in fibre size distribution curves and heat maps, for illustrative purposes.

For measures of ventilatory and metabolic responsiveness to chemostimulation in conscious mice, inspiratory pressure, EMG activities and ventilation during chemostimulation and airway occlusion in anaesthetised mice and respiratory muscle force–frequency relationship in 4- and 16-month-old wild-type and *mdx* groups, all data were statistically compared by two-way mixed analysis of variance (ANOVA) with genotype as the between-subjects factor and gas (plethysmography), time (EMG) or stimulus frequency (muscle function) as the within-subjects factors, followed by Bonferroni *post hoc* tests. A mixed-effects model was used when data points were missing for technical reasons with genotype as the fixed factor and each mouse as the random factor (as the inference was for all mice and not just the mice studied). Data that were not normally distributed were log$_{10}$ transformed prior to statistical treatment to better approximate normality. Sphericity was not assumed; therefore, the Greenhouse–Geisser correction was applied in all tests.

For measures of respiratory muscle function (twitch force, TTP, $\frac{1}{2}$RT, tetanic force, $S_{max}$, $V_{max}$, work, power and $L_o$), respiratory muscle structure and respiratory muscle cytokine concentrations, all data for 4- and 16-month-old wild-type and *mdx* groups were tested for normal

distribution and equal variances. Data sets that were normally distributed and of equal variance were compared statistically using an unpaired two-tailed Student's *t* test. Welch's correction was applied in cases of unequal variance. Data sets that were not normally distributed were compared using Mann Whitney non-parametric tests.

Exact *P* values are reported for all comparisons. $P < 0.05$ was considered statistically significant for most tests. A more conservative $P < 0.05/17$ (i.e. $P < 0.003$) was considered for cytokine analysis by applying the Bonferroni adjustment for multiple comparisons.

## Results

### Baseline ventilation and ventilatory responsiveness to hypoxic-hypercapnia in conscious 4- and 16-month-old mice

Representative respiratory flow traces for 4- and 16-month-old wild-type and *mdx* mice during normoxia ($F_{IO_2} = 0.21$) and hypoxic-hypercapnia ($F_{IO_2} = 0.10$ and $F_{ICO_2} = 0.06$) are shown in Fig. 1*A*. Respiratory and metabolic parameters for 4-month-old wild-type and *mdx* mice are shown in Table 1. Respiratory and metabolic parameters for 16-month-old wild-type and *mdx* mice are shown in Table 2.

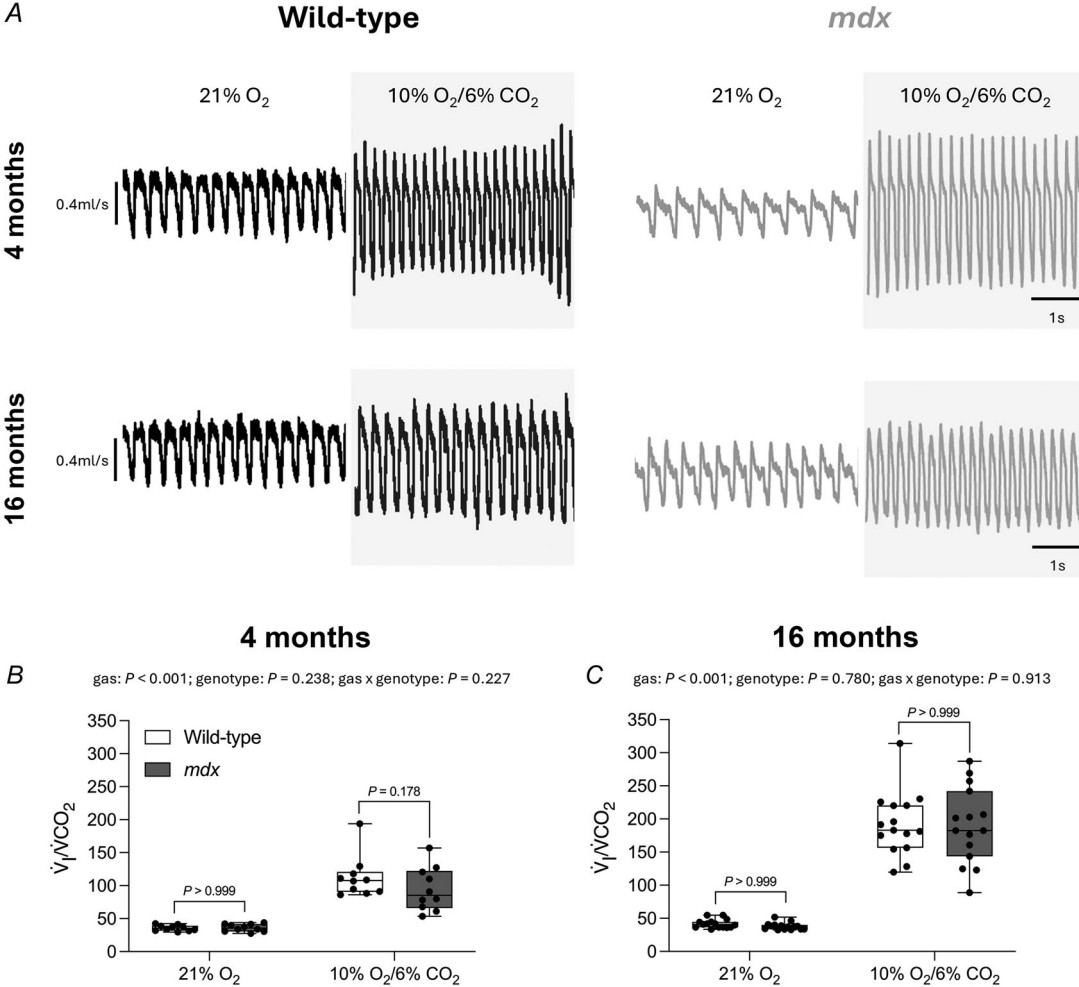

**Figure 1. Ventilation during normoxia and in response to chemostimulation in conscious 4- and 16-month-old wild-type and *mdx* mice**

*A*, representative respiratory flow traces of 4- and 16-month-old wild-type (black) and *mdx* (grey) mice during normoxia (21% $O_2$) and hypoxic-hypercapnia (10% $O_2$/6% $CO_2$; shaded background); inspiration downwards. *B* and *C*, group data for ventilatory equivalent for $CO_2$ ($V_I/\dot{V}_{CO_2}$) in conscious 4- ($n = 9$–12 per group) (*B*) and 16- ($n = 15$ per group) (*C*) month-old wild-type (open) and *mdx* (grey) mice during normoxia and hypoxic-hypercapnia, as measured by whole-body plethysmography. Values are expressed as box and whisker plots (median, 25–75th centiles, minimum to maximum values and scatter plot). Data were statistically compared by two-way mixed ANOVA (gas **x** genotype) with Bonferroni's multiple comparison *post hoc* test. Adjusted *P* values for all comparisons are reported.

**Table 1. Ventilation and metabolism during normoxia and in response to chemostimulation in conscious 4-month-old wild-type and *mdx* mice**

| | 21% $O_2$ ($n = 9$–12) | | 10% $O_2$/6% $CO_2$ ($n = 9$–12) | | Adjusted *P* value | | |
|---|---|---|---|---|---|---|---|
| | Wild-type | *mdx* | Wild-type | *mdx* | gas | genotype | gas × genotype |
| $f_R$ (breaths min$^{-1}$) | $149.2 \pm 18.9$ | $137.8 \pm 12.9$ | $334.9 \pm 34.2$ | $293.1 \pm 42.4$ $^{\epsilon\epsilon}$ | **<0.001** | **0.006** | 0.082 |
| $V_T$ (mL g$^{-1}$) | $0.0061 \pm 0.0011$ | $0.0063 \pm 0.0005$ | $0.0137 \pm 0.0013$ | $0.0141 \pm 0.0023$ | **<0.001** | 0.612 | 0.710 |
| $V_I$ (mL min$^{-1}$ g$^{-1}$) | $0.90 \pm 0.18$ | $0.86 \pm 0.09$ | $4.55 \pm 0.79$ | $4.12 \pm 1.08$ | **<0.001** | 0.273 | 0.302 |
| PIF (mL s$^{-1}$) | $2.3 \pm 0.3$ | $2.7 \pm 0.4$ | $8.1 \pm 1.7$ | $8.1 \pm 2.3$ | **<0.001** | 0.619 | 0.653 |
| PEF (mL s$^{-1}$) | $1.5 \pm 0.2$ | $2.0 \pm 0.3$ | $6.7 \pm 1.6$ | $7.4 \pm 1.8$ | **<0.001** | 0.097 | 0.786 |
| $T_i$ (s) | $0.14 \pm 0.01$ | $0.14 \pm 0.01$ | $0.08 \pm 0.01$ | $0.09 \pm 0.02$ | **<0.001** | 0.101 | 0.070 |
| $T_e$ (s) | $0.29 \pm 0.05$ | $0.31 \pm 0.04$ | $0.11 \pm 0.01$ | $0.12 \pm 0.02$ | **<0.001** | **0.046** | 0.644 |
| $\dot{V}_{O_2}$ (mL min$^{-1}$ g$^{-1}$) | $0.032 \pm 0.008$ | $0.036 \pm 0.005$ | $0.037 \pm 0.013$ | $0.043 \pm 0.021$ | 0.112 | 0.163 | 0.586 |
| $\dot{V}_{CO_2}$ (mL min$^{-1}$ g$^{-1}$) | $0.025 \pm 0.005$ | $0.024 \pm 0.003$ | $0.042 \pm 0.010$ | $0.046 \pm 0.016$ | **<0.001** | 0.287 | 0.469 |
| $\dot{V}_{CO_2}/\dot{V}_{O_2}$ | $0.83 \pm 0.20$ | $0.69 \pm 0.13$ | $1.2 \pm 0.4$ | $1.2 \pm 0.4$ | **<0.001** | 0.175 | 0.454 |
| $V_I/\dot{V}_{O_2}$ | $28.8 \pm 7.5$ | $24.5 \pm 3.8$ | $134.1 \pm 31.8$ | $103.5 \pm 38.3$ $^{\epsilon}$ | **<0.001** | **0.041** | 0.119 |

Group data (mean ± SD) for respiratory frequency ($f_R$), tidal volume ($V_T$), minute ventilation ($V_I$), peak inspiratory flow (PIF), peak expiratory flow (PEF), inspiratory time ($T_i$), expiratory time ($T_e$), oxygen consumption ($\dot{V}_{O_2}$), carbon dioxide production ($\dot{V}_{CO_2}$), respiratory exchange ratio ($\dot{V}_{CO_2}/\dot{V}_{O_2}$), and ventilatory equivalent for $O_2$ ($V_I/\dot{V}_{O_2}$) in 4-month-old wild-type and *mdx* mice during normoxia (21% $O_2$) and hypoxic-hypercapnia (10% $O_2$/6% $CO_2$). Data were statistically compared by two-way mixed ANOVA (gas × genotype) with Bonferroni's multiple comparison *post hoc* test. Data sets that were not normally distributed ($T_i$ and $\dot{V}_{CO_2}/\dot{V}_{O_2}$) were log$_{10}$ transformed prior to statistical analysis. $^{\epsilon}$ Denotes 4-month-old *mdx* different from corresponding wild-type group during chemochallenge; $^{\epsilon}P = 0.021$ and $^{\epsilon\epsilon}P = 0.002$ (*post hoc* adjusted *P* values).

**Table 2. Ventilation and metabolism during normoxia and in response to chemostimulation in conscious 16-month-old wild-type and *mdx* mice**

| | 21% $O_2$ ($n = 15$) | | 10% $O_2$/6% $CO_2$ ($n = 15$) | | Adjusted *P* value | | |
|---|---|---|---|---|---|---|---|
| | Wild-type | *mdx* | Wild-type | *mdx* | gas | genotype | gas × genotype |
| $f_R$ (breaths min$^{-1}$) | $137.0 \pm 14.4$ | $153.6 \pm 17.1$ | $286.9 \pm 40.1$ | $274.7 \pm 24.3$ | **<0.001** | 0.739 | 0.050 |
| $V_T$ (mL$^{-1}$ g$^{-1}$) | $.0073 \pm 0.0014$ | $0.0068 \pm 0.0009$ | $0.0124 \pm 0.0019$ | $0.0112 \pm 0.0016$ | **<0.001** | 0.729 | 0.484 |
| $V_I$ (mL min$^{-1}$ g$^{-1}$) | $0.99 \pm 0.23$ | $1.04 \pm 0.20$ | $3.48 \pm 0.72$ | $3.04 \pm 0.57$ | **<0.001** | 0.568 | 0.054 |
| PIF (mL s$^{-1}$) | $3.2 \pm 0.6$ | $2.8 \pm 0.4$ | $7.9 \pm 1.7$ | $5.6 \pm 0.8$ $^{\$\$\$\$}$ | **<0.001** | **<0.001** | 0.054 |
| PEF (mL s$^{-1}$) | $2.0 \pm 0.5$ | $2.4 \pm 0.4$ $^{\&}$ | $6.9 \pm 1.8$ | $5.3 \pm 0.7$ $^{\$\$}$ | **<0.001** | **<0.001** | 0.802 |
| $T_i$ (s) | $0.16 \pm 0.02$ | $0.14 \pm 0.01$ $^{\#}$ | $0.10 \pm 0.01$ | $0.11 \pm 0.01$ | **<0.001** | 0.499 | **0.006** |
| $T_e$ (s) | $0.30 \pm 0.04$ | $0.26 \pm 0.04$ $^{\%\%}$ | $0.13 \pm 0.02$ | $0.12 \pm 0.01$ | **<0.001** | **0.008** | **0.035** |
| $\dot{V}_{O_2}$ (mL min$^{-1}$ g$^{-1}$) | $0.063 \pm 0.019$ | $0.056 \pm 0.014$ | $0.057 \pm 0.014$ | $0.049 \pm 0.013$ | **0.022** | 0.126 | 0.879 |
| $\dot{V}_{CO_2}$ (mL min$^{-1}$ g$^{-1}$) | $0.024 \pm 0.008$ | $0.028 \pm 0.006$ | $0.019 \pm 0.006$ | $0.017 \pm 0.005$ | **<0.001** | 0.731 | 0.058 |
| $\dot{V}_{CO_2}/\dot{V}_{O_2}$ | $0.39 \pm 0.08$ | $0.51 \pm 0.13$ $^{!!}$ | $0.34 \pm 0.08$ | $0.37 \pm 0.12$ | **<0.001** | **0.019** | 0.059 |
| $V_I/\dot{V}_{O_2}$ | $16.3 \pm 3.4$ | $19.3 \pm 5.0$ | $62.0 \pm 13.2$ | $67.4 \pm 17.8$ | **<0.001** | 0.211 | 0.088 |

Group data (mean ± SD) for respiratory frequency ($f_R$), tidal volume ($V_T$), minute ventilation ($V_I$), peak inspiratory flow (PIF), peak expiratory flow (PEF), inspiratory time ($T_i$), expiratory time ($T_e$), oxygen consumption ($\dot{V}_{O_2}$), carbon dioxide production ($\dot{V}_{CO_2}$), respiratory exchange ratio ($\dot{V}_{CO_2}/\dot{V}_{O_2}$) and ventilatory equivalent for $O_2$ ($V_I/\dot{V}_{O_2}$) in 16-month-old wild-type and *mdx* mice during normoxia (21% $O_2$) and hypoxic-hypercapnia (10% $O_2$/6% $CO_2$). Data were statistically compared by two-way mixed ANOVA (gas × genotype) with Bonferroni's multiple comparison *post hoc* test. Data sets that were not normally distributed ($V_T$, $V_I$, PIF, PEF, $T_i$ and $V_I/\dot{V}_{O_2}$) were log$_{10}$ transformed prior to statistical analysis. $^{\&, \#, \% \text{ and } !}$ Denote 16-month-old *mdx* different from corresponding wild-type group during normoxic conditions; $^{\&}P = 0.012$, $^{\#}P = 0.011$, $^{\%\%}P = 0.001$, $^{!!}P = 0.005$ (*post hoc* adjusted *P* values). $^{\$}$ Denotes 16-month-old *mdx* different from corresponding wild-type group during chemochallenge; $^{\$\$}P = 0.003$ and $^{\$\$\$\$}P < 0.001$ (*post hoc* adjusted *P* values).

Chemostimulation with hypoxic-hypercapnia resulted in a significant increase in $f_R$ (gas: $P < 0.0001$; two-way mixed ANOVA), $V_T$ (gas: $P < 0.001$), $\dot{V}_I$ (gas: $P < 0.001$), PIF (gas: $P < 0.001$), PEF (gas: $P < 0.001$), $\dot{V}_{CO_2}$ (gas: $P < 0.001$), $\dot{V}_{CO_2}/\dot{V}_{O_2}$ (gas: $P < 0.001$), $\dot{V}_I/\dot{V}_{O_2}$ (gas: $P < 0.001$) and $\dot{V}_I/\dot{V}_{CO_2}$ (gas: $P < 0.001$) for both wild-type and *mdx* mice at 4 months of age (Fig. 1*B* and Table 1). Chemostimulation with hypoxic-hypercapnia resulted in a significant decrease in $T_i$ (gas: $P < 0.0001$; two-way mixed ANOVA) and $T_e$ (gas: $P < 0.001$) for both wild-type and *mdx* mice at 4 months of age (Table 1).

Chemostimulation with hypoxic-hypercapnia resulted in a significant increase in $f_R$ (gas: $P < 0.001$; two-way mixed ANOVA), $V_T$ (gas: $P < 0.001$), $\dot{V}_I$ (gas: $P < 0.001$), PIF (gas: $P < 0.001$), PEF (gas: $P < 0.001$), $\dot{V}_I/\dot{V}_{O_2}$ (gas: $P < 0.001$) and $\dot{V}_I/\dot{V}_{CO_2}$ (gas: $P < 0.001$) for both wild-type and *mdx* mice at 16 months of age (Fig. 1*C* and Table 2). Chemostimulation with hypoxic-hypercapnia resulted in a significant decrease in $T_i$ (gas: $P < 0.001$; two-way mixed ANOVA), $T_e$ (gas: $P < 0.001$), $\dot{V}_{O_2}$ (gas: $P = 0.022$), $\dot{V}_{CO_2}$ (gas: $P < 0.001$) and $\dot{V}_{CO_2}/\dot{V}_{O_2}$ (gas: $P < 0.001$) for both wild-type and *mdx* mice at 16 months of age (Table 2).

Ventilation was equivalent between wild-type and *mdx* mice during normoxia and hypoxic-hypercapnia at 4 months (Fig. 1*B*) and 16 months (Fig. 1*C*) of age, as measured by the ventilatory equivalent for $CO_2$ ($\dot{V}_I/\dot{V}_{CO_2}$).

$f_R$ and $\dot{V}_I/\dot{V}_{O_2}$ were lower in *mdx* mice compared to wild-type at 4 months of age (genotype: $P = 0.006$ and $P = 0.041$, respectively; two-way mixed ANOVA) (Table 1), particularly during chemochallenge ($P = 0.002$ and $P = 0.021$ respectively; Bonferroni *post hoc* test). $T_e$ was significantly higher in *mdx* mice compared to wild-type at 4 months of age (genotype: $P = 0.046$; two-way mixed ANOVA) (Table 1). $V_T$, $\dot{V}_I$, PIF, PEF, $T_i$, $\dot{V}_{O_2}$, $\dot{V}_{CO_2}$ and $\dot{V}_{CO_2}/\dot{V}_{O_2}$ were equivalent between wild-type and *mdx* mice at 4 months of age (Table 1).

PIF was significantly lower in *mdx* mice compared to wild-type at 16 months of age (genotype: $P < 0.001$; two-way mixed ANOVA) (Table 2), particularly during chemochallenge ($P < 0.001$; Bonferroni *post hoc* test). PEF was significantly altered in *mdx* mice compared to wild-type at 16 months of age (genotype: $P < 0.001$; two-way mixed ANOVA) (Table 2), with increased PEF in *mdx* mice compared to wild-type during normoxia ($P = 0.012$; Bonferroni *post hoc* test) and reduced PEF in *mdx* mice compared to wild-type during chemochallenge ($P = 0.003$; Bonferroni *post hoc* test). $T_i$ was significantly altered in *mdx* mice compared to wild-type at 16 months of age (gas × genotype: $P = 0.006$; two-way mixed ANOVA) (Table 2), with significantly reduced $T_i$ in *mdx* mice compared to wild-type during normoxia ($P = 0.011$; Bonferroni *post hoc* test). $T_e$ was significantly lower in *mdx* mice compared to wild-type at 16 months of age

(genotype: $P = 0.008$; two-way mixed ANOVA) (Table 2), particularly during normoxia ($P = 0.001$; Bonferroni *post hoc* test). $\dot{V}_{CO_2}/\dot{V}_{O_2}$ was significantly increased in *mdx* mice compared to wild-type at 16 months of age (genotype: $P = 0.019$; two-way mixed ANOVA) (Table 2), particularly during normoxia ($P = 0.005$; Bonferroni *post hoc* test). $f_R$, $V_T$, $\dot{V}_I$, $\dot{V}_{O_2}$, $\dot{V}_{CO_2}$ and $\dot{V}_I/\dot{V}_{O_2}$ were equivalent between wild-type and *mdx* mice at 16 months of age (Table 2).

### Ventilation in anaesthetised 4- and 16-month-old mice

Representative original recordings of oesophageal (inspiratory) pressure, diaphragm and scalene EMG activities and tracheal airflow during baseline conditions (60% $O_2$) at the beginning of the protocol in an anaesthetised 4-month-old wild-type mouse are shown in Fig. 2*A*. Tracheal airflow was examined during four conditions: initial baseline (vagi intact), hypoxic-hypercapnic challenge (vagi intact), baseline following vagotomy and hypoxic-hypercapnic challenge (vagotomised).

There was no significant difference in $\dot{V}_I$ between wild-type and *mdx* groups at 4 and 16 months of age (Fig. 2*B* and *E*).

PIF ($P = 0.008$; Bonferroni's multiple comparison *post hoc*) was significantly lower in 16-month-old *mdx* compared to wild-type during the highest ventilatory demand condition examined (i.e. chemochallenge following vagotomy) (Fig. 2*F*).

### Inspiratory pressure and obligatory and accessory respiratory EMG activities in anaesthetised 4- and 16-month-old mice

Representative original recordings of oesophageal (inspiratory) pressure and obligatory (diaphragm, inter-costal and parasternal) and accessory (sternomastoid, cleidomastoid, scalene and trapezius) respiratory muscle EMG activities during baseline conditions (60% $O_2$; vagi intact) and peak effort during sustained tracheal occlusion (vagotomised) in 4- and 16-month-old wild-type and *mdx* mice are shown in Fig. 3.

Inspiratory pressure and obligatory and accessory respiratory muscle EMG activities were examined during five conditions: initial baseline (vagi intact), hypoxic-hypercapnic challenge (vagi intact), baseline following vagotomy, hypoxic-hypercapnic challenge (vagotomised) and during sustained tracheal occlusion (vagotomised). Gas challenges, vagotomy and tracheal occlusion typically increased inspiratory pressure generation and EMG responsiveness for both wild-type and *mdx* groups. $S_{pO_2}$ profiles during challenges generally appeared quite similar, but statistical analyses revealed

a genotype effect, with $S_{pO_2}$ desaturation greater in wild-type compared to *mdx* at 4 months of age (genotype: $P < 0.001$; two-way mixed ANOVA), but greater in *mdx* compared to wild-type at 16 months of age (genotype: $P = 0.035$; two-way mixed ANOVA). *Post hoc* analysis revealed a group difference in 4-month-old mice during vagi intact hypoxic-hypercapnic challenge ($45 \pm 9\%$ *vs.* $58 \pm 6\%$, mean $\pm$ SD, wild-type *vs. mdx*; $P = 0.004$; Bonferroni's multiple comparison *post hoc*).

Inspiratory pressure across behaviours was lower in 4-month-old *mdx* mice compared to wild-type (genotype: $P = 0.012$; two-way mixed ANOVA) (Fig. 4*A*), but peak inspiratory pressure during airway occlusion was equivalent between genotypes. Inspiratory pressure across behaviours was lower in 16-month-old *mdx*

mice compared to wild-type (genotype: $P = 0.006$; two-way mixed ANOVA) (Fig. 4*E*). Peak inspiratory pressure was significantly reduced in 16-month-old *mdx* mice compared to wild-type during airway occlusion ($P = 0.015$; Bonferroni's multiple comparison *post hoc*) (Fig. 4*E*).

Diaphragm EMG activity was significantly lower in 4-month-old *mdx* mice compared to wild-type (genotype: $P < 0.001$; two-way mixed ANOVA) (Fig. 4*B*) during all conditions examined (baseline: $P = 0.004$; hypoxic-hypercapnic challenge: $P = 0.009$; vagotomy: $P = 0.023$; hypoxic-hypercapnic challenge following vagotomy: $P = 0.017$; airway occlusion: $P = 0.013$; Bonferroni's multiple comparison *post hoc*). Sternomastoid EMG activity was significantly

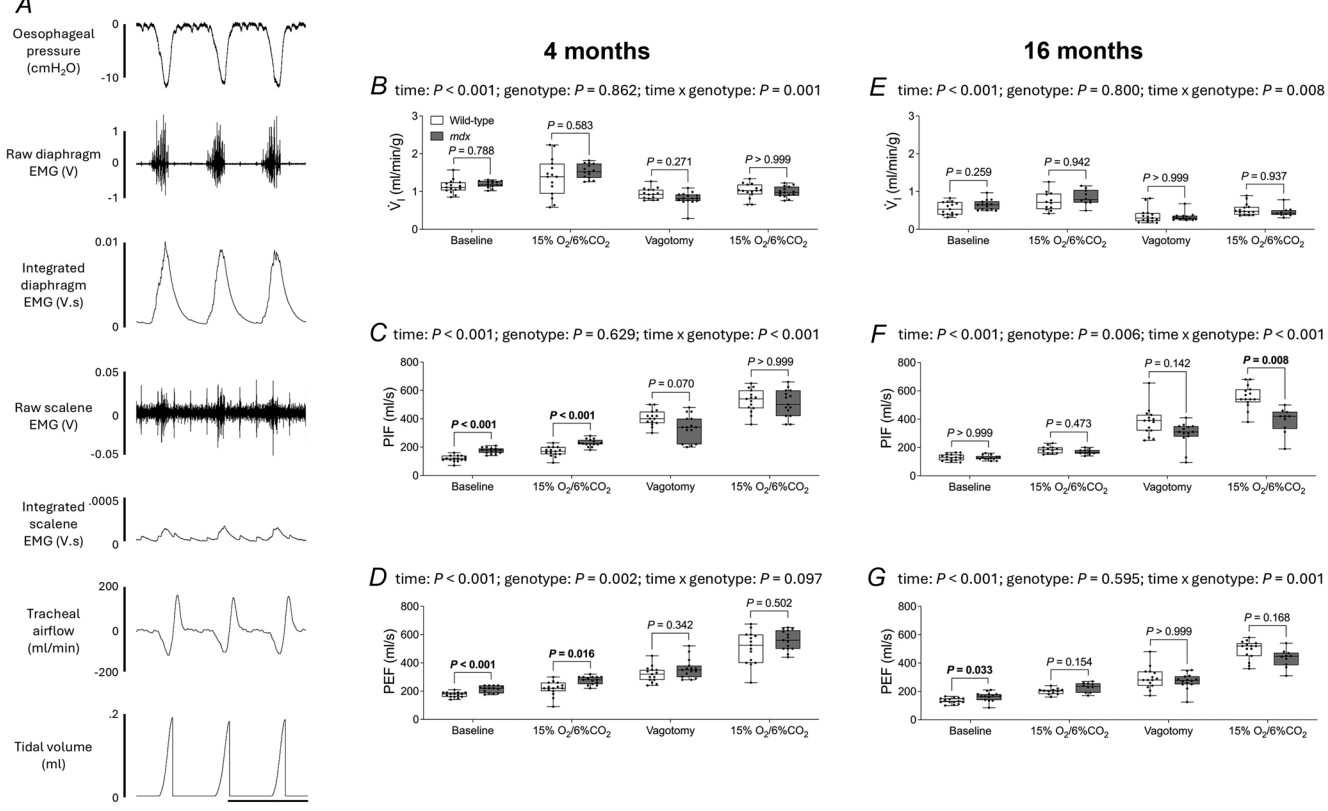

**Figure 2. Ventilation during normoxia and in response to chemostimulation in anaesthetised 4- and 16-month-old wild-type and *mdx* mice**

*A*, representative traces of oesophageal (inspiratory) pressure, raw and integrated diaphragm (obligatory) and scalene (accessory) muscle EMG activities, tracheal airflow and tidal volume for an anaesthetised 4-month-old wild-type mouse during baseline conditions (60% inspired $O_2$; vagi intact). *B–G*, group data for minute ventilation (*B* and *E*), peak inspiratory flow (*C* and *F*) and peak expiratory flow (*D* and *G*) during baseline (60% inspired $O_2$; vagi intact), hypoxic-hypercapnic challenge (15% $O_2$/6% $CO_2$; vagi intact), baseline following vagotomy (60% inspired $O_2$) and hypoxic-hypercapnic challenge (15% $O_2$/6% $CO_2$; vagotomised) in 4- (*B–D*) and 16- (*E–G*) month-old wild-type (*n* = 15 per group; open) and *mdx* (*n* = 14–15 per group; grey) mice. Values are expressed as box and whisker plots (median, 25–75th centiles, minimum to maximum values and scatter plot). Data were statistically compared by two-way mixed ANOVA (time **x** genotype) with Bonferroni's multiple comparison *post hoc* test. Data sets that were not normally distributed (4 months: $V_I$ and PEF; 16 months: $V_I$ and PIF) were $\log_{10}$ transformed prior to statistical analysis. Adjusted *P* values for all comparisons are reported.

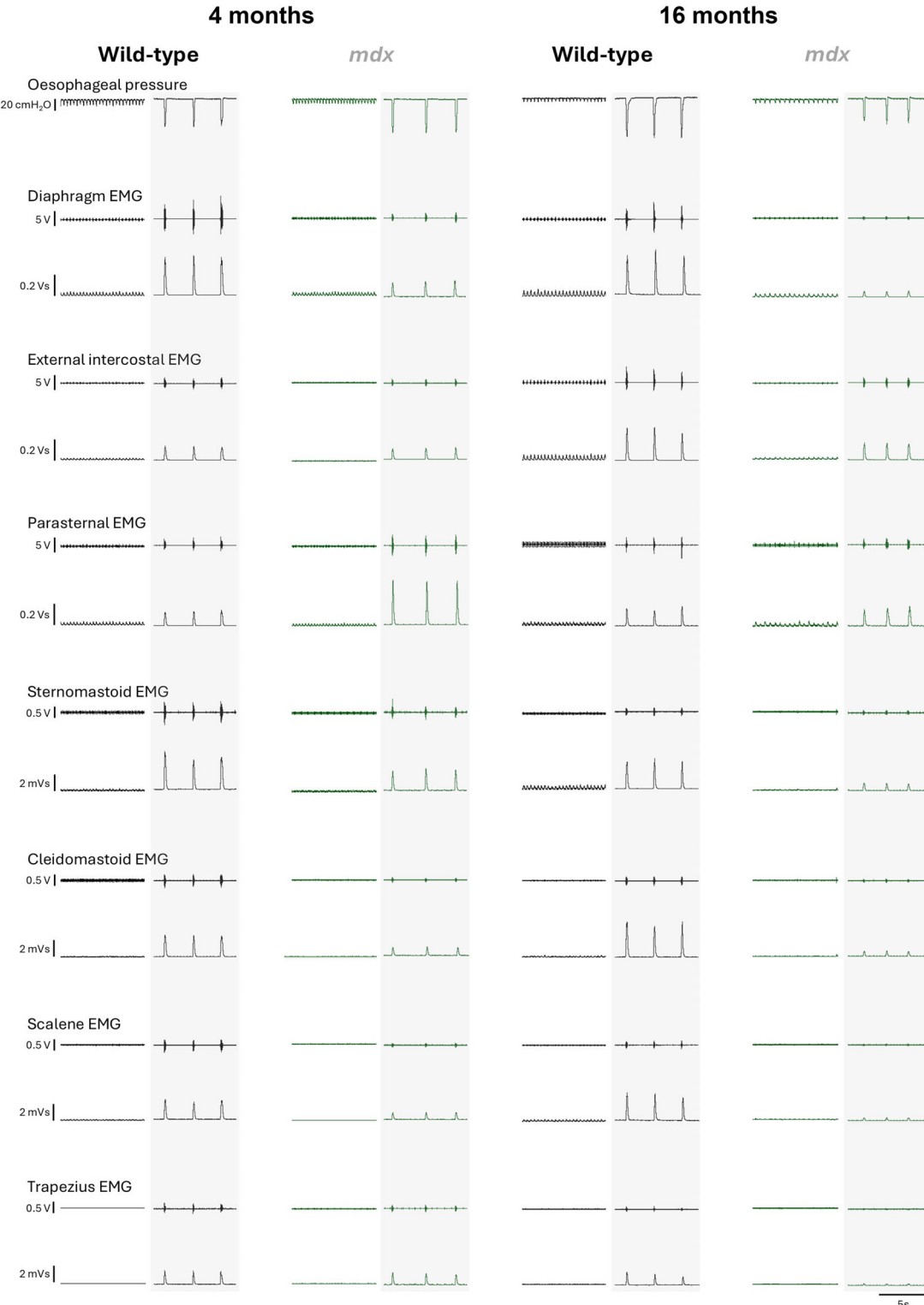

**Figure 3. Inspiratory pressure and obligatory and accessory respiratory muscle EMG activities during baseline and sustained tracheal occlusion in anaesthetised 4- and 16-month-old wild-type and *mdx* mice**

Representative traces of oesophageal (inspiratory) pressure and obligatory (diaphragm, external intercostal and parasternal) and accessory (sternomastoid, cleidomastoid, scalene and trapezius) respiratory muscle raw and integrated electromyogram (EMG) activities in 4- and 16-month-old wild-type (black) and *mdx* (grey) mice during baseline (60% inspired $O_2$; vagi intact) and during sustained tracheal occlusion (vagotomised; shaded background) evoking maximal EMG activities and pressure generation.

**4 months**

**16 months**

**Figure 4. Inspiratory pressure and obligatory and accessory respiratory muscle EMG activities in anaesthetised 4- and 16-month-old wild-type and *mdx* mice**

*A–H*, group data for oesophageal (inspiratory) pressure (*A* and *E*) and obligatory [diaphragm (*B* and *F*), external intercostal (*C* and *G*) and parasternal (*D* and *H*)] respiratory muscle EMG activities in 4- (*A–D*) and 16- (*E–H*) month-old wild-type (*n* = 14–15; open) and *mdx* (*n* = 13–15; grey) mice during baseline (60% inspired $O_2$; vagi intact), hypoxic-hypercapnic challenge (15% $O_2$/6% $CO_2$; vagi intact), baseline following vagotomy (60% inspired $O_2$), hypoxic-hypercapnic challenge (15% $O_2$/6% $CO_2$; vagotomised) and during sustained tracheal occlusion (vagotomised). *I–P*, group data for accessory [sternomastoid (*I* and *M*), cleidomastoid (*J* and *N*), scalene (*K* and *O*) and trapezius (*L* and *P*)] respiratory muscle EMG activities in 4- (*I–L*) and 16- (*M–P*) month-old wild-type (*n* = 14–15; open) and *mdx* (*n* = 13–15; grey) mice during baseline (60% inspired $O_2$; vagi intact), hypoxic-hypercapnic challenge (15% $O_2$/6% $CO_2$; vagi intact), baseline following vagotomy (60% inspired $O_2$), hypoxic-hypercapnic challenge (15% $O_2$/6% $CO_2$; vagotomised) and during sustained tracheal occlusion (vagotomised). Values are expressed as box and whisker plots (median, 25–75th centiles, minimum to maximum values and scatter plot). Data were statistically compared by two-way mixed ANOVA (time **x** genotype) with Bonferroni's multiple comparison *post hoc* test. Data sets that were not normally distributed (4 months: inspiratory pressure, diaphragm EMG, external intercostal EMG, parasternal EMG, cleidomastoid EMG, scalene EMG and trapezius EMG; 16 months: inspiratory pressure, diaphragm EMG, external intercostal EMG, parasternal EMG, sternomastoid EMG, cleidomastoid EMG, scalene EMG and trapezius EMG) were $log_{10}$ transformed prior to statistical analysis. Adjusted *P* values for all comparisons are reported.

lower in 4-month-old *mdx* mice compared to wild-type (genotype: $P = 0.021$; two-way mixed ANOVA) (Fig. 4*I*). Cleidomastoid EMG activity was significantly lower in 4-month-old *mdx* mice compared to wild-type (genotype: $P = 0.025$; two-way mixed ANOVA) (Fig. 4*J*). Scalene EMG activity was significantly lower in 4-month-old *mdx* mice compared to wild-type (genotype: $P < 0.001$; two-way mixed ANOVA) (Fig. 4*K*) during hypoxic-hypercapnic challenge ($P = 0.004$; Bonferroni's multiple comparison *post hoc*), vagotomy ($P < 0.001$) and hypoxic-hypercapnic challenge following vagotomy ($P = 0.004$). External intercostal, parasternal and trapezius EMG activities were equivalent between 4-month-old wild-type and *mdx* mice (Fig. 4*C, D* and *L*).

Diaphragm EMG activity was significantly lower in 16-month-old *mdx* mice compared to wild-type (genotype: $P < 0.001$; two-way mixed ANOVA) (Fig. 4*F*) during hypoxic-hypercapnic challenge ($P = 0.032$; Bonferroni's multiple comparison *post hoc*), vagotomy ($P = 0.033$), hypoxic-hypercapnic challenge following vagotomy ($P < 0.001$) and airway occlusion ($P < 0.001$). External intercostal EMG activity was significantly lower in 16-month-old *mdx* mice compared to wild-type (genotype: $P = 0.016$; two-way mixed ANOVA) (Fig. 4*G*) during airway occlusion ($P < 0.001$; Bonferroni's multiple comparison *post hoc*). Sternomastoid EMG activity was significantly lower in 16-month-old *mdx* mice compared to wild-type (time × genotype: $P = 0.010$; two-way

## 4 months

*I*  time: $P < 0.001$; genotype: $P = 0.021$; time x genotype: $P = 0.004$

**Sternomastoid**

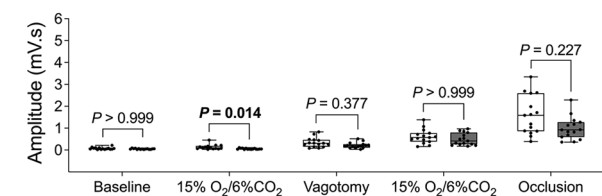

## 16 months

*M*  time: $P < 0.001$; genotype: $P = 0.067$; time x genotype: $P = 0.010$

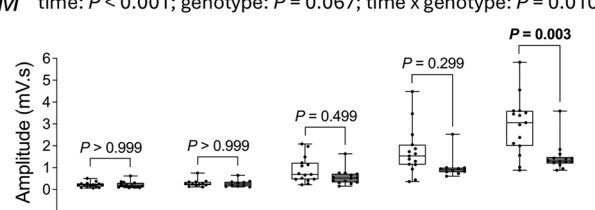

*J*  time: $P < 0.001$; genotype: $P = 0.025$; time x genotype: $P = 0.172$

**Cleidomastoid**

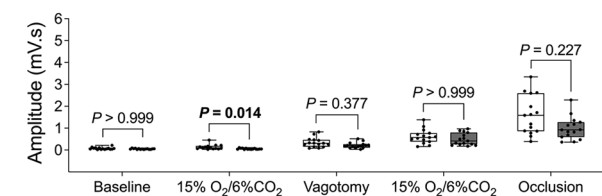

*N*  time: $P < 0.001$; genotype: $P = 0.289$; time x genotype: $P = 0.055$

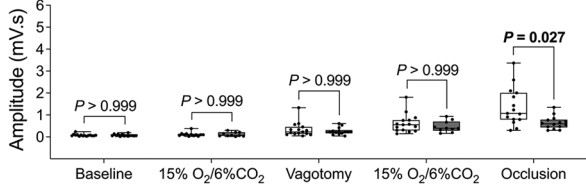

*K*  time: $P < 0.001$; genotype: $P < 0.001$; time x genotype: $P = 0.038$

**Scalene**

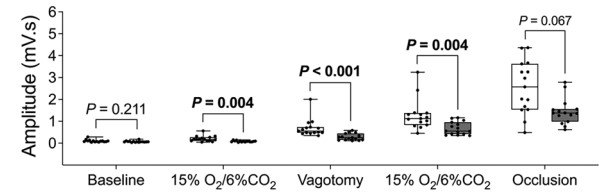

*O*  time: $P < 0.001$; genotype: $P < 0.001$; time x genotype: $P < 0.001$

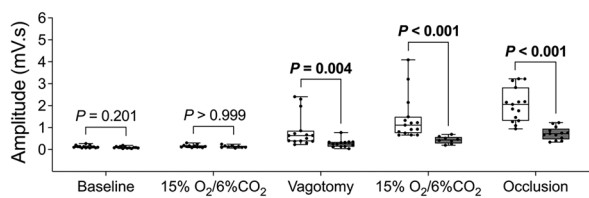

*L*  time: $P < 0.001$; genotype: $P = 0.320$; time x genotype: $P = 0.063$

**Trapezius**

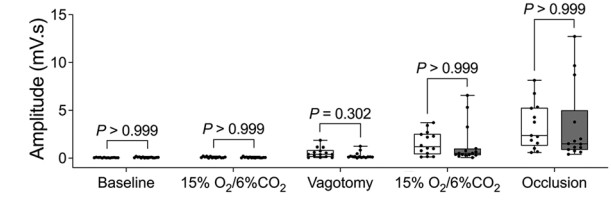

*P*  time: $P < 0.001$; genotype: $P = 0.031$; time x genotype: $P = 0.076$

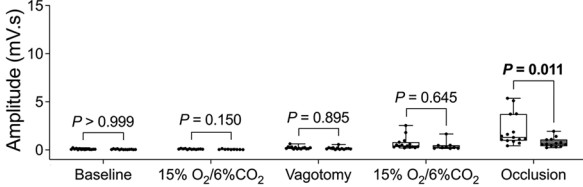

**Figure 4.  Continued**

mixed ANOVA) (Fig. 4*M*) during airway occlusion (*P* = 0.003; Bonferroni's multiple comparison *post hoc*). Cleidomastoid EMG activity was significantly lower in 16-month-old *mdx* mice compared to wild-type during airway occlusion (*P* = 0.027; Bonferroni's multiple comparison *post hoc*) (Fig. 4*N*). Scalene EMG activity was significantly lower in 16-month-old *mdx* mice compared to wild-type (genotype: *P* < 0.001; two-way mixed ANOVA) (Fig. 4*O*) during the baseline period following vagotomy (*P* = 0.004; Bonferroni's multiple comparison *post hoc*), hypoxic-hypercapnic challenge following vagotomy (*P* < 0.001) and airway occlusion (*P* < 0.001). Trapezius EMG activity was significantly lower in 16-month-old *mdx* mice compared to wild-type (genotype: *P* = 0.031; two-way mixed ANOVA) (Fig. 4*P*) during airway occlusion (*P* = 0.011; Bonferroni's multiple comparison *post hoc*). Parasternal EMG activity was equivalent between 16-month-old wild-type and *mdx* mice (Fig. 4*H*).

## Obligatory and accessory respiratory muscle contractile function *ex vivo* in 4- and 16-month-old mice

Figures 5 and 6 show representative original traces of obligatory (diaphragm) (Figs 5*A*, *B* and 6*A*, *B*) and accessory (scalene) (Figs 5*E*, *F* and 6*G*, *H*) respiratory muscle twitch (Fig. 5*A* and *E*) and tetanic (Fig. 5*B* and *F*) contractions, force–frequency relationship (Fig. 6*A* and *G*) and maximal unloaded shortening (Fig. 6*B* and *H*) in 4-month-old wild-type (black) and *mdx* (grey) mice. Four-month-old wild-type and *mdx* twitch kinetics (TTP and $\frac{1}{2}$RT), isotonic contractile parameters ($S_{max}$ and $V_{max}$), work and power are shown in Table 3. Figure 7 shows representative original traces of obligatory (diaphragm) (Fig. 7*A*–*D*) and accessory (scalene) (Fig. 7*H*–*K*) respiratory muscle twitch (Fig. 7*A* and *H*) and tetanic (Fig. 7*B* and *I*) contractions, force–frequency relationship (Fig. 7*C* and *J*) and maximal unloaded shortening (Fig. 7*D* and *K*) in 16-month-old wild-type (black) and *mdx* (grey) mice. Sixteen-month-old wild-type and *mdx* twitch kinetics (TTP and $\frac{1}{2}$RT), isotonic contractile parameters ($S_{max}$ and $V_{max}$), work and power are shown in Table 4.

Twitch and tetanic forces were significantly reduced in 4-month-old *mdx* obligatory (diaphragm and parasternal) and accessory (sternomastoid and cleidomastoid) muscles compared to wild-type (Fig. 5). Diaphragm muscle twitch (*P* < 0.001; Mann–Whitney test) and tetanic (*P* < 0.001; unpaired Student's *t* test) forces were significantly reduced in 4-month-old *mdx* compared to wild-type (Fig. 5*C*). Parasternal muscle twitch (*P* = 0.014; unpaired Student's *t* test) and tetanic (*P* = 0.002; unpaired Student's *t* test with Welch's correction) forces were significantly reduced

in 4-month-old *mdx* compared to wild-type (Fig. 5*D*). Sternomastoid muscle twitch (*P* = 0.004; Mann–Whitney test) and tetanic (*P* = 0.002; Mann–Whitney test) forces were significantly reduced in 4-month-old *mdx* compared to wild-type (Fig. 5*G*). Cleidomastoid muscle twitch (*P* = 0.017; unpaired Student's *t* test with Welch's correction) and tetanic (*P* = 0.018; unpaired Student's *t* test with Welch's correction) forces were significantly reduced in 4-month-old *mdx* compared to wild-type (Fig. 5*H*). Scalene muscle twitch and tetanic forces were equivalent between groups at 4 months of age (Fig. 5*I*).

Twitch and tetanic forces were significantly reduced in 16-month-old *mdx* obligatory (diaphragm) and accessory (scalene) muscles compared to wild-type (Fig. 7). Diaphragm muscle twitch (*P* < 0.001; unpaired Student's *t* test with Welch's correction) and tetanic (*P* < 0.001; Mann–Whitney test) forces were significantly reduced in 16-month-old *mdx* compared to wild-type (Fig. 7*E*). Scalene muscle twitch (*P* < 0.001; unpaired Student's *t* test) and tetanic (*P* = 0.004; unpaired Student's *t* test) forces were significantly reduced in 16-month-old *mdx* compared to wild-type (Fig. 7*L*).

For the force–frequency relationship, obligatory (diaphragm and parasternal) (Fig. 6*C* and *E*) and accessory (sternomastoid and cleidomastoid) (Fig. 6*I* and *K*) respiratory muscle forces were significantly decreased in 4-month-old *mdx* compared to wild-type mice. Forces were significantly decreased in 4-month-old *mdx* diaphragm, parasternal, sternomastoid and cleidomastoid muscles compared to wild-type mice (genotype: *P* < 0.001, *P* = 0.007, *P* = 0.002, and *P* = 0.043 respectively; two-way mixed ANOVA) (Fig. 6*C*, *E*, *I* and *K*). Scalene muscle force–frequency relationship was equivalent between groups at 4 months of age (Fig. 6*M*).

Diaphragm and scalene muscle forces were significantly decreased in 16-month-old *mdx* compared to wild-type across a range of stimulation frequencies (Fig. 7*F* and *M*). Forces were significantly decreased in 16-month-old *mdx* diaphragm and scalene muscles compared to wild-type mice (genotype: *P* < 0.001 and *P* = 0.033, respectively; two-way mixed ANOVA) (Fig. 7*F* and *M*).

Overall, obligatory (diaphragm and parasternal) and accessory (sternomastoid and cleidomastoid) respiratory muscle weakness is evident at 4 months of age in *mdx* mice, except for the scalene muscle. Scalene muscle function is preserved in early dystrophic disease, but weakness is evident at 16 months of age.

Absolute shortening was significantly lower in *mdx* diaphragm muscle compared to wild-type at 4 months of age (*P* = 0.005; unpaired Student's *t* test) (Table 3). When normalised to optimum length, diaphragm muscle shortening was not significantly different in *mdx* compared to wild-type at 4 months of age (*P* = 0.053;

**Table 3. Four-month-old obligatory and accessory respiratory muscle contractile properties**

| | Diaphragm | | |
|---|---|---|---|
| | Wild-type ($n$ = 11–12) | $mdx$ ($n$ = 13–15) | P value |
| TTP (ms) | 17.1 ± 2.3 | 18.0 ± 1.8 | 0.239 |
| $\frac{1}{2}$RT (ms) | 18.3 ± 3.7 | 14.2 ± 1.8 | **0.003** |
| $S_{max}$ (cm) | 0.43 ± 0.11 | 0.31 ± 0.08 | **0.005** |
| $S_{max}$ ($L/L_O$) | 0.52 ± 0.10 | 0.42 ± 0.11 | 0.053 |
| $V_{max}$ (cm s$^{-1}$) | 5.9 ± 1.4 | 5.4 ± 1.7 | 0.407 |
| Work (N/cm$^2 \cdot L/L_O$) | 1.08 ± 0.80 | 0.36 ± 0.28 | **<0.001** |
| Power (N/cm$^2 \cdot L_O$/s) | 6.9 ± 2.7 | 4.5 ± 3.0 | 0.051 |
| $L_O$ (cm) | 0.82 ± 0.10 | 0.74 ± 0.08 | **0.019** |
| | Parasternal | | |
| | Wild-type ($n$ = 11) | $mdx$ ($n$ = 10–11) | P value |
| TTP (ms) | 12.1 ± 0.7 | 12.4 ± 1.6 | 0.548 |
| $\frac{1}{2}$RT (ms) | 9.8 ± 2.4 | 11.4 ± 4.0 | 0.391 |
| $S_{max}$ (cm) | 0.28 ± 0.07 | 0.25 ± 0.08 | 0.388 |
| $S_{max}$ ($L/L_O$) | 0.34 ± 0.06 | 0.30 ± 0.07 | 0.174 |
| $V_{max}$ (cm s$^{-1}$) | 5.1 ± 1.5 | 4.9 ± 1.8 | 0.846 |
| $L_O$ (cm) | 0.80 ± 0.16 | 0.82 ± 0.15 | 0.815 |
| | Sternomastoid | | |
| | Wild-type ($n$ = 10–11) | $mdx$ ($n$ = 11–15) | P value |
| TTP (ms) | 10.8 ± 0.7 | 10.0 ± 0.9 | **0.020** |
| $\frac{1}{2}$RT (ms) | 7.6 ± 1.1 | 10.7 ± 3.2 | **0.003** |
| $S_{max}$ (cm) | 0.10 ± 0.03 | 0.06 ± 0.03 | **0.012** |
| $S_{max}$ ($L/L_O$) | 0.18 ± 0.06 | 0.13 ± 0.08 | 0.137 |
| $V_{max}$ (cm s$^{-1}$) | 1.51 ± 0.62 | 0.97 ± 0.57 | **0.044** |
| $L_O$ (cm) | 0.56 ± 0.17 | 0.46 ± 0.09 | 0.219 |
| | Cleidomastoid | | |
| | Wild-type ($n$ = 9–11) | $mdx$ ($n$ = 8–11) | P value |
| TTP (ms) | 9.6 ± 1.3 | 9.3 ± 1.1 | 0.627 |
| $\frac{1}{2}$RT (ms) | 7.6 ± 0.9 | 8.2 ± 0.8 | 0.068 |
| $S_{max}$ (cm) | 0.12 ± 0.07 | 0.09 ± 0.04 | 0.333 |
| $S_{max}$ ($L/L_O$) | 0.20 ± 0.11 | 0.17 ± 0.09 | 0.321 |
| $V_{max}$ (cm s$^{-1}$) | 2.3 ± 1.5 | 1.7 ± 0.7 | 0.333 |
| $L_O$ (cm) | 0.59 ± 0.19 | 0.52 ± 0.13 | 0.296 |
| | Scalene | | |
| | Wild-type ($n$ = 9) | $mdx$ ($n$ = 8–9) | P value |
| TTP (ms) | 9.6 ± 1.1 | 13.4 ± 2.7 | **0.003** |
| $\frac{1}{2}$RT (ms) | 16.5 ± 3.8 | 17.6 ± 5.6 | 0.621 |
| $S_{max}$ (cm) | 0.11 ± 0.03 | 0.14 ± 0.06 | 0.155 |
| $S_{max}$ ($L/L_O$) | 0.23 ± 0.14 | 0.25 ± 0.11 | 0.236 |
| $V_{max}$ (cm s$^{-1}$) | 1.4 ± 0.7 | 1.5 ± 0.8 | 0.873 |
| $L_O$ (cm) | 0.53 ± 0.18 | 0.59 ± 0.13 | 0.461 |

Group data (mean ± SD) for twitch contraction time to peak (TTP), twitch half-relaxation time ($\frac{1}{2}$RT), peak shortening ($S_{max}$), peak shortening velocity ($V_{max}$), mechanical work, mechanical power and optimum length ($L_O$) of obligatory (diaphragm and parasternal) and accessory (sternomastoid, cleidomastoid and scalene) respiratory muscles from 4-month-old wild-type and $mdx$ mice. Data were statistically compared using unpaired Student's $t$ tests, with Welch's correction used where appropriate. Mann–Whitney non-parametric tests were used to compare data that were not normally distributed.

Mann–Whitney test) (Table 3). Absolute shortening was significantly lower in *mdx* sternomastoid muscle compared to wild-type at 4 months of age ($P = 0.012$; unpaired Student's *t* test) (Table 3). However, when normalised to optimum length, muscle shortening was equivalent between wild-type and *mdx* mice (Table 3). Both absolute and normalised muscle shortening were significantly lower in *mdx* diaphragm muscle compared

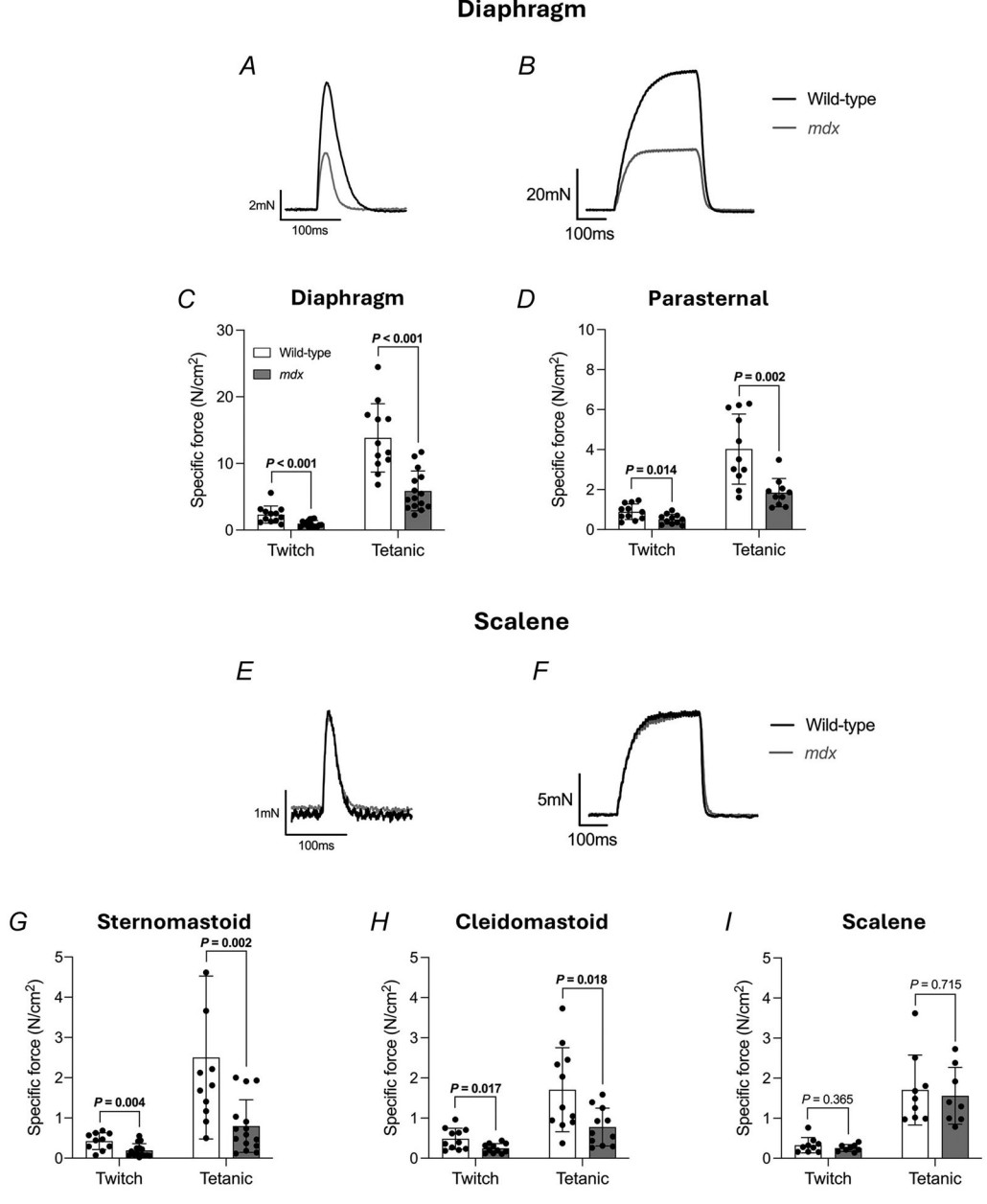

**Figure 5. *Ex vivo* 4-month-old wild-type and *mdx* obligatory and accessory respiratory muscle twitch and tetanic force**
*A*, *B* and *E*, *F*, original traces of *ex vivo* diaphragm (*A* and *B*) and scalene (*E* and *F*) muscle twitch contraction (*A* and *E*) and tetanic contraction (*B* and *F*) for 4-month-old wild-type (black) and *mdx* (grey) preparations. Twitch and tetanic contractions in each respective group are from different animals. *C*, *D* and *G*–*I*, group data for obligatory [diaphragm (*C*) and parasternal (*D*)] and accessory [sternomastoid (*G*), cleidomastoid (*H*) and scalene (*I*)] respiratory muscle twitch and tetanic force in 4-month-old wild-type (open; *n* = 9–12) and *mdx* (grey; *n* = 8–15) preparations. Tetanic force was measured following stimulation at 100 Hz *ex vivo*. Data are shown as mean ± SD and scatter plot. Data were statistically compared using unpaired Student's *t* tests, with Welch's correction used where appropriate. Mann–Whitney non-parametric tests were used to compare data that were not normally distributed. Absolute *P* values for all comparisons are reported.

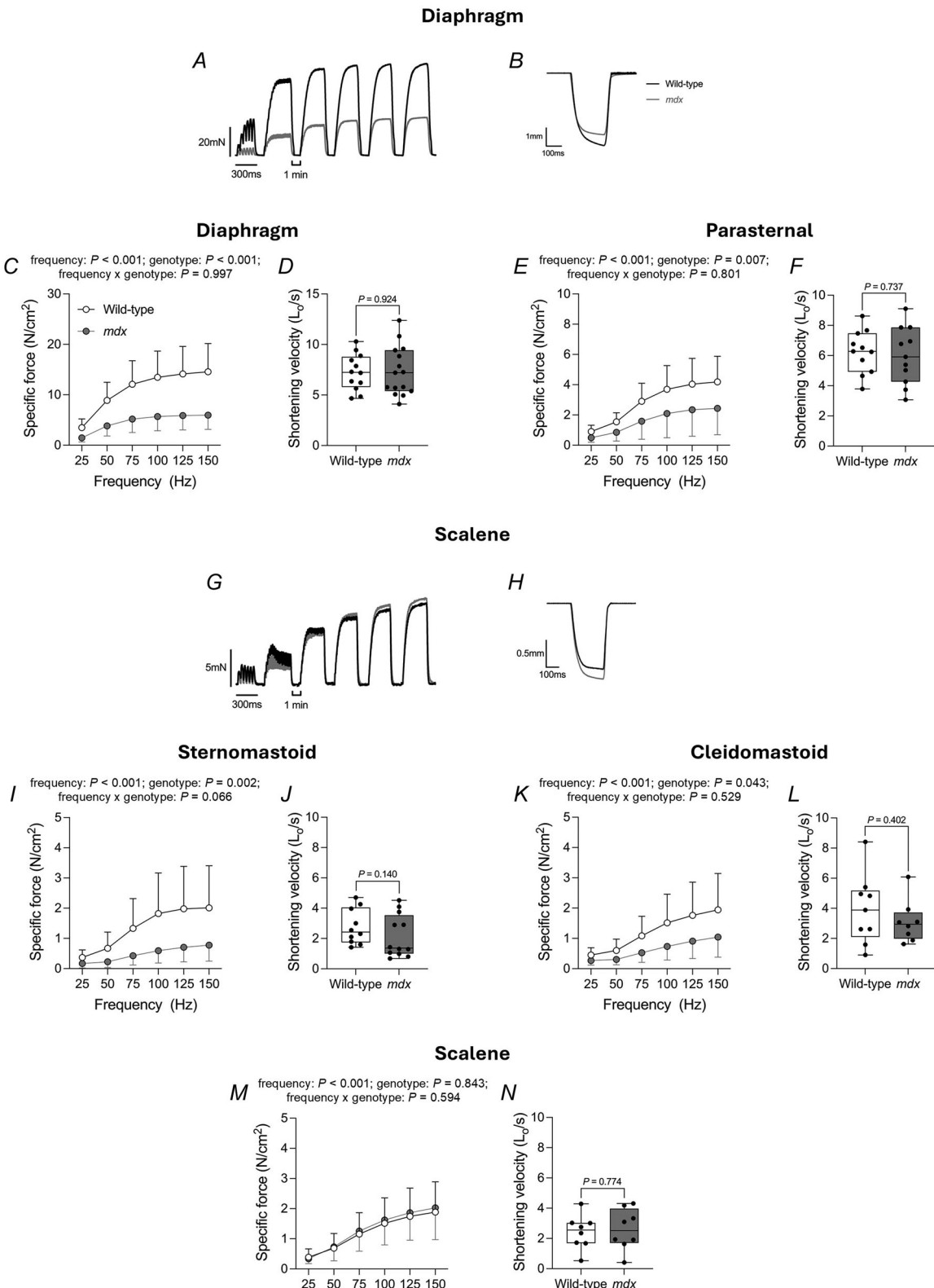

**Figure 6. *Ex vivo* 4-month-old wild-type and *mdx* obligatory and accessory respiratory muscle force–frequency relationship and shortening velocity**

*A*, *B* and *G*, *H*, original traces of *ex vivo* diaphragm (*A* and *B*) and scalene (*G* and *H*) muscle force–frequency relationship (*A* and *G*) and maximum unloaded shortening (*B* and *H*) for 4-month-old wild-type (black) and *mdx* (grey) preparations. Sequential contractions in force–frequency relationships are from the same muscle preparation

in each case. *C, E, I, K* and *M*, group data (mean ± SD) for obligatory [diaphragm (*C*) and parasternal (*E*)] and accessory [sternomastoid (*I*), cleidomastoid (*K*) and scalene (*M*)] respiratory muscle force-frequency relationship *ex vivo* in 4-month-old wild-type (open; *n* = 9–12) and *mdx* (grey; *n* = 8–15) preparations. Data were not normally distributed and underwent $\log_{10}$ transformation prior to statistical comparison by two-way mixed ANOVA (frequency × genotype) followed by Bonferroni *post hoc* test. Diaphragm muscle *post hoc* adjusted *P* values are 0.001 (25 Hz), <0.001 (50 Hz), <0.001 (75 Hz), <0.001 (100 Hz), <0.001 (125 Hz) and <0.001 (150 Hz). Parasternal muscle *post hoc* adjusted *P* values are 0.054 (25 Hz), 0.031 (50 Hz), 0.039 (75 Hz), 0.061 (100 Hz), 0.065 (125 Hz) and 0.054 (150 Hz). Sternomastoid muscle *post hoc* adjusted *P* values are 0.033 (25 Hz), 0.018 (50 Hz), 0.009 (75 Hz), 0.008 (100 Hz), 0.013 (125 Hz) and 0.021 (150 Hz). Cleidomastoid muscle *post hoc* adjusted *P* values are 0.379 (25 Hz), 0.238 (50 Hz), 0.198 (75 Hz), 0.227 (100 Hz), 0.295 (125 Hz) and 0.363 (150 Hz). Scalene muscle *post hoc* adjusted *P* values are >0.999 (25 Hz), >0.999 (50 Hz), >0.999 (75 Hz), >0.999 (100 Hz), >0.999 (125 Hz) and >0.999 (150 Hz). *D, F, J, L* and *N*, group data for diaphragm (*D*), parasternal (*F*), sternomastoid (*J*), cleidomastoid (*L*) and scalene (*N*) muscle maximum unloaded shortening velocity in 4-month-old wild-type (*n* = 8–12) and *mdx* (*n* = 8–15) preparations. Values are expressed as box and whisker plots (median, 25–75% centiles, minimum to maximum values and scatter plot). Data were statistically compared using unpaired Student's *t* tests. Mann–Whitney non-parametric tests were used to compare data that were not normally distributed. Absolute *P* values for all comparisons are reported.

**Table 4. Sixteen-month-old obligatory and accessory respiratory muscle contractile properties**

| | Diaphragm | | |
| --- | --- | --- | --- |
| | Wild-type (*n* = 11–12) | *mdx* (*n* = 14) | *P* value |
| TTP (ms) | 17.7 ± 2.3 | 18.3 ± 3.7 | 0.662 |
| $\frac{1}{2}$RT (ms) | 19.2 ± 3.4 | 21.0 ± 4.5 | 0.295 |
| $S_{max}$ (cm) | 0.42 ± 0.10 | 0.26 ± 0.07 | **<0.001** |
| $S_{max}$ ($L/L_O$) | 0.47 ± 0.09 | 0.34 ± 0.08 | **<0.001** |
| $V_{max}$ (cm s$^{-1}$) | 5.2 ± 1.7 | 2.2 ± 0.8 | **<0.001** |
| Work (N/cm$^2 \cdot L/L_O$) | 1.20 ± 0.98 | 0.26 ± 0.19 | **<0.001** |
| Power (N/cm$^2 \cdot L_O$/s) | 8.8 ± 6.6 | 1.8 ± 1.2 | **0.004** |
| $L_O$ (cm) | 0.91 ± 0.10 | 0.76 ± 0.09 | **<0.001** |
| | Scalene | | |
| | Wild-type (*n* = 15) | *mdx* (*n* = 13) | *P* value |
| TTP (ms) | 9.5 ± 1.4 | 8.0 ± 1.5 | **0.012** |
| $\frac{1}{2}$RT (ms) | 20.7 ± 5.6 | 18.9 ± 4.6 | 0.373 |
| $S_{max}$ (cm) | 0.10 ± 0.05 | 0.06 ± 0.02 | **0.005** |
| $S_{max}$ ($L/L_O$) | 0.17 ± 0.08 | 0.11 ± 0.03 | **0.014** |
| $V_{max}$ (cm s$^{-1}$) | 0.99 ± 0.46 | 0.40 ± 0.15 | **<0.001** |
| $L_O$ (cm) | 0.57 ± 0.07 | 0.49 ± 0.06 | **<0.001** |

Group data (mean ± SD) for twitch contraction time to peak (TTP), twitch half-relaxation time ($\frac{1}{2}$RT), peak shortening ($S_{max}$), peak shortening velocity ($V_{max}$), mechanical work, mechanical power and optimum length ($L_O$) of obligatory (diaphragm) and accessory (scalene) respiratory muscles from 16-month-old wild-type and *mdx* mice. Data were statistically compared using unpaired Student's *t* tests, with Welch's correction used where appropriate. Mann–Whitney non-parametric tests were used to compare data that were not normally distributed.

to wild-type at 16 months of age ($P < 0.001$ and $P < 0.001$, respectively; unpaired Student's *t* test) (Table 4). Similarly, both absolute and normalised muscle shortening were significantly lower in *mdx* scalene muscle compared to wild-type at 16 months of age ($P = 0.005$ and $P = 0.014$, respectively; unpaired Student's *t* test with Welch's correction) (Table 4).

Absolute shortening velocity was significantly lower in *mdx* sternomastoid muscle compared to wild-type at 4 months of age ($P = 0.044$; unpaired Student's *t* test) (Table 3). However, when normalised to optimum length, shortening velocity was equivalent between wild-type and *mdx* mice (Fig. 6*J*). Normalised shortening velocity was equivalent between groups for both obligatory (diaphragm and parasternal; Fig. 6*D* and *F*) and accessory (sternomastoid, cleidomastoid and scalene) (Fig. 6*J, L*, and *N*) respiratory muscles at 4 months of age. Both absolute and normalised shortening velocity were significantly decreased in 16-month-old *mdx* diaphragm ($P < 0.001$ for both; unpaired Student's *t* test with Welch's

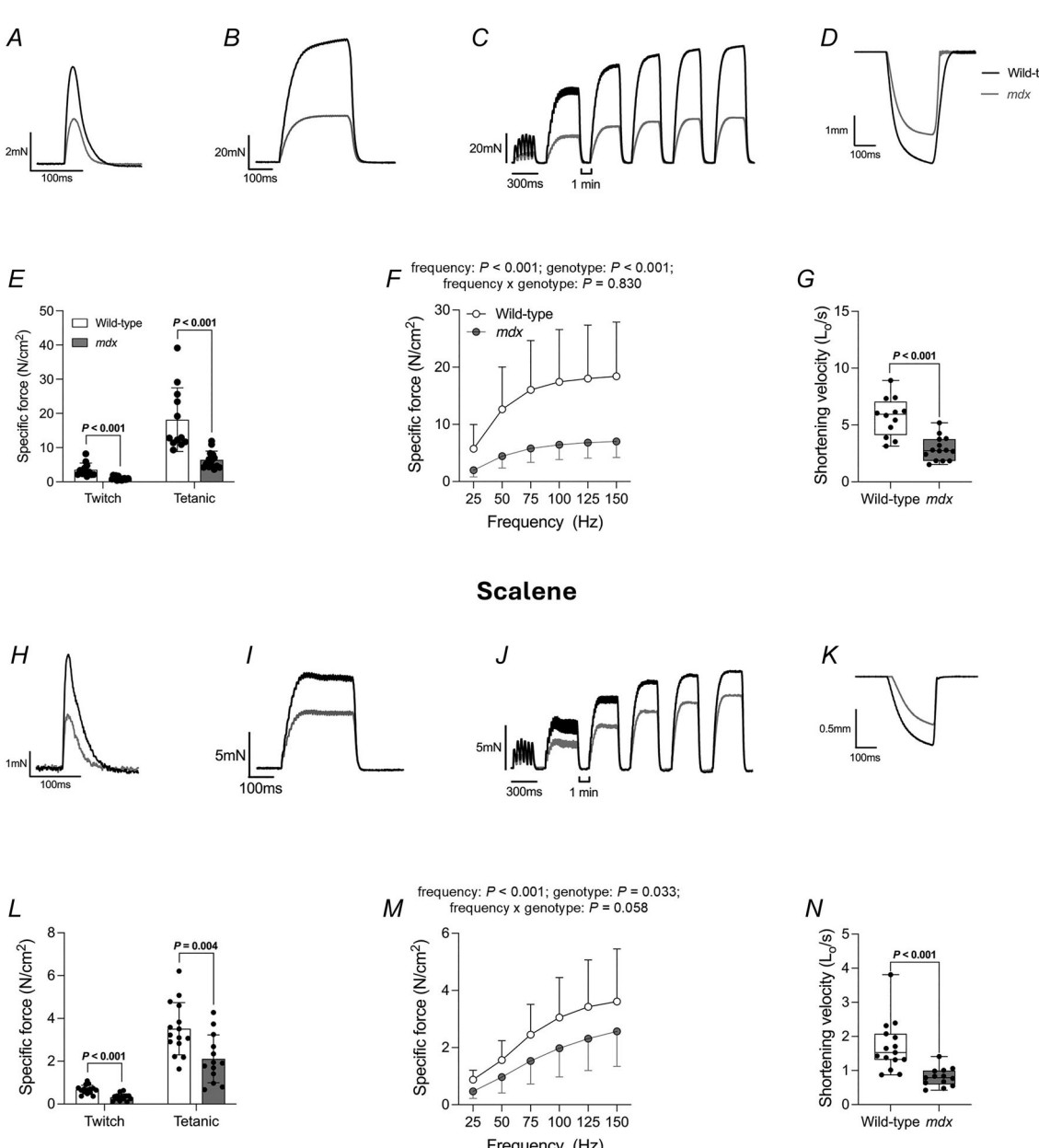

**Figure 7. *Ex vivo* 16-month-old wild-type and *mdx* obligatory and accessory respiratory muscle contractile function**

*A–D* and *H–K*, original traces of *ex vivo* obligatory [diaphragm (*A–D*)] and accessory [scalene (*H–K*)] respiratory muscle twitch contraction (*A* and *H*), tetanic contraction (*B* and *I*), force–frequency relationship (*C* and *J*) and maximum unloaded shortening (*D* and *K*) for 16-month-old wild-type (black) and *mdx* (grey) preparations. *E* and *L*, group data for diaphragm (*E*) and scalene (*L*) muscle twitch and tetanic force in 16-month-old wild-type (open; *n* = 12–15) and *mdx* (grey; *n* = 13–14) preparations. Tetanic force was measured following stimulation at 100 Hz *ex vivo*. Twitch and tetanic contractions in each respective group are from different animals. Sequential contractions in force-frequency relationships are from the same muscle preparation in each case. Data are shown as mean ± SD and scatter plot. Data were statistically compared using unpaired Student's *t* tests, with Welch's correction used where appropriate. Mann–Whitney non-parametric tests were used to compare data that were not normally distributed. Absolute *P* values for all comparisons are reported. *F* and *M*, group data (mean ± SD)

for diaphragm (*F*) and scalene (*M*) muscle force–frequency relationship *ex vivo* in 16-month-old wild-type (open; *n* = 12–15) and *mdx* (grey; *n* = 13–14) preparations. Data were statistically compared by two-way mixed ANOVA (frequency × genotype) followed by Bonferroni *post hoc* test. Data sets that were not normally distributed (scalene) were $\log_{10}$ transformed prior to statistical analysis. Diaphragm muscle *post hoc* adjusted *P* values are 0.002 (25 Hz), <0.001 (50 Hz), <0.001 (75 Hz), <0.001 (100 Hz), <0.001 (125 Hz), and <0.001 (150 Hz). Scalene muscle *post hoc* adjusted *P* values are 0.006 (25 Hz), 0.098 (50 Hz), 0.088 (75 Hz), 0.155 (100 Hz), 0.269 (125 Hz) and 0.509 (150 Hz). *G* and *N*, group data for diaphragm (*G*) and scalene (*N*) muscle maximum unloaded shortening velocity in 16-month-old wild-type (*n* = 12–15) and *mdx* (*n* = 13–14) preparations. Values are expressed as box and whisker plots (median, 25%–75% centiles, minimum to maximum values and scatter plot). Data were statistically compared using unpaired Student's *t* tests, with Welch's correction used where appropriate. Absolute *P* values for all comparisons are reported.

correction and unpaired Student's *t* test, respectively) and scalene (*P* < 0.001 and *P* < 0.001, respectively; unpaired Student's *t* test with Welch's correction) muscles compared to wild-type (Fig. 7*G* and *N* and Table 4).

Work was significantly lower in *mdx* diaphragm muscle compared to wild-type at both 4 and 16 months of age (*P* < 0.001 and *P* < 0.001, respectively; Mann–Whitney test) (Tables 3 and 4). Muscle power was not significantly different in *mdx* diaphragm muscle compared to wild-type at 4 months of age (*P* = 0.051; unpaired Student's *t* test) (Table 3). Power was significantly lower in *mdx* diaphragm muscle compared to wild-type at 16 months of age (*P* = 0.004; unpaired Student's *t* test with Welch's correction) (Table 4).

Overall, muscle dysfunction is more severe in advanced dystrophic disease in both diaphragm and scalene muscles, revealing progressive impairments to muscle function.

## Obligatory and accessory respiratory muscle structure in 4- and 16-month-old mice

Figures 8 and 9 show representative images of obligatory (diaphragm, intercostal and parasternal) (Fig. 8) and accessory (sternomastoid, cleidomastoid, scalene and trapezius) (Fig. 9) respiratory muscle histology and immunofluorescence for 4- and 16-month-old wild-type and *mdx* mice.

The proportion of muscle fibres with centrally located myonuclei was significantly increased in *mdx* obligatory [diaphragm (*P* < 0.001; unpaired Student's *t* test with Welch's correction), intercostal (*P* < 0.001; unpaired Student's *t* test with Welch's correction) and parasternal (*P* < 0.001; unpaired Student's *t* test)] and accessory [sternomastoid (*P* = 0.002; Mann–Whitney test), cleidomastoid (*P* < 0.001; unpaired Student's *t* test), scalene (*P* < 0.001; unpaired Student's *t* test with Welch's correction) and trapezius (*P* < 0.001; unpaired Student's *t* test with Welch's correction)] respiratory muscles compared to wild-type at 4 months of age (Fig. 10*A*). Similarly, the proportion of muscle fibres with centrally located myonuclei was significantly increased in *mdx* obligatory [diaphragm (*P* = 0.004; unpaired Student's

*t* test with Welch's correction), intercostal (*P* < 0.001; unpaired Student's *t* test with Welch's correction) and parasternal (*P* = 0.002; Mann–Whitney test)] and accessory [sternomastoid (*P* < 0.001; unpaired Student's *t* test with Welch's correction), cleidomastoid (*P* = 0.002; unpaired Student's *t* test with Welch's correction), scalene (*P* < 0.001; unpaired Student's *t* test with Welch's correction) and trapezius (*P* < 0.001; unpaired Student's *t* test with Welch's correction)] respiratory muscles compared to wild-type at 16 months of age (Fig. 10*D*). There was a reduction in the proportion of muscle fibres with centrally located myonuclei in 16-month-old *mdx* muscles compared to 4-month-old *mdx* muscles (diaphragm: *P* = 0.003; unpaired Student's *t* test, intercostal: *P* < 0.001; unpaired Student's *t* test, parasternal: *P* < 0.001; unpaired Student's *t* test, sternomastoid: *P* < 0.001; unpaired Student's *t* test, cleidomastoid: *P* = 0.014; unpaired Student's *t* test with Welch's correction, scalene: *P* < 0.001; unpaired Student's *t* test, trapezius: *P* = 0.002; unpaired Student's *t* test).

The relative area of inflammatory cell infiltration was significantly increased in *mdx* obligatory [diaphragm (*P* < 0.001; unpaired Student's *t* test) and parasternal (*P* = 0.021; unpaired Student's *t* test)] and accessory [cleidomastoid (*P* = 0.017; unpaired Student's *t* test), scalene (*P* = 0.009; unpaired Student's *t* test with Welch's correction) and trapezius (*P* = 0.009; Mann–Whitney test)] respiratory muscles compared to wild-type at 4 months of age (Fig. 10*B*). There was no significant difference in the relative area of inflammatory cell infiltration in *mdx* intercostal and sternomastoid muscles compared to wild-type at 4 months of age (Fig. 10*B*). The relative area of inflammatory cell infiltration was significantly increased in *mdx* obligatory [diaphragm (*P* < 0.001; unpaired Student's *t* test), intercostal (*P* = 0.002; Mann–Whitney test) and parasternal (*P* = 0.002; unpaired Student's *t* test)] and accessory [sternomastoid (*P* = 0.003; unpaired Student's *t* test), cleidomastoid (*P* = 0.004; Mann–Whitney test), scalene (*P* < 0.001; unpaired Student's *t* test) and trapezius (*P* = 0.003; unpaired Student's *t* test with Welch's correction)] respiratory muscles compared to wild-type at 16 months of age (Fig. 10*E*). The greatest increase in the relative area of inflammatory cell infiltration from 4

to 16 months of age was in the diaphragm muscle of *mdx* mice, with a significantly greater area in 16-month-old *mdx* compared to 4-month-old *mdx* mice ($P < 0.001$; unpaired Student's *t* test).

The relative area of collagen labelled with Sirius Red staining was increased in *mdx* obligatory [diaphragm ($P = 0.001$; unpaired Student's *t* test with Welch's correction), intercostal ($P < 0.001$; unpaired Student's *t* test) and parasternal ($P < 0.001$; unpaired Student's *t* test)] and accessory [sternomastoid ($P < 0.001$; unpaired Student's *t* test), cleidomastoid ($P = 0.010$; unpaired Student's *t* test with Welch's correction), scalene ($P < 0.001$; unpaired Student's *t* test with Welch's

correction) and trapezius ($P = 0.035$; unpaired Student's *t* test)] respiratory muscles compared to wild-type at 4 months of age (Fig. 10C). The relative area of collagen labelled with Sirius Red staining was increased in *mdx* obligatory [diaphragm ($P < 0.001$; unpaired Student's *t* test with Welch's correction) and intercostal ($P = 0.002$; Mann–Whitney test)] and accessory [sternomastoid ($P = 0.005$; unpaired Student's *t* test), cleidomastoid ($P = 0.005$; unpaired Student's *t* test with Welch's correction), scalene ($P = 0.002$; unpaired Student's *t* test with Welch's correction) and trapezius ($P = 0.009$; unpaired Student's *t* test)] respiratory muscles compared to wild-type at 16 months of age (Fig. 10F).

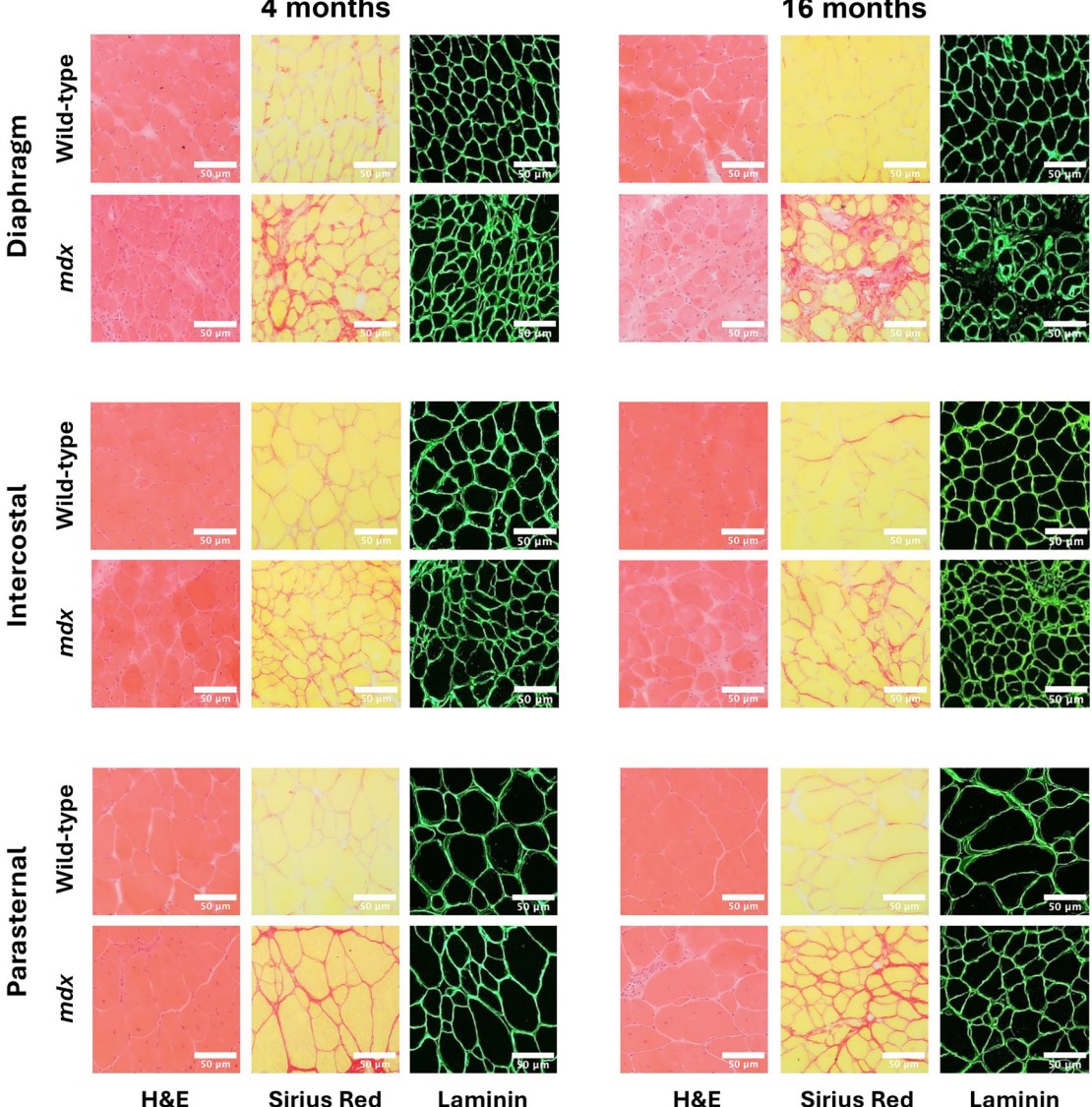

**4 months** — **16 months**

H&E — Sirius Red — Laminin — H&E — Sirius Red — Laminin

**Figure 8. Four- and 16-month-old obligatory respiratory muscle histology and immunofluorescence**
Representative images of transverse sections of diaphragm, intercostal, and parasternal muscles histologically stained with haematoxylin and eosin (H&E) and Sirius Red and immunofluorescence labelled for laminin from 4- (left) and 16- (right) month-old wild-type and *mdx* mice (*n* = 6 per group). Scale bars = 50 μm.

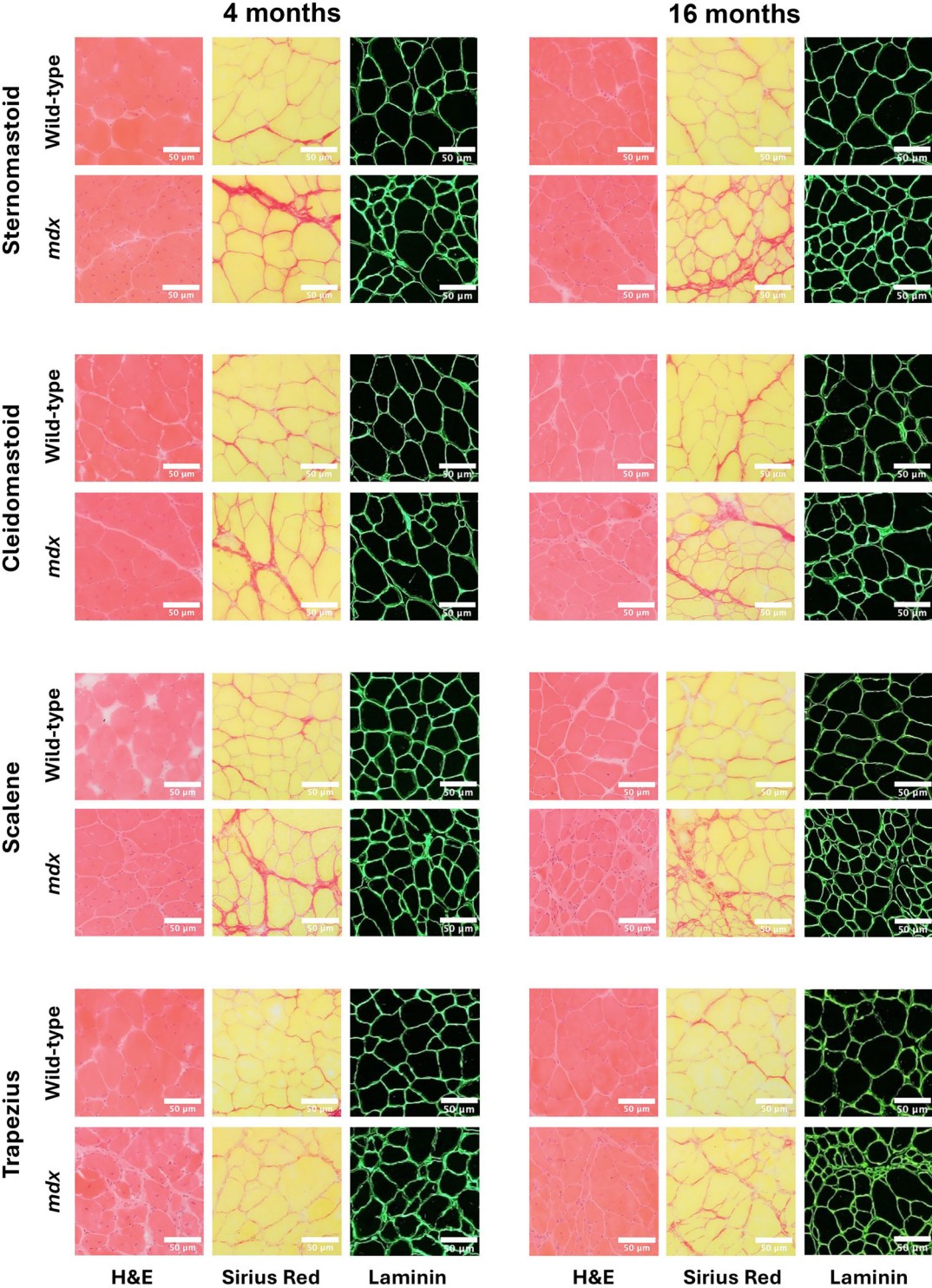

**Figure 9. Four- and 16- month-old accessory respiratory muscle histology and immunofluorescence**
Representative images of transverse sections of sternomastoid, cleidomastoid, scalene, and trapezius muscles histologically stained with haematoxylin and eosin (H&E) and Sirius Red and immunofluorescence labelled for laminin from 4- (left) and 16- (right) month-old wild-type and *mdx* mice ($n$ = 6 per group). Scale bars = 50 μm.

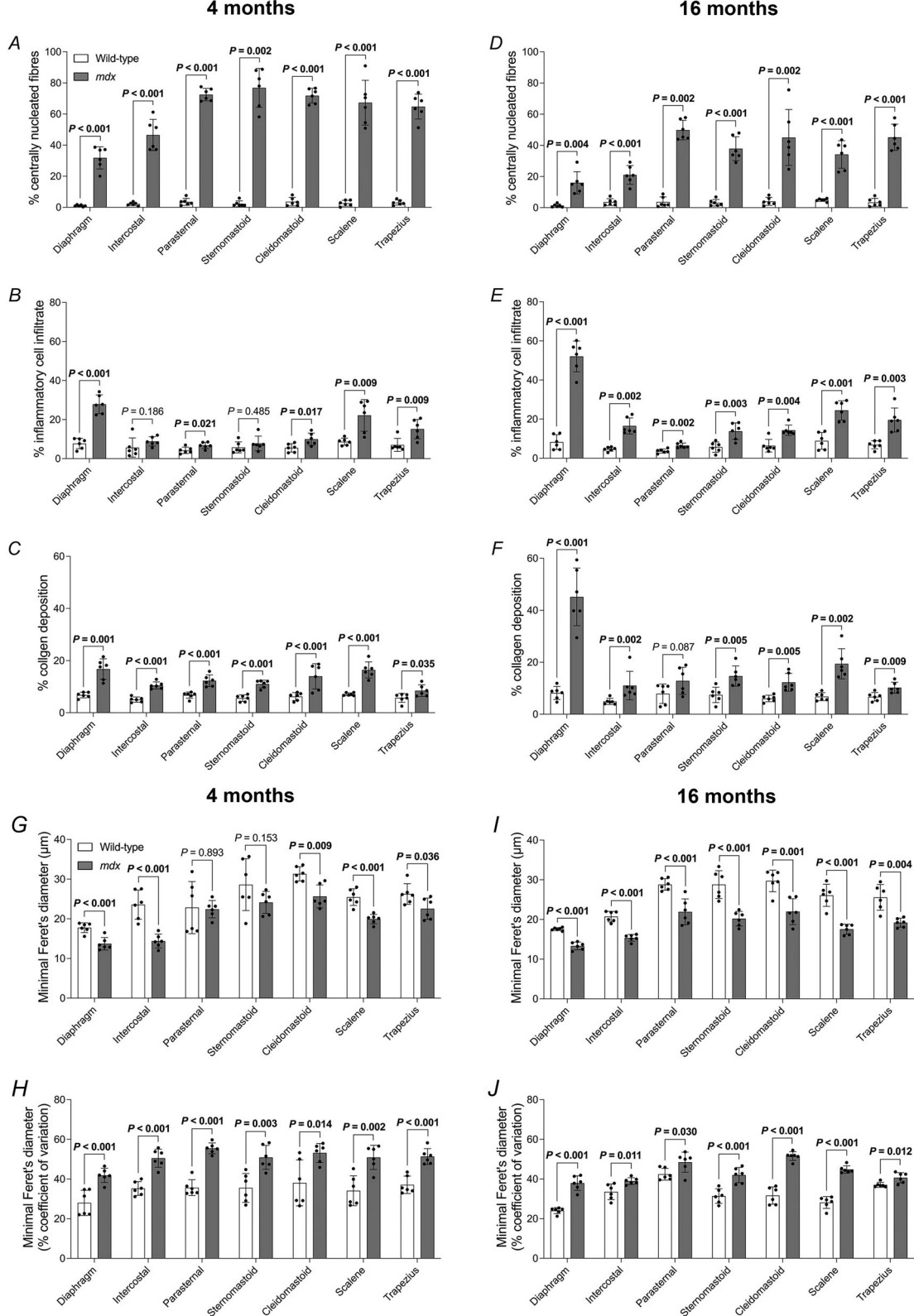

**Figure 10. Four- and 16- month-old obligatory and accessory respiratory muscle structure**
*A* and *D*, group data showing the percentage of central nucleation in diaphragm, intercostal, parasternal, sternomastoid, cleidomastoid, scalene and trapezius muscles from 4- (*A*) and 16- (*D*) month-old wild-type (open) and *mdx* (grey) mice (*n* = 6 per group). *B* and *E*, group data showing the percentage of infiltration of inflammatory cells in diaphragm, intercostal, parasternal, sternomastoid, cleidomastoid, scalene and trapezius muscles from 4-

(*B*) and 16- (*E*) month-old wild-type (open) and *mdx* (grey) mice (*n* = 6 per group). *C* and *F*, group data showing the percentage of collagen deposition in diaphragm, intercostal, parasternal, sternomastoid, cleidomastoid, scalene and trapezius muscles from 4- (*C*) and 16- (*F*) month-old wild-type (open) and *mdx* (grey) mice (*n* = 6 per group). *G* and *I*, group data showing muscle fibre size, as measured by minimal Feret's diameter in diaphragm, intercostal, parasternal, sternomastoid, cleidomastoid, scalene and trapezius muscles from 4- (*G*) and 16- (*I*) month-old wild-type (open) and *mdx* (grey) mice (*n* = 6 per group). *H* and *J*, group data showing the coefficient of variation of muscle fibre size in diaphragm, intercostal, parasternal, sternomastoid, cleidomastoid, scalene and trapezius muscles from 4- (*H*) and 16- (*J*) month-old wild-type (open) and *mdx* (grey) mice (*n* = 6 per group). Data are expressed as mean ± SD and scatter plot. Data were statistically compared using unpaired Student's *t* tests, with Welch's correction used where appropriate. Mann–Whitney non-parametric tests were used to compare data that was not normally distributed. Absolute *P* values for all comparisons are reported.

The greatest increase in the relative area of collagen labelled with Sirius Red staining from 4 to 16 months of age was in the diaphragm muscle of *mdx* mice, with a significantly greater area in 16-month-old *mdx* compared to 4-month-old *mdx* mice ($P < 0.001$; unpaired Student's *t* test with Welch's correction).

There was a significant reduction in muscle fibre size (minimal Feret's diameter) for *mdx* obligatory [diaphragm ($P < 0.001$; unpaired Student's *t* test) and intercostal ($P < 0.001$; unpaired Student's *t* test)] and accessory [cleidomastoid ($P = 0.009$; Mann–Whitney test), scalene ($P < 0.001$; unpaired Student's *t* test) and trapezius ($P = 0.036$; unpaired Student's *t* test)] respiratory muscles compared to wild-type at 4 months of age (Fig. 10*G*). There was no significant difference in myofibre size in *mdx* parasternal and sternomastoid muscles compared to wild-type at 4 months of age (Fig. 10*G*). There was a significant reduction in muscle fibre size (minimal Feret's diameter) for *mdx* obligatory [diaphragm ($P < 0.001$; unpaired Student's *t* test), intercostal ($P < 0.001$; unpaired Student's *t* test) and parasternal ($P < 0.001$; unpaired Student's *t* test)] and accessory [sternomastoid ($P < 0.001$; unpaired Student's *t* test), cleidomastoid ($P = 0.001$; unpaired Student's *t* test), scalene ($P < 0.001$; unpaired Student's *t* test) and trapezius ($P = 0.004$; unpaired Student's *t* test with Welch's correction)] respiratory muscles compared to wild-type at 16 months of age (Fig. 10*I*). There is a leftward shift in the frequency distribution of myofibre size in *mdx* obligatory and accessory respiratory muscles at 4 and 16 months of age, based on minimal Feret's diameter (Fig. 11).

There was a significant increase in the coefficient of variation of muscle fibre size (minimal Feret's diameter) for *mdx* obligatory [diaphragm ($P < 0.001$; unpaired Student's *t* test), intercostal ($P < 0.001$; unpaired Student's *t* test) and parasternal ($P < 0.001$; unpaired Student's *t* test)] and accessory [sternomastoid ($P = 0.003$; unpaired Student's *t* test), cleidomastoid ($P = 0.014$; unpaired Student's *t* test), scalene ($P = 0.002$; unpaired Student's *t* test) and trapezius ($P < 0.001$; unpaired Student's *t* test)] respiratory muscles compared to wild-type at 4 months of age (Fig. 10*H*). There was a significant increase in the coefficient of variation of muscle fibre size (minimal Feret's diameter) for *mdx* obligatory [diaphragm ($P < 0.001$; unpaired Student's *t* test), intercostal ($P = 0.011$; unpaired Student's *t* test) and parasternal ($P = 0.030$; unpaired Student's *t* test)] and accessory [sternomastoid ($P < 0.001$; unpaired Student's *t* test), cleidomastoid ($P < 0.001$; unpaired Student's *t* test), scalene ($P < 0.001$; unpaired Student's *t* test) and trapezius ($P = 0.012$; unpaired Student's *t* test)] respiratory muscles compared to wild-type at 16 months of age (Fig. 10*J*).

## Obligatory and accessory respiratory muscle cytokine concentrations in 4- and 16-month-old mice

Figure 12 shows heat maps summarising the fold-change of cytokines relative to the 4-month-old wild-type group for 4- and 16-month-old wild-type and *mdx* obligatory (diaphragm, intercostal and parasternal) (Fig. 12*A–C*) and accessory (sternomastoid, scalene and trapezius) (Fig. 12*D–F*) respiratory muscles.

There was a significantly increased concentration of IL-1$\beta$ (Table 5) ($P < 0.001$; unpaired Student's *t* test), IL-6 (Table 5) ($P < 0.001$), KC/GRO (Table 5) ($P < 0.001$), TNF-$\alpha$ (Table 5) ($P < 0.001$), MIP-1$\alpha$ (Table 5) ($P < 0.001$), MIP-2 (Table 5) ($P < 0.001$), MCP-1 (Table 5) ($P = 0.002$) and IP-10 (Table 5) ($P < 0.001$) in *mdx* diaphragm muscle compared to wild-type at 4 months of age. There was a significantly decreased concentration of IL-9 (Table 5) ($P = 0.001$) in *mdx* diaphragm muscle compared to wild-type at 4 months of age. The concentrations of IL-2, IL-5, IL-10, IL-12p70, IL-15, IL17A/F, IL-27p28/IL-30 and IL-33 were equivalent between wild-type and *mdx* diaphragm muscle at 4 months of age.

Similarly, there was a significantly increased concentration of IL-1$\beta$ (Table 5) ($P < 0.001$), IL-2 (Table 5) ($P < 0.001$), IL-6 (Table 5) ($P < 0.001$), IL-12p70 (Table 5) ($P = 0.002$), IL-33 (Table 5) ($P < 0.001$), KC/GRO (Table 5) ($P < 0.001$), TNF-$\alpha$ (Table 5) ($P < 0.001$), MIP-1$\alpha$ (Table 5) ($P < 0.001$), MIP-2 (Table 5) ($P < 0.001$), MCP-1 (Table 5) ($P < 0.001$) and IP-10 (Table 5) ($P < 0.001$) in *mdx* diaphragm muscle compared to wild-type at 16 months of age. There was a significantly decreased concentration of IL-27p28/IL-30 (Table 5) ($P < 0.001$) in *mdx* diaphragm

muscle compared to wild-type at 16 months of age. The concentrations of IL-5, IL-9, IL-10, IL-15 and IL-17A/F were equivalent between wild-type and *mdx* diaphragm muscle at 16 months of age.

Similar to the diaphragm muscle, there was a significantly increased concentration of IL-1$\beta$, IL-6, IL-33, KC/GRO, TNF-$\alpha$, MIP-1$\alpha$, MIP-2, MCP-1 and IP-10 and a significantly decreased concentration of IL-27p28/IL-30 in *mdx* intercostal, parasternal,

sternomastoid, scalene and trapezius muscles compared to wild-type at 4 months of age, with some exceptions (Tables 6–10). Likewise, there was a significantly increased concentration of IL-1$\beta$, IL-6, IL-33, KC/GRO, TNF-$\alpha$, MIP-1$\alpha$, MIP-2, MCP-1 and IP-10 in *mdx* intercostal, parasternal, sternomastoid, scalene and trapezius muscles compared to wild-type at 16 months of age, with some exceptions (Tables 6–10).

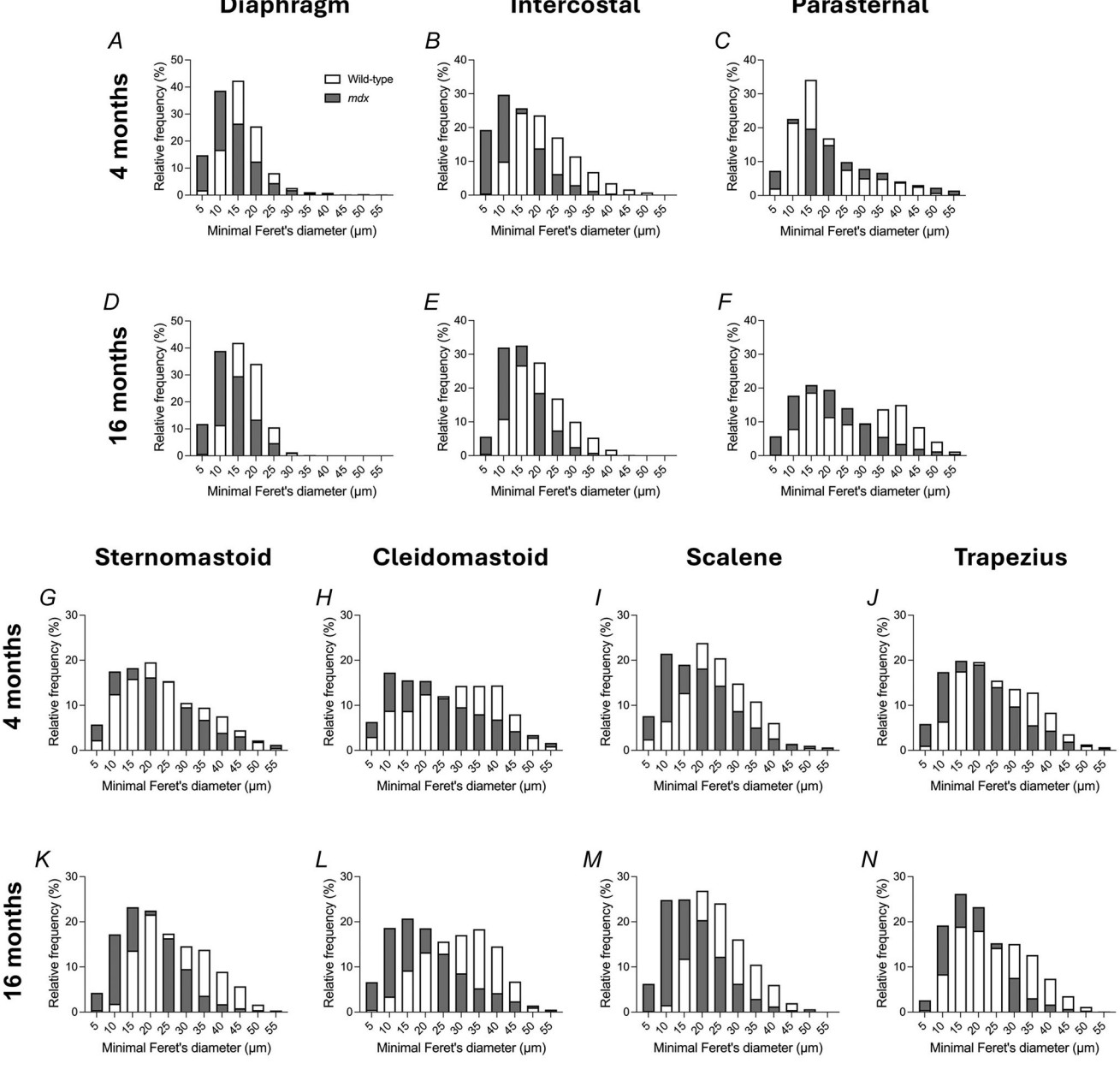

**Figure 11. Four- and 16- month-old obligatory and accessory respiratory muscle fibre size distribution**
Frequency distribution curves showing obligatory (*A–F*) and accessory (*G–N*) respiratory muscle fibre size, as measured by minimal Feret's diameter for 4- (*A–C* and *G–J*) and 16- (*D–F* and *K–N*) month-old wild-type (open) and *mdx* (grey) diaphragm (*A* and *D*), intercostal (*B* and *E*), parasternal (*C* and *F*), sternomastoid (*G* and *K*), cleidomastoid (*H* and *L*), scalene (*I* and *M*) and trapezius (*J* and *N*) muscles (*n* = 6 per group).

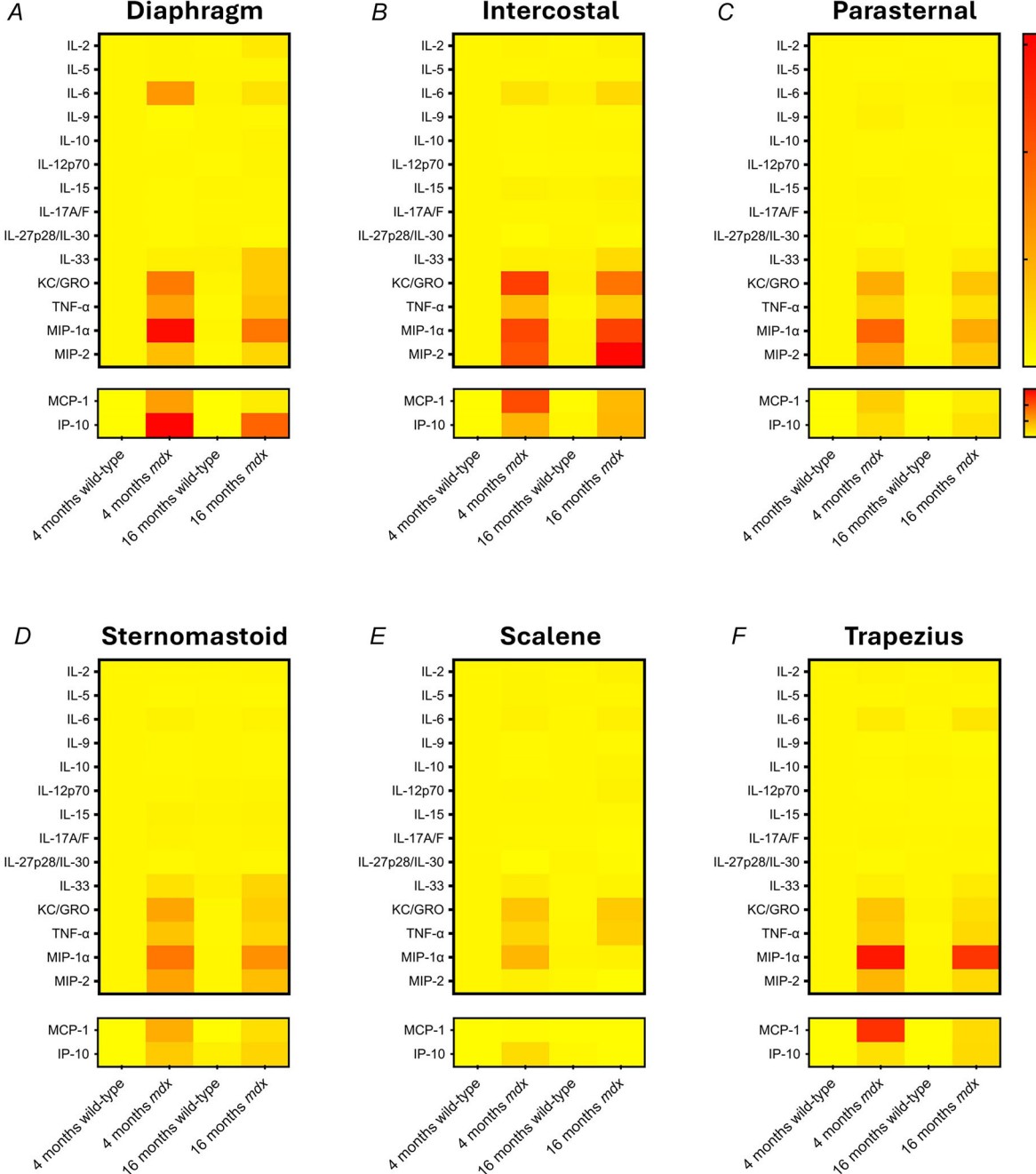

**Figure 12. Four- and 16- month-old obligatory and accessory respiratory muscle cytokine concentration heat maps**

Heat maps illustrating fold-change in cytokine concentration in 4- and 16- month-old wild-type and *mdx* obligatory [diaphragm (*A*), intercostal (*B*) and parasternal (*C*)] and accessory [sternomastoid (*D*), scalene (*E*) and trapezius (*F*)] respiratory muscles relative to the 4-month-old wild-type group ($n = 7$–10 per group). Red represents an increase in concentration. Cytokines include interleukin-2 (IL-2), interleukin-5 (IL-5), interleukin-6 (IL-6), interleukin-9 (IL-9), interleukin-10 (IL-10), interleukin-12p70 (IL-12p70), interleukin-15 (IL-15), interleukin-17A/F (IL-17A/F), interleukin-27p28/interleukin-30 (IL-27p28/IL-30), interleukin-33 (IL-33), keratinocyte chemoattractant/human growth-regulated oncogene (KC/GRO), tumour necrosis factor-$\alpha$ (TNF-$\alpha$), macrophage inflammatory protein-1$\alpha$ (MIP-1$\alpha$), macrophage inflammatory protein-2 (MIP-2), monocyte chemoattractant protein-1 (MCP-1) and interferon gamma-induced protein 10 (IP-10).

**Table 5. Four- and 16-month-old diaphragm muscle cytokine concentrations**

| (pg mg⁻¹ protein homogenate) | 4 months diaphragm | | | 16 months diaphragm | | |
|---|---|---|---|---|---|---|
| | Wild-type (*n* = 9 - 10) | *mdx* (*n* = 8–10) | *P* value | Wild-type (*n* = 8–10) | *mdx* (*n* = 9–10) | *P* value |
| IL-1$\beta$ | 0 ± 0 | 0.65 ± 0.51 | **<0.001** | 0 ± 0 | 0.057 ± 0.048 | **<0.001** |
| IL-2 | 0.047 ± 0.015 | 0.059 ± 0.013 | 0.069 | 0.046 ± 0.012 | 0.120 ± 0.056 | **<0.001** |
| IL-5 | 0.036 ± 0.007 | 0.043 ± 0.008 | 0.034 | 0.040 ± 0.007 | 0.041 ± 0.004 | 0.688 |
| IL-6 | 0.47 ± 0.08 | 5.85 ± 5.24 | **<0.001** | 0.61 ± 0.24 | 1.59 ± 0.71 | **<0.001** |
| IL-9 | 1.67 ± 0.38 | 1.02 ± 0.34 | **0.001** | 1.73 ± 0.38 | 1.53 ± 0.36 | 0.247 |
| IL-10 | 0.11 ± 0.03 | 0.10 ± 0.02 | 0.200 | 0.12 ± 0.03 | 0.16 ± 0.05 | 0.064 |
| IL-12p70 | 3.19 ± 0.94 | 3.48 ± 0.89 | 0.501 | 2.59 ± 1.09 | 3.96 ± 0.85 | **0.002** |
| IL-15 | 10.61 ± 1.36 | 8.96 ± 0.64 | 0.007 | 12.21 ± 2.38 | 11.54 ± 1.86 | 0.492 |
| IL-17A/F | 0.19 ± 0.07 | 0.14 ± 0.02 | 0.044 | 0.19 ± 0.06 | 0.20 ± 0.07 | 0.789 |
| IL-27p28/IL-30 | 0.68 ± 0.10 | 0.55 ± 0.09 | 0.005 | 0.81 ± 0.09 | 0.50 ± 0.11 | **<0.001** |
| IL-33 | 23.96 ± 5.55 | 42.94 ± 22.61 | 0.028 | 45.85 ± 25.43 | 151.02 ± 26.27 | **<0.001** |
| KC/GRO | 0.22 ± 0.10 | 3.52 ± 0.88 | **<0.001** | 0.25 ± 0.09 | 1.42 ± 0.43 | **<0.001** |
| TNF-$\alpha$ | 0.057 ± 0.017 | 0.654 ± 0.282 | **<0.001** | 0.048 ± 0.025 | 0.423 ± 0.223 | **<0.001** |
| MIP-1$\alpha$ | 0.16 ± 0.04 | 4.62 ± 0.48 | **<0.001** | 0.18 ± 0.05 | 2.59 ± 1.15 | **<0.001** |
| MIP-2 | 0.19 ± 0.07 | 1.50 ± 0.46 | **<0.001** | 0.14 ± 0.07 | 0.92 ± 0.51 | **<0.001** |
| MCP-1 | 1.80 ± 0.50 | 61.62 ± 35.74 | **0.002** | 1.14 ± 0.12 | 11.13 ± 8.73 | **<0.001** |
| IP-10 | 0.10 ± 0.16 | 9.25 ± 3.40 | **<0.001** | 0.051 ± 0.115 | 5.688 ± 2.455 | **<0.001** |

Group data (mean ± SD) for diaphragm muscle cytokine concentrations for 4- and 16-month-old wild-type and *mdx* mice. Cytokines include interleukin-1$\beta$ (IL-1$\beta$), interleukin-2 (IL-2), interleukin-5 (IL-5), interleukin-6 (IL-6), interleukin-9 (IL-9), interleukin-10 (IL-10), interleukin-12p70 (IL-12p70), interleukin-15 (IL-15), interleukin-17A/F (IL-17A/F), interleukin-27p28/interleukin-30 (IL-27p28/IL-30), interleukin-33 (IL-33), keratinocyte chemoattractant/human growth-regulated oncogene (KC/GRO), tumour necrosis factor-$\alpha$ (TNF-$\alpha$), macrophage inflammatory protein-1$\alpha$ (MIP-1$\alpha$), macrophage inflammatory protein-2 (MIP-2), monocyte chemoattractant protein-1 (MCP-1) and interferon gamma-induced protein 10 (IP-10). Data were statistically compared using unpaired Student's *t* tests, with Welch's correction used where appropriate. Mann–Whitney non-parametric tests were used to compare data that were not normally distributed. *P* < 0.003 was considered statistically significant, following Bonferroni correction for multiple comparisons.

## Discussion

The main findings of the present study are: (1) ventilatory capacity is preserved during normoxia and chemostimulation with hypoxic-hypercapnia in conscious mice in early (4 months) and advanced (16 months) dystrophic disease; (2) minute ventilation is equivalent during normoxia and chemostimulation with hypoxic-hypercapnia in anaesthetised wild-type and *mdx* mice in early and advanced dystrophic disease; (3) peak inspiratory pressure-generating capacity is preserved in early dystrophic disease but is significantly lower than wild-type in advanced dystrophic disease; (4) peak diaphragm muscle EMG activity is lower in *mdx* compared to wild-type mice, evident from early stages of dystrophic disease, while peak obligatory (external intercostal and parasternal) and accessory (sternomastoid, cleidomastoid, scalene and trapezius) EMG activities are equivalent between groups in early dystrophic disease; (5) there is a decline in obligatory and accessory respiratory muscle EMG activities in advanced dystrophic disease; (6) there is profound muscle weakness evident in obligatory (diaphragm and parasternal) and accessory (sternomastoid and cleidomastoid) respiratory muscles in early dystrophic disease; (7) the functional capacity of the scalene (accessory) muscle is preserved in early dystrophic disease, but scalene muscle weakness and dysfunction is evident in advanced dystrophic disease; (8) diaphragm muscle dysfunction is exacerbated in advanced dystrophic disease; (9) there is profound structural remodelling in obligatory (diaphragm, intercostal and parasternal) and accessory (sternomastoid, cleidomastoid, scalene and trapezius) respiratory muscles in early and advanced dystrophic disease; (10) the greatest increase in the relative area of inflammatory cell infiltration and collagen deposition (fibrosis) in respiratory muscles between 4 and 16 months of age is in the diaphragm of *mdx* mice; and (11) there are increased concentrations of pro-inflammatory cytokines and chemokines in *mdx* obligatory (diaphragm, intercostal and parasternal) and accessory (sternomastoid, scalene and trapezius) respiratory muscles, which generally appear to peak in early disease.

Despite the profound diaphragm muscle weakness that has been established as early as 4 weeks of age in *mdx* mice (O'Halloran et al., 2023), we reasoned that ventilation would be protected over much of the course of mild to moderate disease progression because of the remarkable reserve capacity of the diaphragm muscle,

**Table 6. Four- and 16-month-old intercostal muscle cytokine concentrations**

| (pg mg$^{-1}$ protein homogenate) | 4 months intercostal | | | 16 months intercostal | | |
|---|---|---|---|---|---|---|
| | Wild-type (n = 9–10) | *mdx* (n = 8–10) | P value | Wild-type (n = 8–10) | *mdx* (n = 8–10) | P value |
| IL-1β | 0.0073 ± 0.0031 | 0.220 ± 0.125 | **<0.001** | 0.017 ± 0.016 | 0.233 ± 0.163 | **<0.001** |
| IL-2 | 0.030 ± 0.006 | 0.040 ± 0.007 | 0.003 | 0.030 ± 0.005 | 0.045 ± 0.005 | **<0.001** |
| IL-5 | 0.021 ± 0.004 | 0.023 ± 0.003 | 0.267 | 0.025 ± 0.005 | 0.025 ± 0.004 | 0.760 |
| IL-6 | 0.30 ± 0.05 | 1.00 ± 0.42 | **0.002** | 0.53 ± 0.28 | 1.37 ± 0.60 | **0.002** |
| IL-9 | 1.03 ± 0.12 | 0.91 ± 0.18 | 0.089 | 1.02 ± 0.25 | 0.95 ± 0.30 | 0.622 |
| IL-10 | 0.13 ± 0.02 | 0.13 ± 0.03 | 0.726 | 0.14 ± 0.03 | 0.16 ± 0.04 | 0.384 |
| IL-12p70 | 2.72 ± 1.45 | 2.30 ± 0.71 | 0.616 | 2.45 ± 1.29 | 2.50 ± 1.08 | 0.853 |
| IL-15 | 3.73 ± 0.39 | 5.67 ± 0.72 | **<0.001** | 4.86 ± 1.06 | 6.56 ± 1.71 | 0.015 |
| IL-17A/F | 0.047 ± 0.010 | 0.047 ± 0.007 | 0.989 | 0.047 ± 0.013 | 0.059 ± 0.026 | 0.222 |
| IL-27p28/IL-30 | 0.75 ± 0.12 | 0.41 ± 0.06 | **<0.001** | 0.83 ± 0.18 | 0.52 ± 0.12 | **<0.001** |
| IL-33 | 11.91 ± 1.82 | 21.08 ± 3.33 | **<0.001** | 19.41 ± 5.04 | 50.74 ± 17.74 | **0.002** |
| KC/GRO | 0.11 ± 0.03 | 2.53 ± 1.08 | **<0.001** | 0.22 ± 0.15 | 1.80 ± 0.88 | **<0.001** |
| TNF-α | 0.025 ± 0.009 | 0.205 ± 0.069 | **<0.001** | 0.027 ± 0.015 | 0.165 ± 0.071 | **<0.001** |
| MIP-1α | 0.061 ± 0.009 | 1.351 ± 0.352 | **<0.001** | 0.097 ± 0.056 | 1.402 ± 0.858 | **0.002** |
| MIP-2 | 0.021 ± 0.026 | 0.431 ± 0.094 | **<0.001** | 0.038 ± 0.037 | 0.624 ± 0.379 | **<0.001** |
| MCP-1 | 0.79 ± 0.10 | 50.00 ± 30.47 | **<0.001** | 1.21 ± 0.88 | 19.20 ± 7.92 | **<0.001** |
| IP-10 | 0.10 ± 0.04 | 2.59 ± 1.03 | **<0.001** | 0.29 ± 0.36 | 2.55 ± 1.23 | 0.003 |

Group data (mean ± SD) for intercostal muscle cytokine concentrations for 4- and 16-month-old wild-type and *mdx* mice. Cytokines include interleukin-1β (IL-1β), interleukin-2 (IL-2), interleukin-5 (IL-5), interleukin-6 (IL-6), interleukin-9 (IL-9), interleukin-10 (IL-10), interleukin-12p70 (IL-12p70), interleukin-15 (IL-15), interleukin-17A/F (IL-17A/F), interleukin-27p28/interleukin-30 (IL-27p28/IL-30), interleukin-33 (IL-33), keratinocyte chemoattractant/human growth-regulated oncogene (KC/GRO), tumour necrosis factor-α (TNF-α), macrophage inflammatory protein-1α (MIP-1α), macrophage inflammatory protein-2 (MIP-2), monocyte chemoattractant protein-1 (MCP-1) and interferon gamma-induced protein 10 (IP-10). Data were statistically compared using unpaired Student's *t* tests, with Welch's correction used where appropriate. Mann–Whitney non-parametric tests were used to compare data that were not normally distributed. $P < 0.003$ was considered statistically significant, following Bonferroni correction for multiple comparisons.

given that peak ventilation during chemoactivation can be achieved with <40% of maximum diaphragm muscle force (Greising et al., 2016; Mantilla et al., 2010; Sieck & Fournier, 1989). As such, ventilatory behaviours in *mdx* mice may be within the force-generating capacity of even the dystrophic diaphragm and/or facilitated by compensation from extra-diaphragmatic muscles.

Ventilation was assessed during normoxia and in response to hypoxic-hypercapnic challenge in conscious mice using whole-body plethysmography, confirming the capacity for *mdx* mice to enhance ventilation during chemoreflex challenge, by increasing both respiratory frequency and tidal volume, in agreement with our previous findings in young *mdx* mice (Burns, Drummond, et al., 2019; Burns, Murphy, et al., 2019) and extending observations up to 16 months of age (this study). The ventilatory equivalent for $CO_2$ ($\dot{V}_\text{I}/\dot{V}_{CO_2}$) was similar between groups at both ages, confirming the capacity for *mdx* to maintain basal breathing and enhance ventilation in response to chemoreflex challenge in both early and advanced dystrophic disease. $\dot{V}_\text{I}$ was preserved in *mdx* mice despite reduced $f_\text{R}$ in early disease, again further suggesting that there is no mechanical insufficiency across the ventilatory range in *mdx* mice up to 16 months of age.

It is difficult to determine whether the reduced frequency in *mdx*, which has been reported elsewhere (Delaney & O'Halloran, 2024; Mhandire et al., 2022), reflects a true ventilatory difference between genotypes or is the result of behavioural differences between groups, given that mice are conscious, highlighting a limitation of the whole-body plethysmography technique. Interestingly, a more anxious phenotype has previously been reported in *mdx* (Sekiguchi et al., 2009; Vaillend & Chaussenot, 2017), but this might ordinarily be expected to result in increased $f_\text{R}$. Interestingly, PIF, PEF, $T_\text{i}$ and $T_\text{e}$ were significantly different between wild-type and *mdx* mice at 16 months of age, indicating altered respiratory timing and flow in advanced dystrophic disease, despite overall preservation of ventilatory performance.

To circumvent some of the limitations in the use of whole-body plethysmography, in anaesthetised animals, ventilatory parameters including $V_\text{T}$, PIF and PEF were assessed directly via pneumotachometer during conditions of sequentially increasing ventilatory demand (baseline; chemoreflex challenge; baseline following vagotomy; chemoreflex challenge following vagotomy). Direct measures offer an advantage over plethysmography notwithstanding the confounding factor of anaesthesia.

**Table 7. Four- and 16-month-old parasternal muscle cytokine concentrations**

| (pg mg$^{-1}$ protein homogenate) | 4 months parasternal | | | 16 months parasternal | | |
|---|---|---|---|---|---|---|
| | Wild-type (*n* = 9–10) | *mdx* (*n* = 9–10) | *P* value | Wild-type (*n* = 8–10) | *mdx* (*n* = 9–10) | *P* value |
| IL-1$\beta$ | 0.0002 ± 0.0004 | 0.0722 ± 0.0320 | **<0.001** | 0.0048 ± 0.0094 | 0.0478 ± 0.0289 | **<0.001** |
| IL-2 | 0.093 ± 0.006 | 0.093 ± 0.010 | 0.903 | 0.085 ± 0.010 | 0.087 ± 0.010 | 0.600 |
| IL-5 | 0.065 ± 0.006 | 0.058 ± 0.005 | 0.019 | 0.069 ± 0.007 | 0.060 ± 0.005 | 0.006 |
| IL-6 | 1.07 ± 0.11 | 1.49 ± 0.38 | **0.002** | 1.22 ± 0.24 | 1.59 ± 0.35 | 0.013 |
| IL-9 | 1.75 ± 0.33 | 3.08 ± 1.63 | 0.031 | 1.86 ± 0.43 | 1.97 ± 0.31 | 0.540 |
| IL-10 | 0.25 ± 0.04 | 0.24 ± 0.05 | 0.629 | 0.24 ± 0.03 | 0.24 ± 0.03 | 0.943 |
| IL-12p70 | 5.73 ± 1.46 | 5.51 ± 1.32 | 0.729 | 5.61 ± 1.30 | 4.80 ± 0.94 | 0.123 |
| IL-15 | 5.69 ± 0.35 | 7.09 ± 0.76 | **<0.001** | 5.63 ± 0.27 | 5.90 ± 0.56 | 0.077 |
| IL-17A/F | 0.084 ± 0.006 | 0.099 ± 0.009 | **<0.001** | 0.087 ± 0.010 | 0.080 ± 0.006 | 0.071 |
| IL-27p28/IL-30 | 0.45 ± 0.08 | 0.33 ± 0.02 | **<0.001** | 0.50 ± 0.10 | 0.43 ± 0.03 | 0.052 |
| IL-33 | 4.10 ± 0.59 | 9.02 ± 2.93 | **<0.001** | 4.74 ± 0.82 | 9.91 ± 2.57 | **<0.001** |
| KC/GRO | 0.10 ± 0.01 | 1.03 ± 0.50 | **<0.001** | 0.12 ± 0.02 | 0.71 ± 0.28 | **<0.001** |
| TNF-$\alpha$ | 0.026 ± 0.006 | 0.143 ± 0.049 | **<0.001** | 0.022 ± 0.002 | 0.095 ± 0.046 | **<0.001** |
| MIP-1$\alpha$ | 0.065 ± 0.006 | 1.228 ± 0.749 | **<0.001** | 0.060 ± 0.007 | 0.660 ± 0.333 | **<0.001** |
| MIP-2 | 0.038 ± 0.008 | 0.424 ± 0.295 | **<0.001** | 0.041 ± 0.009 | 0.250 ± 0.130 | **<0.001** |
| MCP-1 | 0.80 ± 0.03 | 13.61 ± 12.92 | 0.018 | 0.81 ± 0.10 | 4.89 ± 3.18 | **<0.001** |
| IP-10 | 0.085 ± 0.026 | 1.011 ± 0.228 | **<0.001** | 0.15 ± 0.15 | 0.84 ± 0.36 | **<0.001** |

Group data (mean ± SD) for parasternal muscle cytokine concentrations for 4- and 16-month-old wild-type and *mdx* mice. Cytokines include interleukin-1$\beta$ (IL-1$\beta$), interleukin-2 (IL-2), interleukin-5 (IL-5), interleukin-6 (IL-6), interleukin-9 (IL-9), interleukin-10 (IL-10), interleukin-12p70 (IL-12p70), interleukin-15 (IL-15), interleukin-17A/F (IL-17A/F), interleukin-27p28/interleukin-30 (IL-27p28/IL-30), interleukin-33 (IL-33), keratinocyte chemoattractant/human growth-regulated oncogene (KC/GRO), tumour necrosis factor-$\alpha$ (TNF-$\alpha$), macrophage inflammatory protein-1$\alpha$ (MIP-1$\alpha$), macrophage inflammatory protein-2 (MIP-2), monocyte chemoattractant protein-1 (MCP-1) and interferon gamma-induced protein 10 (IP-10). Data were statistically compared using unpaired Student's *t* tests, with Welch's correction used where appropriate. Mann–Whitney non-parametric tests were used to compare data that were not normally distributed. *P* < 0.003 was considered statistically significant, following Bonferroni correction for multiple comparisons.

Bilateral vagotomy abolishes the negative feedback inhibition of breathing mediated via pulmonary stretch afferents via the vagus nerves, namely the Hering–Breuer reflex, allowing the assessment of breathing that exceeds volume-related measures in conscious mice, which is further enhanced by superimposing chemoactivation of respiratory motor outflow. In spontaneously breathing mice, chemostimulation elicits a significantly greater phrenic motor response following vagotomy compared to vagi intact (Kline et al., 2002). $\dot{V}_I$ was equivalent between anaesthetised wild-type and *mdx* mice at 4 and 16 months during all experimental conditions, confirming observations in conscious mice from the present study using plethysmography and consistent with our previous reports in anaesthetised animals (Burns, Drummond, et al., 2019; Burns, Murphy, et al., 2019; O'Halloran et al., 2023). Similar to our plethysmography studies, PIF was significantly lower in anaesthetised 16-month-old *mdx* compared to wild-type during the highest ventilatory demand condition examined (chemo-reflex challenge following vagotomy). We have previously reported reduced PIF in anaesthetised *mdx* compared to wild-type, which emerges at 16 months of age (O'Halloran et al., 2023). Our results demonstrate that ventilatory

performance is generally well preserved in *mdx* mice up to 16 months of age when decreases in peak flow metrics emerge.

By contrast to ventilation, which is relatively well protected until end-stage disease, we hypothesised that non-ventilatory behaviours, such as airway occlusion, airway clearance and respiratory reflexes such as coughing and sneezing, which are critical for protecting pulmonary function and require substantially greater respiratory muscle activation and force production, are impaired in advanced dystrophic disease. Dysphagia and poor airway protection are features of DMD, increasing the risk of infection, sleep disordered breathing including obstructive airway events and ultimately respiratory failure (LoMauro et al., 2014, 2017; Toussaint et al., 2016). Peak respiratory pressures are low in DMD and progressively decline with disease severity (Khirani et al., 2014).

We previously established that peak inspiratory pressure is preserved in young *mdx* mice, up to 8 months of age (Burns, Murphy, et al., 2019; O'Halloran et al., 2023). In the present study, inspiratory pressure-generating capacity was assessed in anaesthetised wild-type and *mdx* mice in early and

**Table 8. Four- and 16-month-old sternomastoid muscle cytokine concentrations**

| (pg mg$^{-1}$ protein homogenate) | 4 months sternomastoid | | | 16 months sternomastoid | | |
|---|---|---|---|---|---|---|
| | Wild-type ($n$ = 9–10) | *mdx* ($n$ = 9–10) | $P$ value | Wild-type ($n$ = 8–10) | *mdx* ($n$ = 8–10) | $P$ value |
| IL-1$\beta$ | 0.004 ± 0.002 | 0.094 ± 0.073 | **<0.001** | 0.004 ± 0.003 | 0.113 ± 0.034 | **<0.001** |
| IL-2 | 0.064 ± 0.009 | 0.068 ± 0.010 | 0.302 | 0.062 ± 0.008 | 0.072 ± 0.006 | 0.005 |
| IL-5 | 0.072 ± 0.010 | 0.060 ± 0.008 | 0.008 | 0.066 ± 0.008 | 0.060 ± 0.008 | 0.119 |
| IL-6 | 0.98 ± 0.09 | 1.46 ± 0.67 | 0.004 | 0.97 ± 0.20 | 1.39 ± 0.21 | **<0.001** |
| IL-9 | 1.70 ± 0.43 | 1.51 ± 0.41 | 0.325 | 1.78 ± 0.25 | 1.52 ± 0.32 | 0.057 |
| IL-10 | 0.23 ± 0.04 | 0.20 ± 0.05 | 0.121 | 0.23 ± 0.03 | 0.20 ± 0.04 | 0.224 |
| IL-12p70 | 4.27 ± 1.43 | 4.64 ± 1.15 | 0.522 | 5.38 ± 1.26 | 5.51 ± 1.49 | 0.835 |
| IL-15 | 3.73 ± 0.34 | 5.85 ± 0.86 | **<0.001** | 4.76 ± 0.79 | 5.78 ± 0.49 | 0.004 |
| IL-17A/F | 0.096 ± 0.016 | 0.131 ± 0.019 | **<0.001** | 0.12 ± 0.01 | 0.14 ± 0.03 | 0.039 |
| IL-27p28/IL-30 | 0.58 ± 0.09 | 0.43 ± 0.05 | **<0.001** | 0.66 ± 0.14 | 0.54 ± 0.07 | 0.024 |
| IL-33 | 14.24 ± 3.05 | 49.96 ± 12.70 | **<0.001** | 23.08 ± 4.35 | 71.45 ± 16.59 | **<0.001** |
| KC/GRO | 0.19 ± 0.045 | 2.10 ± 1.34 | **0.001** | 0.17 ± 0.03 | 1.15 ± 0.40 | **<0.001** |
| TNF-$\alpha$ | 0.033 ± 0.007 | 0.231 ± 0.139 | **<0.001** | 0.029 ± 0.007 | 0.158 ± 0.039 | **<0.001** |
| MIP-1$\alpha$ | 0.10 ± 0.004 | 1.68 ± 1.80 | **<0.001** | 0.100 ± 0.007 | 1.355 ± 0.427 | **<0.001** |
| MIP-2 | 0.047 ± 0.015 | 0.523 ± 0.418 | **<0.001** | 0.050 ± 0.014 | 0.371 ± 0.092 | **<0.001** |
| MCP-1 | 0.66 ± 0.06 | 18.9 ± 21.5 | **<0.001** | 0.63 ± 0.04 | 7.40 ± 6.92 | 0.013 |
| IP-10 | 0.082 ± 0.011 | 1.496 ± 0.951 | **0.001** | 0.41 ± 0.40 | 1.19 ± 0.48 | **0.001** |

Group data (mean ± SD) for sternomastoid muscle cytokine concentrations for 4- and 16-month-old wild-type and *mdx* mice. Cytokines include interleukin-1$\beta$ (IL-1$\beta$), interleukin-2 (IL-2), interleukin-5 (IL-5), interleukin-6 (IL-6), interleukin-9 (IL-9), interleukin-10 (IL-10), interleukin-12p70 (IL-12p70), interleukin-15 (IL-15), interleukin-17A/F (IL-17A/F), interleukin-27p28/interleukin-30 (IL-27p28/IL-30), interleukin-33 (IL-33), keratinocyte chemoattractant/human growth-regulated oncogene (KC/GRO), tumour necrosis factor-$\alpha$ (TNF-$\alpha$), macrophage inflammatory protein-1$\alpha$ (MIP-1$\alpha$), macrophage inflammatory protein-2 (MIP-2), monocyte chemoattractant protein-1 (MCP-1) and interferon gamma-induced protein 10 (IP-10). Data were statistically compared using unpaired Student's $t$ tests, with Welch's correction used where appropriate. Mann–Whitney non-parametric tests were used to compare data that were not normally distributed. $P < 0.003$ was considered statistically significant, following Bonferroni correction for multiple comparisons.

advanced dystrophic disease, during conditions of sequentially increasing ventilatory demand (baseline; chemoreflex challenge; baseline following vagotomy; chemoreflex challenge following vagotomy; airway occlusion). We demonstrated that inspiratory pressure generation was equivalent during basal breathing ($F_{IO_2}$ = 0.60) in wild-type and *mdx* mice in early disease. Acute chemoactivation ($F_{IO_2}$ = 0.15 and $F_{ICO_2}$ = 0.06), vagotomy and subsequent acute chemo-activation ($F_{IO_2}$ = 0.15 and $F_{ICO_2}$ = 0.06) all incrementally increased ventilatory drive and the magnitude of the enhanced inspiratory pressure was equivalent between wild-type and *mdx* mice in early disease, confirming the compensation of mechanical deficits in early dystrophic disease. Peak inspiratory pressure was determined during sustained tracheal occlusion until task failure, which results in maximum activation of the respiratory muscles (Burns, Murphy, et al., 2019; O'Halloran et al., 2023), generating impressive sub-atmospheric inspiratory pressures. Peak inspiratory pressure was preserved in *mdx* mice at 4 months of age but was significantly decreased in *mdx* mice at 16 months of age, indicating early compensation that is partly lost as disease progresses.

Based on our previous work (Burns, Murphy, et al., 2019; O'Halloran et al., 2023) and knowledge that the diaphragm, the principal inspiratory muscle, is profoundly impaired early in disease (reduced peak EMG and force-generating capacity), we hypothesised that accessory respiratory muscle support in early disease compensates for impaired diaphragm function, under-pinning the maintenance of peak inspiratory pressure generation. However, we posit that these muscles will ultimately succumb to dystrophic injury. We hypothesise that injury and remodelling of accessory respiratory muscles underpins the decline in peak pressure generation in advanced disease in the *mdx* mouse and ultimately contributes to progressive respiratory system compromise and failure, against the backdrop of profound obligatory muscle dysfunction.

In anaesthetised wild-type and *mdx* mice, we examined obligatory (diaphragm, external intercostal and parasternal) and accessory (sternomastoid, cleidomastoid, scalene and trapezius) respiratory muscle electrical activation under conditions of incremental neural drive in early and advanced disease. Our study confirmed reduced diaphragm EMG activity at 4 months of age (Burns, Murphy, et al., 2019; O'Halloran et al., 2023). Some accessory (sternomastoid, cleidomastoid and scalene) EMG activities were also lower in *mdx* compared to wild-type mice in early dystrophic disease. Interestingly,

**Table 9. Four- and 16-month-old scalene muscle cytokine concentrations**

| (pg mg$^{-1}$ protein homogenate) | 4 months scalene | | | 16 months scalene | | |
|---|---|---|---|---|---|---|
| | Wild-type (n = 8–10) | mdx (n = 8–10) | P value | Wild-type (n = 9–10) | mdx (n = 7– 10) | P value |
| IL-1$\beta$ | 0.013 ± 0.002 | 0.052 ± 0.032 | 0.007 | 0.013 ± 0.005 | 0.051 ± 0.022 | **<0.001** |
| IL-2 | 0.040 ± 0.012 | 0.050 ± 0.004 | 0.038 | 0.039 ± 0.012 | 0.060 ± 0.012 | **0.001** |
| IL-5 | 0.021 ± 0.003 | 0.027 ± 0.007 | 0.009 | 0.023 ± 0.004 | 0.021 ± 0.005 | 0.225 |
| IL-6 | 0.62 ± 0.09 | 1.04 ± 0.53 | **0.001** | 0.65 ± 0.21 | 1.15 ± 0.46 | 0.003 |
| IL-9 | 1.87 ± 0.64 | 1.52 ± 0.53 | 0.198 | 1.96 ± 0.52 | 1.45 ± 0.30 | 0.016 |
| IL-10 | 0.20 ± 0.03 | 0.17 ± 0.04 | 0.079 | 0.20 ± 0.04 | 0.26 ± 0.07 | 0.054 |
| IL-12p70 | 1.89 ± 1.16 | 2.11 ± 1.28 | 0.696 | 1.97 ± 1.19 | 2.93 ± 2.06 | 0.217 |
| IL-15 | 4.57 ± 1.09 | 4.74 ± 1.97 | 0.827 | 4.81 ± 1.86 | 3.93 ± 2.00 | 0.085 |
| IL-17A/F | 0.099 ± 0.031 | 0.088 ± 0.037 | 0.476 | 0.089 ± 0.040 | 0.060 ± 0.020 | 0.054 |
| IL-27p28/IL-30 | 0.178 ± 0.102 | 0.078 ± 0.052 | 0.013 | 0.20 ± 0.11 | 0.13 ± 0.04 | 0.059 |
| IL-33 | 6.34 ± 0.83 | 13.51 ± 10.88 | 0.173 | 5.41 ± 4.72 | 7.77 ± 12.42 | 0.781 |
| KC/GRO | 0.14 ± 0.02 | 0.94 ± 0.41 | **<0.001** | 0.13 ± 0.03 | 0.87 ± 0.46 | **0.001** |
| TNF-$\alpha$ | 0.021 ± 0.005 | 0.104 ± 0.028 | **<0.001** | 0.023 ± 0.006 | 0.120 ± 0.051 | **<0.001** |
| MIP-1$\alpha$ | 0.018 ± 0.017 | 0.160 ± 0.169 | 0.036 | 0.026 ± 0.022 | 0.029 ± 0.006 | 0.364 |
| MIP-2 | 0.054 ± 0.021 | 0.104 ± 0.074 | 0.086 | 0.050 ± 0.021 | 0.022 ± 0.002 | **0.002** |
| MCP-1 | 0.67 ± 0.09 | 1.04 ± 0.57 | 0.109 | 0.63 ± 0.07 | 0.53 ± 0.003 | **0.002** |
| IP-10 | 0.067 ± 0.017 | 0.819 ± 0.643 | 0.005 | 0.143 ± 0.107 | 0.079 ± 0.008 | 0.230 |

Group data (mean ± SD) for scalene muscle cytokine concentrations for 4- and 16-month-old wild-type and *mdx* mice. Cytokines include interleukin-1$\beta$ (IL-1$\beta$), interleukin-2 (IL-2), interleukin-5 (IL-5), interleukin-6 (IL-6), interleukin-9 (IL-9), interleukin-10 (IL-10), interleukin-12p70 (IL-12p70), interleukin-15 (IL-15), interleukin-17A/F (IL-17A/F), interleukin-27p28/interleukin-30 (IL-27p28/IL-30), interleukin-33 (IL-33), keratinocyte chemoattractant/human growth-regulated oncogene (KC/GRO), tumour necrosis factor-$\alpha$ (TNF-$\alpha$), macrophage inflammatory protein-1$\alpha$ (MIP-1$\alpha$), macrophage inflammatory protein-2 (MIP-2), monocyte chemoattractant protein-1 (MCP-1) and interferon gamma-induced protein 10 (IP-10). Data were statistically compared using unpaired Student's *t* tests, with Welch's correction used where appropriate. Mann–Whitney non-parametric tests were used to compare data that were not normally distributed. $P < 0.003$ was considered statistically significant, following Bonferroni correction for multiple comparisons.

external intercostal, parasternal and trapezius EMG activities were equivalent between wild-type and *mdx* mice in early disease. In support of our hypothesis, further impairments to respiratory muscle electrical activation were evident at 16 months of age with the emergence of lower external intercostal and trapezius muscle EMG activities in *mdx* compared to wild-type mice. Phrenic nerve axonopathy and neuromuscular junction remodelling and dysfunction have been reported in *mdx* mice, probably underpinning the reduced respiratory EMG activities in *mdx* mice (Carlson & Roshek, 2001; Dhindsa et al., 2020; Personius & Sawyer, 2006; Pratt et al., 2015).

Our study reveals that *mdx* obligatory and accessory EMGs (and inspiratory pressure) are relatively well preserved within the ventilatory range and deficits are not revealed until the performance of non-ventilatory behaviours (maximal activation of respiratory muscles) in 16-month-old *mdx* mice. Although there was evidence in our study of a progressive change in accessory muscle EMG performance between 4 and 16 months of age in *mdx* mice, the difference between early and advanced disease was not striking. Impairments in accessory EMGs appeared early, coincident with diaphragm muscle

impairments suggesting early recruitment of accessory muscles in DMD. Further impairments in accessory EMGs were noted in 16-month-old *mdx* mice, which may have contributed to the decline in peak inspiratory performance but presumably do not fully account for it. Further analyses described below suggest that progressive decline in diaphragm performance probably contributed to the manifestation of decreased peak performance in *mdx* mice at 16 months of age.

Consistent with previous reports, we determined that diaphragm muscle weakness is evident in early disease in the *mdx* mouse model of DMD (Burns, Ali, et al., 2017; Burns, Roy, et al., 2017; Burns et al., 2018; Burns, Drummond, et al., 2019; Burns, Murphy, et al., 2019; Coirault et al., 2003; O'Halloran et al., 2023). Diaphragm muscle weakness was evidenced by significantly reduced twitch and tetanic force in *mdx* mice compared to wild-type mice. Specific force was considerably reduced within the stimulation frequency range of 25–150 Hz for *mdx* diaphragm compared to wild-type. This frequency range corresponds to a broad range of ventilatory and non-ventilatory behaviours, including basal breathing (Fogarty et al., 2018). Non-ventilatory behaviours including respiratory reflexes (e.g. cough and sneeze) and

**Table 10. Four- and 16-month-old trapezius muscle cytokine concentrations**

| (pg mg$^{-1}$ protein homogenate) | 4 months trapezius | | | 16 months trapezius | | |
|---|---|---|---|---|---|---|
| | Wild-type ($n$ = 8–10) | *mdx* ($n$ = 8–10) | $P$ value | Wild-type ($n$ = 8–10) | *mdx* ($n$ = 9–10) | $P$ value |
| IL-1$\beta$ | 0.011 ± 0.003 | 0.268 ± 0.155 | **<0.001** | 0.011 ± 0.003 | 0.264 ± 0.122 | **<0.001** |
| IL-2 | 0.077 ± 0.015 | 0.108 ± 0.018 | **<0.001** | 0.076 ± 0.010 | 0.112 ± 0.023 | **<0.001** |
| IL-5 | 0.19 ± 0.04 | 0.20 ± 0.02 | 0.607 | 0.22 ± 0.03 | 0.19 ± 0.02 | 0.022 |
| IL-6 | 0.89 ± 0.12 | 2.13 ± 0.87 | 0.005 | 0.92 ± 0.15 | 2.64 ± 1.49 | 0.005 |
| IL-9 | 2.21 ± 0.26 | 1.82 ± 0.55 | 0.066 | 1.92 ± 0.62 | 1.46 ± 0.60 | 0.109 |
| IL-10 | 0.17 ± 0.01 | 0.16 ± 0.03 | 0.258 | 0.20 ± 0.04 | 0.16 ± 0.04 | 0.045 |
| IL-12p70 | 3.97 ± 2.10 | 3.36 ± 1.03 | 0.616 | 3.57 ± 1.88 | 3.16 ± 0.86 | 0.853 |
| IL-15 | 6.59 ± 0.73 | 7.06 ± 0.58 | 0.105 | 5.77 ± 2.25 | 6.00 ± 2.50 | 0.720 |
| IL-17A/F | 0.13 ± 0.02 | 0.16 ± 0.03 | 0.059 | 0.14 ± 0.03 | 0.15 ± 0.02 | 0.309 |
| IL-27p28/IL-30 | 1.00 ± 0.24 | 0.71 ± 0.06 | 0.004 | 1.09 ± 0.24 | 1.08 ± 0.11 | 0.921 |
| IL-33 | 23.03 ± 5.61 | 44.36 ± 7.58 | **<0.001** | 24.50 ± 10.15 | 57.98 ± 20.87 | **<0.001** |
| KC/GRO | 0.37 ± 0.05 | 2.54 ± 1.59 | **0.002** | 0.45 ± 0.14 | 1.42 ± 0.41 | **<0.001** |
| TNF-$\alpha$ | 0.050 ± 0.020 | 0.322 ± 0.104 | **<0.001** | 0.048 ± 0.026 | 0.232 ± 0.041 | **<0.001** |
| MIP-1$\alpha$ | 0.080 ± 0.008 | 2.253 ± 0.790 | **<0.001** | 0.091 ± 0.026 | 1.961 ± 0.900 | **<0.001** |
| MIP-2 | 0.12 ± 0.02 | 1.08 ± 0.65 | **0.001** | 0.15 ± 0.06 | 0.56 ± 0.22 | **<0.001** |
| MCP-1 | 0.73 ± 0.04 | 52.94 ± 48.70 | 0.008 | 0.77 ± 0.15 | 9.15 ± 4.56 | **<0.001** |
| IP-10 | 0.18 ± 0.13 | 1.93 ± 0.73 | **<0.001** | 0.22 ± 0.12 | 2.38 ± 1.15 | **<0.001** |

Group data (mean ± SD) for trapezius muscle cytokine concentrations for 4- and 16-month-old wild-type and *mdx* mice. Cytokines include interleukin-1$\beta$ (IL-1$\beta$), interleukin-2 (IL-2), interleukin-5 (IL-5), interleukin-6 (IL-6), interleukin-9 (IL-9), interleukin-10 (IL-10), interleukin-12p70 (IL-12p70), interleukin-15 (IL-15), interleukin-17A/F (IL-17A/F), interleukin-27p28/interleukin-30 (IL-27p28/IL-30), interleukin-33 (IL-33), keratinocyte chemoattractant/human growth-regulated oncogene (KC/GRO), tumour necrosis factor-$\alpha$ (TNF-$\alpha$), macrophage inflammatory protein-1$\alpha$ (MIP-1$\alpha$), macrophage inflammatory protein-2 (MIP-2), monocyte chemoattractant protein-1 (MCP-1) and interferon gamma-induced protein 10 (IP-10). Data were statistically compared using unpaired Student's *t* tests, with Welch's correction used where appropriate. Mann–Whitney non-parametric tests were used to compare data that were not normally distributed. $P < 0.003$ was considered statistically significant, following Bonferroni correction for multiple comparisons.

airway clearance are important for the maintenance of a patent airway and protection of pulmonary performance. Infection and sleep disordered breathing including obstructive airway events are well-recognised features of DMD, contributing to disease progression and ultimately respiratory failure (LoMauro et al., 2017; Sheehan et al., 2018). Force increases in *mdx* diaphragm muscle as a function of stimulation frequency in early and advanced disease, indicating a capacity to increase breathing in response to increased neural drive despite the intrinsic muscle weakness, which has been reported in *mdx* diaphragm muscle as early as 1 month of age (O'Halloran et al., 2023). However, as stimulation frequency increases, the force-generating capacity of *mdx* muscles are increasingly compromised.

Force loss in *mdx* diaphragm muscle was accompanied by significant decreases in muscle $S_{max}$ and work- and power-generating capacity, evident from early disease, with the emergence of impaired $V_{max}$ in *mdx* diaphragm in advanced disease, indicating altered cross-bridge cycling kinetics in *mdx* muscles, which may occur as a result of fibre type transitions in dystrophic muscle. Single fibre studies in rats have shown that type I fibres have slower shortening velocities (and reduced specific force),

which is consistent with slower cross-bridge cycling kinetics (Geiger et al., 2000; Sieck & Prakash, 1997). Impaired muscle shortening is evident in *mdx* diaphragm in early disease, and is exacerbated in advanced dystrophic disease, indicating progressive muscle stiffness in *mdx* muscles, present from early disease.

Given our previous and current findings that peak inspiratory pressure is preserved in early dystrophic disease despite diaphragm muscle weakness and reduced diaphragm and accessory respiratory muscle EMG activities (Burns, Drummond, et al., 2019; Burns, Murphy, et al., 2019; O'Halloran et al., 2023), we extended our functional assessments to include additional obligatory (parasternal) and accessory (sternomastoid, cleidomastoid and scalene) respiratory muscles. We hypothesised that early maintenance of peak inspiratory pressure was the result of accessory muscle compensation, which we anticipated would be progressively lost in advanced disease leading to reduction in peak pressure generation.

We determined that parasternal, sternomastoid and cleidomastoid muscle weakness is evident in early disease, evidenced by significantly reduced twitch and tetanic force in *mdx* mice compared to wild-type mice. Inter-

estingly, the functional capacity of the scalene muscle was preserved in *mdx* in early disease. However, given the much lower force generated by the scalene compared to diaphragm muscle, it is unlikely that the scalene muscle exclusively compensates for the deficits in diaphragm muscle performance in *mdx* mice. Additionally, the electrical activation of the scalene muscle is reduced in *mdx* mice, which might compromise function *in vivo*. Consistent with our hypothesis of progressive decline in accessory muscle function, scalene muscle weakness presents in advanced disease, evidenced by significantly lower twitch and tetanic force in *mdx* compared to wild-type mice. TTP is also lower in *mdx* scalene muscle compared to wild-type muscle in advanced disease, indicating altered muscle kinetics. Scalene muscle $S_{max}$ and $V_{max}$ are lower in *mdx* compared to wild-type mice in advanced disease.

Muscle weakness occurs in DMD because of structural, biochemical and physiological alterations in muscle because of the lack of dystrophin, leading to progressive muscle degeneration and impaired regeneration, ultimately impacting function. $Ca^{2+}$ dysregulation, increased reactive oxygen species and structural remodelling, including changes in myosin isoform composition (shift from myosin heavy chain type IIx to IIa), inflammation, necrosis and fibrosis have been reported in *mdx* muscles, contributing to muscle dysfunction and weakness (Burns et al., 2018; Coirault et al., 1999; Deconinck & Dan, 2007; Whitehead et al., 2006). Structural abnormalities and remodelling are a feature of dystrophic muscle because of muscle fibre damage, degeneration and regeneration and contribute to the functional deficits. Interestingly, diaphragm muscle weakness presents in *mdx* in the first few weeks of life, before fibrosis is evident revealing that weakness is inherent to the contractile tissue (Coirault et al., 2003; O'Halloran et al., 2023).

Analysis of muscle fibre size and distribution revealed substantial remodelling in obligatory (diaphragm, intercostal and parasternal) and accessory (sternomastoid, cleidomastoid, scalene and trapezius) respiratory muscles in early dystrophic disease, with increased variation of *mdx* muscle fibre size, an increased proportion of smaller fibres in *mdx* and a reduced proportion of larger muscle fibres in *mdx* muscles compared to wild-type mice. Large fibres produce greater force than small muscle fibres (Mantilla & Sieck, 2003) and thus the shift in fibre size distribution in *mdx* obligatory and accessory respiratory muscles probably has functional implications, consistent with *mdx* muscles exhibiting lower force- and power-generating capacity than wild-type muscles *ex vivo*. Further remodelling was evident in advanced disease, with significantly reduced myofibre size in all obligatory (diaphragm, intercostal and parasternal) and accessory (sternomastoid, cleidomastoid, scalene and trapezius)

muscles examined in *mdx* compared to wild-type mice. The increased proportion of smaller muscle fibres in dystrophic muscle is the result of ongoing muscle fibre damage and repair processes preventing muscle fibre MyHC isoform maturation during muscle development, with consequences for muscle function. We have previously reported an increased proportion of type IIa muscle fibres and a decreased proportion of type IIx muscle fibres in young *mdx* diaphragm muscle (Burns et al., 2018) with functional implications, given that type IIx fibres produce greater force than type IIa fibres.

Systemic and muscle inflammation are cardinal features of DMD, secondary to muscle fibre damage as a result of dystrophin deficiency. Muscle biopsies and plasma samples from DMD boys and *mdx* mice have revealed elevated expression of chemokines and pro-inflammatory cytokines (Cruz-Guzmán et al., 2015; De Paepe & De Bleecker, 2013; De Pasquale et al., 2012; Messina et al., 2011; Rufo et al., 2011). Immune cells are recruited to damaged muscle regions, where they infiltrate and mount an inflammatory response through activation of cytokines and recruitment of additional immune cells to the damaged muscle to repair injured fibres (De Paepe & De Bleecker, 2013). Although inflammation is a natural response to tissue damage and important to stimulate muscle regeneration, chronic inflammation in DMD disrupts normal physiological function and exacerbates muscle damage, interfering with muscle repair.

In the present study, we report significantly higher concentrations of pro-inflammatory cytokines IL-1$\beta$, IL-6, IL-33 and TNF-$\alpha$ and chemokines KC/GRO, MIP-1$\alpha$, MIP-2, MCP-1 and IP-10 in *mdx* obligatory (diaphragm, intercostal and parasternal) and accessory (sternomastoid, scalene and trapezius) respiratory muscles in both early and advanced dystrophic disease (with some exceptions), indicating a pro-inflammatory signature, consistent with previous reports in *mdx* muscle (Burns et al., 2018; Howard et al., 2021; Pelosi et al., 2015).

The elevated expression of inflammatory cytokines in *mdx* muscles was associated with a significant increase in the relative area of putative inflammatory cell infiltration in *mdx* obligatory (diaphragm and intercostal) and accessory (cleidomastoid, scalene and trapezius) respiratory muscles compared to wild-type muscles in early disease. The relative area of putative inflammatory cell infiltration was slightly further increased in *mdx* obligatory (parasternal and intercostal) and accessory (sternomastoid, cleidomastoid, scalene and trapezius) respiratory muscles examined in advanced disease. Interestingly, the relative area of infiltration of putative inflammatory cells substantially increased between 4 and 16 months of age in the diaphragm muscle of *mdx* mice.

The relative density of muscle fibres with centrally located nuclei in *mdx* obligatory (diaphragm, intercostal and parasternal) and accessory (sternomastoid,

cleidomastoid, scalene and trapezius) respiratory muscles was significantly increased compared to wild-type muscles in early and advanced disease, indicating muscle fibre repair and regeneration following damage. Notably, the proportion of muscle fibres with central nuclei was reduced in 16-month-old *mdx* muscles compared to 4-month-old *mdx* muscles, indicating exhausted regenerative capacity of the muscle fibres in advanced disease, reflective of impaired muscle repair leading to subsequent myofibre necrosis and gradual replacement of muscle fibres with adipose and connective tissue leading to fibrosis, which further disrupts muscle regeneration (Carpenter & Karpati, 1979; Juban et al., 2018; Tidball & Villalta, 2010).

We determined significantly greater collagen content (fibrosis) in *mdx* obligatory (diaphragm, intercostal and parasternal) and accessory (sternomastoid, cleidomastoid, scalene and trapezius) respiratory muscles compared to wild-type, consistent with previous observations in *mdx* diaphragm muscles (Barros Maranhão et al., 2015; Burns et al., 2018; Burns, Drummond, et al., 2019; Gosselin & McCormick, 2004; Gutpell et al., 2015; Ishizaki et al., 2008; O'Halloran et al., 2023; Smith et al., 2016; Stedman et al., 1991) and probably driven by TGF-$\beta$-dependent pathways (Burks & Cohn, 2011; Juban et al., 2018; O'Halloran et al., 2023; Zhou & Lu, 2010). Interestingly, the relative area of collagen deposition appeared to be equivalent between 4- and 16-month-old *mdx* muscles, except for the diaphragm muscle. Collagen deposition substantially increased in 16-month-old *mdx* diaphragm muscle, compared to 4-month-old *mdx* diaphragm muscle, indicating increased fibrosis as disease progresses. Fibrosis probably has functional implications for muscle, consistent with *mdx* diaphragm muscle exhibiting lower force- and power-generating capacity *ex vivo*, evident from early disease. In addition, reductions in $L_O$, $S_{max}$ and $V_{max}$ were affected to a greater extent in diaphragm muscle in advanced dystrophic disease, revealing progressive dysfunction consistent with greater fibrosis. Studies have suggested that diaphragm muscle remodelling and repurposing in early dystrophic disease provides a mechanical advantage to extra-diaphragmatic muscles, allowing ventilatory and non-ventilatory behaviours to be protected (Mead et al., 2014; O'Halloran et al., 2023). However, progressive fibrosis of the respiratory muscles has deleterious effects for respiratory performance. We anticipated that progressive changes in accessory muscles might be most striking in *mdx* mice comparing 4- and 16-month-old animals, but it is evident that the dominant phenotype appears in the diaphragm. It thus appears probable that impairment of peak inspiratory performance is dependent upon progressive diaphragm dysfunction accompanied by progressive accessory muscle dysfunction. We expect that this phenotype extends to end-stage disease with profound loss of contractile performance of the diaphragm and a persistent worsening of the form, function and control of extra-diaphragmatic muscles of breathing culminating in respiratory failure.

## Limitations

It can be difficult to reliably compare EMG signals between muscles and animals. To facilitate comparisons, we adopted a systematic approach to EMG recording and analysis. Electrode placement was standardised *a priori* as described in the Methods, consistent with our previous approach (Burns, Murphy, et al., 2019; O'Halloran et al., 2023). All signals were processed in the same fashion (amplification, filtering and integration). Integrated EMG data are presented with absolute units to allow for faithful comparisons between muscles and between groups (i.e. data are not obscured by presenting relative changes as percentage changes or normalised to a reference value). The principal conclusion we have drawn from EMG analyses is that there is lower activation in *mdx* mice particularly during high demand behaviours. The intramuscular electrode measures multi-unit activity, which is phasic with inspiration. The altered EMG signal in *mdx* mice reflects decreased complexity in the composite signal and lower amplitude motor units. We have quantified the composite signal using the standard approach of measuring the amplitude of the rectified integrated signal (the outcome is the same if area under the curve of the integral is assessed). An absence of large amplitude units is discernible from our recordings sampled at 20 kHz. However, spike sorting of motor units would be required to fully characterise these changes in *mdx* muscles. Additionally, we acknowledge that assessments were made under general anaesthesia, which depresses motor drive and hence the complexity of EMG signals.

We are confident that we recorded external intercostal EMG based on placement of the electrode. However, for histology, although the tissue was arranged to facilitate identification of external intercostal muscle separate to internal intercostal, there were occasions where this could not explicitly be determined with absolute confidence. Given the limited tissue in mouse and the requirement for a minimum amount of protein for the cytokine assays, intercostal bundles were used and again we cannot be certain that it was exclusively external intercostal. For this reason, we have labelled the relevant Figures and Tables as 'Intercostal' and the data should be regarded accordingly.

Kyphosis is a feature of advanced DMD further contributing to respiratory involvement in neuromuscular disease. Kyphosis was not present in early disease, but it was evident in some *mdx* mice at 16 months of age, but it was not systematically scored in our study.

Late-stage DMD is characterised by cardiorespiratory morbidity with the emergence of dilated cardiomyopathy. We did not assess cardiac function in *mdx* mice at any age in our study. It is established that *mdx* mice do not show evidence of dilated cardiomyopathy as young adults but do so before 1 year of age, with evidence of fibrosis in late-stage disease (Quinlan et al., 2004). We acknowledge that cardiac burden is a factor relevant to our considerations of respiratory performance comparing 4- and 16-month-old *mdx* mice since cardiac burden has implications for respiratory control. Notably, however, ventilatory capacity was generally well maintained in 16-month-old *mdx* mice, again highlighting a remarkable compensation although it is generally regarded that the cardiac manifestation of dilated cardiomyopathy and diffuse chamber fibrosis in *mdx* mice is mild compared to human DMD.

**Conclusions.** In conclusion, despite the profound muscle weakness evident in *mdx* obligatory and accessory respiratory muscles, peak performance and ventilatory behaviours are protected until 16 months of age in *mdx* mice, indicating compensation within the respiratory control system. This compensation is eroded as disease progresses leading to the loss of peak inspiratory pressure-generating capacity in *mdx* mice. Progressive decline in diaphragm and extra-diaphragmatic muscles appears to contribute to respiratory system compromise in advanced disease. Studies are required in older *mdx* mice to reveal the timeline and causes of ventilatory insufficiency to fully elucidate the mechanisms underpinning respiratory system failure in end-stage disease. Given the logistical challenges posed by the substantial compensation presenting in the *mdx* model of DMD, the present study highlights the need for more progressive models of disease, with reduced lifespan to feasibly study end-stage disease, when profound respiratory system impairments are evident. Comprehensive characterisation of the respiratory control system at end-stage disease in pre-clinical models of DMD such as D2.*mdx* and dystrophin/utrophin double knockout *mdx*/utrn$^{-/-}$ mice (Delaney & O'Halloran, 2024) is required to provide a platform to test existing and emerging therapies to mitigate respiratory system dysfunction.

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

## Additional information

### Data availability statement

Individual data points are shown in Figures for major data sets. All other data are reported in Tables as mean ± SD. Data are available from the corresponding author on reasonable request.

### Competing interests

The authors declare that they have no competing interests.

### Author contributions

A.D.S was responsible for experimental design; acquisition of whole-body plethysmography, EMG, muscle function, histology, immunofluorescence and cytokine data; data and statistical analysis; interpretation of data; preparation of figures and drafting of the manuscript; and revisions and responses to peer review. D.P.B was responsible for experimental design;

acquisition of whole-body plethysmography and muscle function data; data and statistical analysis; and interpretation of data. K.W, A.M, R.D and S.E.D were responsible for acquisition of histology and immunofluorescence data, and data analysis. G.J was responsible for acquisition of cytokine data and data analysis. K.D.O'H was responsible for experimental design; acquisition of EMG data; interpretation of data; drafting of the manuscript; and revisions and responses to peer review. All authors have read and approved the final version of the manuscript submitted for publication and agree to be accountable for all aspects of the work. All persons designated as authors qualify for authorship, and all those who qualify for authorship are listed.

## Funding

Research Ireland (formerly Science Foundation Ireland): SFI FFP/19/6628 INSPIRE DMD. ADS was funded by an Eli Lilly Doctoral Scholarship. RD was funded by The Physiological Society. KW and AM were supported by Erasmus+ funding from the European Commission. Additional stipend support for ADS was provided by the Department of Physiology, University College Cork.

## Acknowledgements

We are grateful to staff of the Biological Services Unit, University College Cork for excellent service in the management of the mouse colonies and provision of animal care ensuring the highest standards of welfare. Our special thanks to Anthony Marullo for generating the graphical abstract.

## Keywords

accessory muscles of breathing, cytokine concentrations, diaphragm, Duchenne muscular dystrophy, fibrosis, muscle structure and function, peak inspiratory pressure

## Supporting information

Additional supporting information can be found online in the Supporting Information section at the end of the HTML view of the article. Supporting information files available:

**Peer Review History**

