## [Peer Review History · The Journal of Physiology]

Obligatory and accessory respiratory muscle structure, function and control in early and advanced disease in the *mdx* mouse model of DMD

Aoife D Slyne, David P Burns, Karina Woeller, Amandine May, Roisin Dowd, Sarah E Drummond, Grzegorz Jasionek, and Ken D O'Halloran

DOI: 10.1113/JP288709

Corresponding author(s): Ken O'Halloran (k.ohalloran@ucc.ie)

The following individual(s) involved in review of this submission have agreed to reveal their identity: Carlos B Mantilla (Referee #2); Sean Williams (Referee #3)

Review Timeline:

Submission Date:	09-Feb-2025
Editorial Decision:	11-Mar-2025
Revision Received:	03-Apr-2025
Editorial Decision:	22-Apr-2025
Revision Received:	08-May-2025
Accepted:	16-May-2025

Senior Editor: Harold Schultz

Reviewing Editor: Kevin Murach

Transaction Report:

Dear Ken,

Re: JP-RP-2025-288709 "**Obligatory and accessory respiratory muscle structure, function and control in early and advanced disease in the *mdx* mouse model of DMD**" by Aoife D Slyne, David P Burns, Karina Woeller, Amandine May, Roisin Dowd, Sarah E Drummond, Grzegorz Jasioneck, and Ken D O'Halloran

Thank you for submitting your manuscript to The Journal of Physiology. It has been assessed by a Reviewing Editor and by 3 expert referees and we are pleased to tell you that it is potentially acceptable for publication following satisfactory major revision.

LANGUAGE EDITING AND SUPPORT FOR PUBLICATION: If you would like help with English language editing, or other article preparation support, Wiley Editing Services offers expert help, including English Language Editing, as well as translation, manuscript formatting, and figure formatting at www.wileyauthors.com/eoo/preparation. You can also find resources for Preparing Your Article for general guidance about writing and preparing your manuscript at www.wileyauthors.com/eoo/prepresources.

REVISION CHECKLIST:

We look forward to receiving your revised submission.

Best wishes,

Harold Schultz
Senior Editor
The Journal of Physiology

REQUIRED ITEMS

- Author photo and profile. First or joint first authors are asked to provide a short biography (no more than 100 words for one author or 150 words in total for joint first authors) and a portrait photograph. These should be uploaded and clearly labelled together in a Word document with the revised version of the manuscript. See Information for Authors for further details.

- Papers must comply with the Statistics Policy: https://jp.msubmit.net/cgi-bin/main.plex?form_type=display_requirements#statistics.

In summary:

- If $n \leq 30$, all data points must be plotted in the figure in a way that reveals their range and distribution. A bar graph with data points overlaid, a box and whisker plot or a violin plot (preferably with data points included) are acceptable formats.

- If $n > 30$, then the entire raw dataset must be made available either as supporting information, or hosted on a not-for-profit repository, e.g. FigShare, with access details provided in the manuscript.

- 'n' clearly defined (e.g. x cells from y slices in z animals) in the Methods. Authors should be mindful of pseudoreplication.

- All relevant 'n' values must be clearly stated in the main text, figures and tables.

- The most appropriate summary statistic (e.g. mean or median and standard deviation) must be used. Standard Error of the Mean (SEM) alone is not permitted.

- Exact p values must be stated. Authors must not use 'greater than' or 'less than'. Exact p values must be stated to three significant figures even when 'no statistical significance' is claimed.

- A Data Availability Statement is required for all papers reporting original data. This must be in the Additional Information section of the manuscript itself. It must have the paragraph heading 'Data Availability Statement'. All data supporting the results in the paper must be either: in the paper itself; uploaded as Supporting Information for Online Publication; or archived in an appropriate public repository. The statement needs to describe the availability or the absence of shared data. Authors must include in their statement: a link to the repository they have used, or a statement that it is available as Supporting Information; reference the data in the appropriate section(s) of their manuscript; and cite the data they have shared in the References section. Whenever possible, the scripts and other artefacts used to generate the analyses presented in the paper should also be publicly archived. If sharing data compromises ethical standards or legal requirements then authors are not expected to share it, but must note this in their statement. For more information, see our Statistics Policy.

- Please include an Abstract Figure file, as well as the Figure Legend text within the main article file. The Abstract Figure is a piece of artwork designed to give readers an immediate understanding of the research and should summarise the main conclusions. If possible, the image should be easily 'readable' from left to right or top to bottom. It should show the physiological relevance of the manuscript so readers can assess the importance and content of its findings. Abstract Figures should not merely recapitulate other figures in the manuscript. Please try to keep the diagram as simple as possible and without superfluous information that may distract from the main conclusion(s). Abstract Figures must be provided by authors no later than the revised manuscript stage and should be uploaded as a separate file during online submission labelled as File Type 'Abstract Figure'. Please also ensure that you include the figure legend in the main article file. All Abstract Figures should be created using BioRender. Authors should use The Journal's premium BioRender account to export high-resolution images. Details on how to use and access the premium account are included as part of this email.

EDITOR COMMENTS

Reviewing Editor:

Your work has been evaluated by two experts in the field. Both view the work as quite influential, and noted the careful and thoughtful manner in which the experiments were performed. While the overall critiques are minor, there are several points that require attention before further consideration can be made. Also, in the revision, please upload high-resolution figures so that the data can be scrutinized more closely.

Please also see 'Required Items' above.

Senior Editor:

Comments for Authors to ensure the paper complies with the Statistics Policy (Required):
see comments from Statistical Editor below (Referee #3).

Comments to the Author:

Thank you for submitting your research article to the Journal of Physiology for consideration. The article has been reviewed by experts in the field and found to be potentially acceptable for publication pending adequate revision to address all of the concerns raised. Please address all comments from the external referees and the statistical and reviewing editors as well as addressing the list of requirements or publication in the Journal provided in this letter and the Journal's guidelines for rigor and reproducibility.

REFEREE COMMENTS

Referee #1:

The manuscript by Slyne et al., centers on the muscle structure of the mdx mice during early and advanced stages of dystrophic disease progression. Emphasis is placed on the diaphragm muscles and remodeling that occurs in Duchenne muscular dystrophy (DMD) in these dystrophin-deficient mice. The authors identified significant muscle weakness and inspiration deficits that occurs at 16 months of age in mdx mice. By 20 months of age, the mdx mice have advanced diaphragm muscle weakness and inspiration deficits, thus relying on their "extra-diaphragmatic muscles". The authors concluded that the diaphragm and other skeletal muscle compensation occurs with the progressive muscle wasting and cytokine influx, with only the scalene and trapezius being spared.

Overall, this is an interesting and important study that seeks to clarify key DMD disease-relevant biomarkers and indicators within the mdx mouse strain. The experiments are designed logically, and are appropriately justified in power, number, and statistical analysis. My comments are mostly moderate and minor in scope as the manuscript is well-written and the experimental data is comprehensive in scope. If the authors can clarify certain aspects of the study, it would be a good addition to the DMD biology and biomarker fields.

General Comments:

1. Abstract, line 43. This statement is confusing: "More convenient models of rapid, progressive muscular dystrophy are required to study end-stage disease". What do the authors mean exactly by "convenient". This sentence is unclear and should be clarified.
2. The role of the heart, specifically dilated cardiomyopathy (DCM), that occurs towards the end stages of DMD. Did the authors notice an increased burden of DCM in their mice. Specifically, the increased cardiac burden would likely place a similar metabolic and inspiration burden on the mdx mice. Did the authors perform any ejection fraction measurements in the cohorts? Some discussion of the cardiac burden in late stage DMD as a factor of increased respiratory demand should be addressed in the discussion section.
3. Minor comment. Figures 10A-N. Did the authors run statistical analysis on the relative frequency for their min. ferret diameter measurements?
4. Figures 13 and 14. Can you clarify what the quantification or units of value are in the interleukin/cytokine values?

Referee #2:

This is a very comprehensive study of respiratory muscle adaptations in 4- and 16-month-old control and mdx mice. The work builds upon prior work by this team and provides an important, comprehensive evaluation of various respiratory muscles at early and late disease stages.

RESULTS. There are several issues in the presentation of findings that need attention to improve the reader's ability to interpret the results.

-Selection of respiratory muscles studied. The rationale for the inclusion of some respiratory muscles in the various measurements in the study is not clearly presented because not all measures were conducted in all muscles.

-Airway (tracheal) occlusion vs. obstruction. These various terms appear to be used interchangeably in both the text and figures. Based on the methods, it appears that it was tracheal occlusion. Please, consider using only one term consistently across the manuscript.

-Histological analyses. There is no information provided on the number of fibers analyzed histologically for each animal. Please, correct in the methods and results sections (and/or figures 8-11).

-Statistical analyses. For example, analyses of EMG amplitude were conducted using repeated measures two-way ANOVA, but in examining Figure 3 it is evident that for several muscles and measures (e.g. airway occlusion) the variance across groups is not equal. A mixed linear model may be more appropriate, as suggested was used for missing data. Please, justify. Note this issue applies to multiple other measurements, not just EMG. In addition, there is limited presentation of the main effects (shown only in the figures), yet the text expands on many post hoc analyses that are not adjusted for the many comparisons. This issue not only makes the text difficult to follow but also confounds interpretation. For instance, the use of 4 decimal points of precision in presenting the p values of these post hoc analyses is unwarranted.

-Figure quality. Some of the figures (both PDF and jpg formats) are of low quality. For example, figure 3 is blurry and figures 13 and 14 are not legible at all. In this regard the axis labels can be reviewed to ensure that only the necessary information is included, allowing for the use of larger fonts (e.g., in figure 3, the y-axis labels repeat the figure titles at the left of the figure). The H&E stains shown in figures 8 and 9 are too blurry to see centrally located nuclei or inflammatory infiltration.

Specific comments.

-Intercostal muscle. In some sections of the manuscript, it is specified that the external intercostal muscle was studied, but many other sections do not clarify this aspect. Please, correct accordingly.

-Line 222. Only oesophageal pressures were measured when it is common when studying respiratory mechanics to measure both oesophageal and gastric pressure to determine transdiaphragmatic pressures. Please, justify.

-Line 255. Sustained, terminal tracheal occlusion was used to elicit higher force respiratory muscle activation, but there is no support provided for the statement that this manoeuvre elicits "near maximal oesophageal pressure and respiratory muscle EMG activities". It is likely that this statement is an oversimplification of the complex, integrated responses across respiratory muscles. Please, expand or correct accordingly.

-Line 262-6. EMG amplitudes were quantified across muscles, animals, age groups and disease conditions. Quantification of EMG is exceedingly challenging and unlikely to yield the desired comparative results. Please, provide support for this methodological approach (e.g., reliability across muscles present bilaterally, consistency of M-wave amplitude, etc.).

-Line 275. Section 2.6. The importance of the segregation of these measures into four different cohorts is not immediately clear. It appears functional measurements were not performed across all muscles of interest or age groups. Please, justify accordingly.

-Lines 417-9. The cleidomastoid muscle was not examined in the multiplex cytokine assay. Please, justify.

-Lines 461 and 467. The listing of changes following chemostimulation incorrectly indicates "significant increase in ..." but in both age groups some of the measures decreased (e.g., inspiratory and expiratory times).

-Line 498, Figure 4 is discussed in section 3.2 prior to the presentation of figures 2 and 3. Please, consider changing the figure numbers to reflect the order of the presentation in the text.

-Line 523. Data on SpO₂ are not presented but an effect of hypoxic-hypercapnic exposure is expected. It seems of interest to know whether mdx mice were more likely to experience greater desaturation (either duration or depth).

-Figure 2, legend. The text does not clarify the nature of measurements shown in the shaded sections (presumably tracheal occlusion).

-Figure 3 and legend. The chemostimulation is inconsistently presented (the figure has 15% O₂/5% CO₂ but the text indicates 10%O₂/6% CO₂). Please, correct accordingly across the manuscript.

-Section 4 and Figures 5, 6, 7 (including legends). The use of the words obligatory and accessory in lieu of specifically stating the muscles tested is unnecessary and possibly confounding. Please, specify the muscle in the figure itself.

-Section 4. Statements in the manuscript regarding trends when findings approach 0.05 should be removed given the many comparisons being conducted and lack of statistical adjustment.

-Figures 5, 6, 7 (legends). Clarify if the representative tracings in each grouping are from the same animal for each group.

-Figure 11. The organization of this figure does not facilitate comparisons across age groups. The two age groups could be placed horizontally and the seven different muscles vertically, for instance.

-Figure 12 (and legend). The order in defining the abbreviations in the legend could match that of the figure to facilitate review.

-Figures 13 and 14 cannot be evaluated in their present form. In addition, comparisons between the 4- and 16-month-old groups (for both wildtype and mdx groups) are not facilitated by having the data in two separate figures. There are 17 cytokines, 4 groups and 6 muscles but if I am following correctly the main comparisons are for each cytokine across groups and muscles. If organized accordingly these comparisons could be presented in a single figure (as was done for figure 10).

DISCUSSION. A few comments deserve attention:

- This section starts by summarizing the findings. Each of the summary statements could be presented in the corresponding section in the results.

- Muscle fiber type composition varies across the muscles studied and is likely altered in mdx mice in early and late disease

stages. This aspect is not sufficiently considered. By clustering the data on the minimal Feret's diameter within each animal, within muscle variance in fiber dimensions are obscured. It seems of interest to identify whether disease stage increases this variance (e.g., in figure 11), not just the proportion of smaller fibers.

- Limitations related to the difficulty in interpreting quantitative EMG analyses are not presented.

- The impact of skeletal changes (e.g., onset of kyphosis) in response to altered muscle function and which could impair respiratory system mechanics is not considered.

Referee #3:

I would concur with reviewer 2's main comments; there is a fleeting mention of a 'mixed model' being used when missing data were present, but no details were provided and so it is unclear whether this refers to a linear mixed model (in which case specifying what the random and fixed effects were is needed), or a mixed 2-way ANOVA. Clearer justification and description of this aspect is needed, and it should be made clear in the results if/when this model has been used.

Can the authors confirm that all the factors analysed are indeed repeated measures? One would presume that 'genotype' is a between-subjects factor, and thus a two-way mixed ANOVA should be used.

For the repeated-measure ANOVAs, the authors should describe how the assumption of sphericity was assessed and handled. Where sphericity is an issue, the Greenhouse-Geisser correction should be applied when GG epsilon (ϵ) < 0.75, and the Huynh-Feldt correction was used otherwise. Failure to address violations of sphericity can lead to unreliable results, so these adjustments must be made explicit. The authors should also state how normality was assessed (e.g., Shapiro-Wilk test?).

Bonferroni correction is generally more conservative and widely recommended when controlling for family-wise error rates in within-subject designs.

P-values must be stated to three significant figures (not decimal places) even when 'no statistical significance' is being reported (i.e. for anything >0.001, please report to 3 significant figures, e.g. 0.00236 or 0.523, etc.). These should be stated in the main text, figures and their legends and tables. The only exception to this is if p is less than 0.001, in which case '<' is permitted.

Lastly, the quality of the figures should be improved, and p-values should be included directly on the figures in accordance with the journal's statistics guidelines (https://jp.msubmit.net/cgi-bin/main.plex?form_type=display_requirements#statistics). This will enhance the clarity and accessibility of the results.

END OF COMMENTS

REQUIRED ITEMS

- Author photo and profile. First or joint first authors are asked to provide a short biography (no more than 100 words for one author or 150 words in total for joint first authors) and a portrait photograph. These should be uploaded and clearly labelled together in a Word document with the revised version of the manuscript. See Information for Authors for further details.

RESPONSE: A photo and short bio are now included.

- Papers must comply with the Statistics Policy: https://jp.msubmit.net/cgi-bin/main.plex?form_type=display_requirements#statistics.

In summary:

- If $n \leq 30$, all data points must be plotted in the figure in a way that reveals their range and distribution. A bar graph with data points overlaid, a box and whisker plot or a violin plot (preferably with data points included) are acceptable formats.

- If $n > 30$, then the entire raw dataset must be made available either as supporting information, or hosted on a not-for-profit repository, e.g. FigShare, with access details provided in the manuscript.

- 'n' clearly defined (e.g. x cells from y slices in z animals) in the Methods. Authors should be mindful of pseudoreplication.

- All relevant 'n' values must be clearly stated in the main text, figures and tables.

- The most appropriate summary statistic (e.g. mean or median and standard deviation) must be used. Standard Error of the Mean (SEM) alone is not permitted.

- Exact p values must be stated. Authors must not use 'greater than' or 'less than'. Exact p values must be stated to three significant figures even when 'no statistical significance' is claimed.

RESPONSE: Noted, thank you.

- A Data Availability Statement is required for all papers reporting original data. This must be in the Additional Information section of the manuscript itself. It must have the paragraph heading 'Data Availability Statement'. All data supporting the results in the paper must be either: in the paper itself; uploaded as Supporting Information for Online Publication; or archived in an appropriate public repository. The statement needs to describe the availability or the absence of shared data. Authors must include in their statement: a link to the repository they have used, or a statement that it is available as Supporting Information; reference the data in the appropriate section(s) of their manuscript; and cite the data they have shared in the References section. Whenever possible, the scripts and other artefacts used to generate the analyses presented in the paper should also be publicly archived. If sharing data compromises ethical standards or legal requirements then authors are not expected to share it, but must note this in their statement. For more information, see our Statistics Policy.

RESPONSE: A data availability statement is now added.

- Please include an Abstract Figure file, as well as the Figure Legend text within the main article file. The Abstract Figure is a piece of artwork designed to give readers an immediate understanding of the research and should summarise the main conclusions. If possible, the image should be easily 'readable' from left to right or top to bottom. It should show the physiological relevance of the manuscript so readers can assess the importance and content of its findings. Abstract Figures should not merely recapitulate other figures in the manuscript. Please try to keep the diagram as simple as possible and without superfluous information that may distract from the main conclusion(s). Abstract Figures must be provided by authors no later than the revised manuscript stage and should be uploaded as a separate file during online submission labelled as File Type 'Abstract Figure'. Please also ensure that you include the figure legend in the main article file. All Abstract Figures should be created using BioRender. Authors should use The Journal's premium BioRender account to export high-resolution images. Details on how to use and access the premium account are included as part of this email.

RESPONSE: An abstract Figure and legend are now added.

EDITOR COMMENTS

Reviewing Editor:

Your work has been evaluated by two experts in the field. Both view the work as quite influential, and noted the careful and thoughtful manner in which the experiments were performed. While the overall critiques are minor, there are several points that require attention before further consideration can be made. Also, in the revision, please upload high-resolution figures so that the data can be scrutinized more closely.

Please also see 'Required Items' above.

RESPONSE: Thank you for this helpful summary and for the positive appraisal of our manuscript. Apologies for the poor quality of the jpeg images, which have distorted the original high quality figures. We have provided a pdf file of Figures during re-submission to facilitate review.

Senior Editor:

Comments for Authors to ensure the paper complies with the Statistics Policy (Required): see comments from Statistical Editor below (Referee #3).

RESPONSE: Thank you. We have responded to each point raised by referee 3.

Comments to the Author:

Thank you for submitting your research article to the Journal of Physiology for consideration. The article has been reviewed by experts in the field and found to be potentially acceptable for publication pending adequate revision to address all of the concerns raised. Please

address all comments from the external referees and the statistical and reviewing editors as well as addressing the list of requirements or publication in the Journal provided in this letter and the Journal's guidelines for rigor and reproducibility.

RESPONSE: Thank you. We have addressed all points raised.

REFEREE COMMENTS

Referee #1:

The manuscript by Slyne et al., centers on the muscle structure of the mdx mice during early and advanced stages of dystrophic disease progression. Emphasis is placed on the diaphragm muscles and remodeling that occurs in Duchenne muscular dystrophy (DMD) in these dystrophin-deficient mice. The authors identified significant muscle weakness and inspiration deficits that occurs at 16 months of age in mdx mice. By 20 months of age, the mdx mice have advanced diaphragm muscle weakness and inspiration deficits, thus relying on their "extra-diaphragmatic muscles". The authors concluded that the diaphragm and other skeletal muscle compensation occurs with the progressive muscle wasting and cytokine influx, with only the scalene and trapezius being spared.

Overall, this is an interesting and important study that seeks to clarify key DMD disease-relevant biomarkers and indicators within the mdx mouse strain. The experiments are designed logically, and are appropriately justified in power, number, and statistical analysis. My comments are mostly moderate and minor in scope as the manuscript is well-written and the experimental data is comprehensive in scope. If the authors can clarify certain aspects of the study, it would be a good addition to the DMD biology and biomarker fields.

RESPONSE: Thank you for your positive appraisal of our work and for highlighting the importance of studies in this area.

General Comments:

1. Abstract, line 43. This statement is confusing: More convenient models of rapid, progressive

muscular dystrophy are required to study end-stage disease". What do the authors mean

exactly by "convenient". This sentence is unclear and should be clarified.

RESPONSE: Thank you for pointing this out. We are referring to the logistical challenge of studying respiratory compromise in *mdx* mice which necessitates the study of animals >16 months of age. Therefore, our point is that more logistically convenient models are required to facilitate the study of respiratory compromise in DMD models. We prefer this term to 'better models' as it is often debatable as to whether one model is better than another. We have revised the text to state 'logistically convenient'.

2. The role of the heart, specifically dilated cardiomyopathy (DCM), that occurs towards the end stages of DMD. Did the authors notice an increased burden of DCM in their mice. Specifically, the increased cardiac burden would likely place a similar metabolic and inspiration burden on the *mdx* mice. Did the authors perform any ejection fraction measurements in the cohorts? Some discussion of the cardiac burden in late stage DMD as a factor of increased respiratory demand should be addressed in the discussion section.

RESPONSE: Thank you for raising this important point. We did not assess cardiac function in *mdx* mice at any age in our study. It is established that *mdx* mice do not show evidence of dilated cardiomyopathy as young adults but do so before 1 year of age with evidence of fibrosis in late-stage disease (Quinlan et al., 2004; PMID: 15336690). We acknowledge that cardiac burden is a factor relevant to our considerations of respiratory performance comparing 4- and 16-month-old *mdx* mice since cardiac burden has implications for respiratory control. Notably, however, ventilatory capacity was generally well maintained in 16-month-old *mdx* mice, again highlighting a remarkable compensation although it is generally regarded that the cardiac manifestation of dilated cardiomyopathy and diffuse chamber fibrosis in *mdx* mice is mild compared to human DMD. We have revised the text to draw focus to these points including the discussion in a new limitations section.

3. Minor comment. Figures 10A-N. Did the authors run statistical analysis on the relative frequency for their min. ferret diameter measurements?

RESPONSE: No, these plots are provided for illustrative purposes to demonstrate the left-shift in *mdx* muscle fibre size. In response to referee 2's comments we performed an analysis of coefficient of variation of muscle fibre size, and these data are now included in the revised manuscript (added to Fig. 10).

4. Figures 13 and 14. Can you clarify what the quantification or units of value are in the interleukin/cytokine values?

RESPONSE: The units are pg/mg of protein homogenate. This information was provided on the Y-axis of the graphs and in the methods section. Owing to the large number of panels in Figs 13 and 14 we have decided to show these data in Tables in the revised manuscript (Tables 5-10). We apologise for the inconvenience.

Referee #2:

This is a very comprehensive study of respiratory muscle adaptations in 4- and 16-month-old control and *mdx* mice. The work builds upon prior work by this team and provides an important, comprehensive evaluation of various respiratory muscles at early and late disease stages.

RESPONSE: Thank you for your positive appraisal of our work, recognising the importance of the work for the field.

RESULTS. There are several issues in the presentation of findings that need attention to improve the reader's ability to interpret the results.

-Selection of respiratory muscles studied. The rationale for the inclusion of some respiratory muscles in the various measurements in the study is not clearly presented because not all measures were conducted in all muscles.

RESPONSE: There was limited availability of 16-month-old mice and therefore whereas we performed structure and function analysis in 7 respiratory muscles at 4-months of age (studying up to 2 muscles per animal for functional analysis), we were restricted in the options at 16-months of age. In our experimental set-up for functional analysis, a maximum of 2 muscles per animal could be studied. We elected to study diaphragm as the major muscle of breathing and scalene given that it is an important accessory muscle of breathing per se and given that it showed normal forces at 4 months of age allowing us to examine our hypothesis that accessory muscle dysfunction (emergence and/or worsening) contributes to respiratory compromise in advanced disease. The text is revised to provide the rationale/justification for our design.

-Airway (tracheal) occlusion vs. obstruction. These various terms appear to be used interchangeably in the both the text and figures. Based on the methods, it appears that it was tracheal occlusion. Please, consider using only one term consistently across the manuscript.

RESPONSE: Thank you for pointing this out. We agree a consistent term is best to avoid confusion and we have chosen tracheal occlusion as the best descriptor. The text is revised throughout the revised manuscript.

-Histological analyses. There is no information provided on the number of fibers analyzed histologically for each animal. Please, correct in the methods and results sections (and/or figures 8-11).

RESPONSE: Thank you for this comment. We used a standard grid applying stereological principles to assess muscle. Total fibre counts per unit area varied between muscles (because of differences in fibre size) and between wild-type and *mdx* mice (because of disease). We have now incorporated into the Methods information on the average total number of fibres counted per region contributing to the computed group means for each muscle in each group. Different magnification was used for H&E (20X) and laminin (10X). Typically, several hundred fibres were analysed in each animal.

-Statistical analyses. For example, analyses of EMG amplitude were conducted using repeated measures two-way ANOVA, but in examining Figure 3 it is evident that for several muscles and measures (e.g. airway occlusion) the variance across groups is not equal. A mixed linear model may be more appropriate, as suggested was used for missing data. Please, justify. Note this issue applies to multiple other measurements, not just EMG.

RESPONSE: Thank you for raising this important issue. We performed a two-way mixed ANOVA (within and between factors) for relevant data sets with no missing values. When values within a set were missing for technical reasons (which violates an assumption for

ANOVA), the software (Prism) ran a mixed-effects model with genotype as the fixed factor and each mouse as the random factor (as the inference was for all mice and not just the mice studied). We acknowledge that variance was not equal and for this reason we did not assume sphericity in the data sets. Therefore, the Greenhouse Geisser correction was routinely applied to all data. We apologise for not stating this in the original manuscript. Please see additional responses to the statistics editor below. We have revised the manuscript to highlight the approach undertaken.

In addition, there is limited presentation of the main effects (shown only in the figures), yet the text expands on many post hoc analyses that are not adjusted for the many comparisons. This issue not only makes the text difficult to follow but also confounds interpretation. For instance, the use of 4 decimal points of precision in presenting the p values of these post hoc analyses is unwarranted.

RESPONSE: In the original article our primary interest was in the pair-wise comparisons (post hoc analyses) as these provide for commentary on genotype similarities or differences in each parameter (e.g., EMG) in the various behaviours of interest across the ventilatory and non-ventilatory range (baseline, gas challenge, vagotomy, gas challenge, occlusion). Concerning the main effects: Time was significant as expected *a priori* since we are assessing the effects of stimuli that are known to increase respiratory drive. In general, there were striking genotype effects. The pair-wise comparisons allow us to draw focus to the behaviour wherein the genotype effect is strongest. Please note that our post hoc tests did account for multiple comparisons and the P values reported are adjusted P values. We should have made this clear in the original submission and we are grateful to you for raising this point. The journal policy is to present P values to 3 significant numbers and so at least 5 decimal places are required on occasion.

However, we acknowledge and agree with the referee that the main effects are important, and in the context of our study especially genotype effects and therefore we have included commentary on the latter in the text section of the results, elaborating further on post hoc comparisons when these were significant.

-Figure quality. Some of the figures (both PDF and jpg formats) are of low quality. For example, figure 3 is blurry and figures 13 and 14 are not legible at all. In this regard the axis labels can be reviewed to ensure that only the necessary information is included, allowing for the use of larger fonts (e.g., in figure 3, the y-axis labels repeat the figure titles at the left of the figure. The H&E stains shown in figures 8 and 9 are too blurry to see centrally located nuclei or inflammatory infiltration.

RESPONSE: We sincerely apologise for this inconvenience. It appears there was an issue with the quality of Figs 13 and 14 following file conversion during submission, which is not evident in the original files that we uploaded. Nevertheless, owing to the large number of panels in Figs 13 and 14 we have decided to show these data in Tables in the revised manuscript. We have edited axes labels to allow us to increase font size. The original histological images are high quality. We have provided a pdf of the figures during re-submission to facilitate review.

Specific comments.

-Intercostal muscle. In some sections of the manuscript, it is specified that the external intercostal muscle was studied, but many other sections do not clarify this aspect. Please, correct accordingly.

RESPONSE: In EMG recordings, we are confident that we recorded external intercostal EMG based on placement of the electrode. For histology, although the tissue was arranged to facilitate identification of external intercostal muscle separate to internal intercostal, there were occasions where this could not explicitly be determined with absolute confidence. Therefore, on balance, we have elected to label the Figure 'Intercostal'. Given the limited tissue in mouse and the requirement for a minimum amount of protein for the cytokine assays, intercostal bundles were used and therefore we cannot be certain that it was exclusively external intercostal. Therefore, the figure is labelled 'Intercostal'. These limitations are now acknowledged in the revised manuscript.

-Line 222. Only oesophageal pressures were measured when it is common when studying respiratory mechanics to measure both oesophageal and gastric pressure to determine transdiaphragmatic pressures. Please, justify.

RESPONSE: Transdiaphragmatic pressure (Pdi) is the gold standard assessment of diaphragm function *in situ*. However, our research question was broader than assessment of diaphragm performance in *mdx* mice compared to wild-type mice. Rather, we were curious to determine respiratory capacity per se, which relates to the composite action of multiple inspiratory muscles. Therefore, we assessed peak inspiratory pressure measuring oesophageal pressure generated by the concerted action of all inspiratory muscles. Given that diaphragm force is severely curtailed in *mdx* mice, it is given that Pdi would be lower in *mdx* mice, even at 4-months of age (and younger). Indeed, one could easily be misled had this measure been taken in assuming that peak inspiratory performance is impaired in *mdx* mice at 4-months of age, given that Pdi is often taken as a proxy for peak performance (whereas peak performance demonstrably is not impaired in *mdx* at 4 months because of compensation by extra-diaphragmatic muscles). Moreover, to assess Pdi in rodents, it seems that abdominal banding is required, and diaphragm performance is assessed under isometric conditions. Our focus was on the physiological capacity of the diaphragm and accessory muscles to generate peak inspiratory pressure as an estimate of peak respiratory performance. Therefore, the assessment of inspiratory pressure across a range of behaviours was the most appropriate measure. This argument was previously presented in a letter to the editor that we published (Burns et al. 2019 PMID: 31368167), in response to a letter to the editor by Khurram, 2019 (PMID: 31131877) relating to our first description of compensation of peak inspiratory pressure in young *mdx* mice (Burns et al., 2019; PMID: 30570134).

-Line 255. Sustained, terminal tracheal occlusion was used to elicit higher force respiratory muscle activation, but there is no support provided for the statement that this manoeuvre elicits "near maximal oesophageal pressure and respiratory muscle EMG activities". It is likely that this statement is an oversimplification of the complex, integrated responses across respiratory muscles. Please, expand or correct accordingly.

RESPONSE: We have determined that peak inspiratory pressures and respiratory EMG activities coincide with the nadir pressure achieved during sustained tracheal occlusion, held until task failure, that is the inability to sustain peak inspiratory pressures [text revised to include this definition]. We do not see greater values for inspiratory pressure or EMG across

any other behaviour in the animals. Therefore, our assessment is of the maximum provoked values that can be achieved in the experimental preparation. Because animals are anaesthetised, we acknowledge that there is likely some attenuation of the maximum peak values that might be achieved in the conscious state.

We are unclear why the referee sees this as an oversimplification of the complex integrated respiratory responses across respiratory muscles. The tracheal occlusion provides a most powerful drive to breathe, which manifests 'maximal' motor drive to the respiratory muscles which we record (EMGs), whose collective action result in the mechanical changes that culminate in a peak inspiratory pressure, which we record (P_{insp}). The physiology is complex, but the experimental approach is rather straight-forward as the measures we made, by design, provide an index of the efficacy of the system to achieve peak performance.

-Line 262-6. EMG amplitudes were quantified across muscles, animals, age groups and disease conditions. Quantification of EMG is exceedingly challenging and unlikely to yield the desired comparative results. Please, provide support for this methodological approach (e.g., reliability across muscles present bilaterally, consistency of M-wave amplitude, etc.).

RESPONSE: Thank you for this important comment. It can be difficult to reliably compare EMG signals between muscles and animals. To facilitate comparisons, we adopted a systematic approach to EMG recording and analysis. Electrode placement was standardized *a priori* as described in the Methods consistent with our previous approach (Burns et al., 2019; O'Halloran et al., 2023). All signals were processed in the same fashion (amplification, filtering, integration). Integrated EMG data are presented with absolute units to allow for faithful comparisons between muscles and between groups, i.e., data are not obscured by presenting relative changes as percentage changes or normalized to a reference value. The principal conclusion we have drawn from EMG analyses is that there is lower activation in *mdx* mice particularly during high demand behaviours. The intramuscular electrode measures multi-unit activity, which is phasic with inspiration. The altered EMG signal in *mdx* mice reflects decreased complexity in the composite signal (lower power) and lower amplitude motor units. We have quantified the composite signal using the standard approach of measuring the amplitude of the rectified integrated signal (the outcome is the same if area under the curve of the integral is assessed). Loss of large amplitude units is discernible from our recordings sampled at 20kHz. However, spike sorting of motor units would be required to fully characterize these changes in *mdx* muscles. We acknowledge that assessments were made under general anaesthesia, which depresses motor drive and hence the complexity of EMG signals.

We have added a limitations section to the discussion and have included the above discussion.

-Line 275. Section 2.6. The importance of the segregation of these measures into four different cohorts is not immediately clear. It appears functional measurements were not performed across all muscles of interest or age groups. Please, justify accordingly.

RESPONSE: Apologies for any confusion caused. Our experimental set-up allowed for the assessment of 2 muscles per animal. We did not wish to measure additional muscles beyond these in a given animal, which would have necessitated a delay in assessment after killing. Therefore, to assess the suite of muscles studied, it was necessary to examine different muscles in different animals. We have described earlier in the rebuttal the limited availability of 16-month-old mice and the rationale for choosing diaphragm and scalene at 16-months. This is now clarified in the revised text.

-Lines 417-9. The cleidomastoid muscle was not examined in the multiplex cytokine assay. Please, justify.

RESPONSE: This related to budget. The assays are expensive and we had budget for 6 not 7 plates. We reasoned that sternomastoid and cleidomastoid would be similar based upon functional data collected in the study and on the basis that these are neighbour complementary muscles, considered one muscle in humans.

-Lines 461 and 467. The listing of changes following chemostimulation incorrectly indicates "significant increase in ..." but in both age groups some of the measures decreased (e.g., inspiratory and expiratory times).

RESPONSE: Thank you for spotting this error. The text is now revised.

-Line 498, Figure 4 is discussed in section 3.2 prior to the presentation of figures 2 and 3. Please, consider changing the figure numbers to reflect the order of the presentation in the text.

RESPONSE: Thank you for spotting this error. The figure numbers have been changed.

-Line 523. Data on SpO₂ are not presented but an effect of hypoxic-hypercapnic exposure is expected. It seems of interest to know whether mdx mice were more likely to experience greater desaturation (either duration or depth).

RESPONSE: Thank you for raising this. SpO₂ profiles during challenges generally appeared quite similar, but statistical analyses revealed a genotype effect, with SpO₂ desaturation greater in wild-type compared to *mdx* at 4-months of age, but greater in *mdx* compared to wild-type at 16-months of age. Post hoc analysis revealed a group difference in 4-month-old mice (vagi intact; $P = 0.000359$; Bonferroni's multiple comparisons *post hoc*). The text has been revised to describe this outcome.

-Figure 2, legend. The text does not clarify the nature of measurements shown in the shaded sections (presumably tracheal occlusion).

RESPONSE: Yes, this is tracheal occlusion and is now included in the legend. Thank you.

-Figure 3 and legend. The chemostimulation is inconsistently presented (the figure has 15% O₂/5% CO₂ but the text indicates 10%O₂/6% CO₂). Please, correct accordingly across the manuscript.

RESPONSE: Apologies for the confusion. The chemostimulatory challenge in conscious mice was elicited with 10%O₂/6% CO₂, whereas the chemostimulatory challenge in anaesthetized mice was elicited with 15%O₂/6% CO₂, given that baseline breathing was maintained with 60% O₂ to maintain adequate oxygenation under anaesthesia. This was reflected in the text, but the Figures erroneously displayed 15% O₂/5% CO₂ and this has now been corrected. Thank you.

-Section 4 and Figures 5, 6, 7 (including legends). The use of the words obligatory and accessory in lieu of specifically stating the muscles tested is unnecessary and possibly confounding. Please, specify the muscle in the figure itself.

RESPONSE: Obligatory and accessory were intended as titles for each panel, however we recognise the referee's point. These are now removed from the figure and the specific muscle is named instead.

-Section 4. Statements in the manuscript regarding trends when findings approach 0.05 should be removed given the many comparisons being conducted and lack of statistical adjustment.

RESPONSE: The values reported are adjusted P values, to account for multiple comparisons, but we fully accept the point and all comments relating to trends are removed and statements are revised.

-Figures 5, 6, 7 (legends). Clarify if the representative tracings in each grouping are from the same animal for each group.

RESPONSE: The twitch and tetanic tracings were not from the same animals, as representative tracings reflective of the mean group values were selected. However, the original tracings in the force-frequency curves are sequential contractions in respective muscle preparations. This detail is now included in the legends.

-Figure 11. The organization of this figure does not facilitate comparisons across age groups. The two age groups could be placed horizontally and the seven different muscles vertically, for instance.

RESPONSE: Thank you for this comment. We have revised the layout of the figure to facilitate comparisons across age groups.

-Figure 12 (and legend). The order in defining the abbreviations in the legend could match that of the figure to facilitate review.

RESPONSE: Thank you for pointing this out. Corrected in the revised manuscript.

-Figures 13 and 14 cannot be evaluated in their present form. In addition, comparisons between the 4- and 16-month-old groups (for both wildtype and mdx groups) are not facilitated by having the data in two separate figures. There are 17 cytokines, 4 groups and 6 muscles but if I am following correctly the main comparisons are for each cytokine across groups and muscles. If organized accordingly these comparisons could be presented in a single figure (as was done for figure 10).

RESPONSE: Thank you for these comments. Given the complexity of the figure given the volume of data, and in consideration of the referee's points, we have presented the data in Tables 5-10 in the revised manuscript. The primary objective was to compare wild-type and *mdx* for a given analyte at each age. However, to facilitate comparisons between 4- and 16-month-old mice, these data are now juxtaposed in the revised manuscript for each muscle.

DISCUSSION. A few comments deserve attention:

- This section starts by summarizing the findings. Each of the summary statements could be presented in the corresponding section in the results.

RESPONSE: Thank you for this suggestion. We prefer to use conservative headings in the Results section leaving consideration of the findings to the Discussion.

- Muscle fiber type composition varies across the muscles studied and is likely altered in mdx mice in early and late disease stages. This aspect is not sufficiently considered. By clustering the data on the minimal Feret's diameter within each animal, within muscle variance in fiber dimensions are obscured. It seems of interest to identify whether disease stage increases this variance (e.g., in figure 11), not just the proportion of smaller fibers.

RESPONSE: Thank you for these comments. We have previously described fibre-type differences in *mdx* mice compared to wild-type (Burns et al., 2018; PMID: 30160301). We recognise the referee's concern. To address this, we calculated the coefficient of variation of fibre size and these data are now presented in Figure 10. There is greater variability in *mdx* muscle fibre size. Variability in diaphragm muscle fibre size is consistent between early and advanced disease. The key difference in the diaphragm comparing 4mo and 16mo *mdx* is the profound further increase in fibrosis and consequently fewer fibres.

- Limitations related to the difficulty in interpreting quantitative EMG analyses are not presented.

RESPONSE: Our approach to standardise assessment is described above. Nevertheless, we acknowledge that there are limitations to the approach, and this is now included in the revised manuscript.

- The impact of skeletal changes (e.g., onset of kyphosis) in response to altered muscle function and which could impair respiratory system mechanics is not considered.

RESPONSE: Kyphosis is a feature of advanced DMD, further contributing to respiratory involvement in neuromuscular disease. Kyphosis was not present in early disease, but it was evident in some *mdx* mice at 16-months of age, but it was not systematically scored in our study. This point is now added to the revised manuscript.

Referee #3:

I would concur with reviewer 2's main comments; there is a fleeting mention of a 'mixed model' being used when missing data were present, but no details were provided and so it is unclear whether this refers to a linear mixed model (in which case specifying what the random and fixed effects were is needed), or a mixed 2-way ANOVA. Clearer justification and description of this aspect is needed, and it should be made clear in the results if/when this model has been used.

Can the authors confirm that all the factors analysed are indeed repeated measures? One would presume that 'genotype' is a between-subjects factor, and thus a two-way mixed ANOVA should be used.

For the repeated-measure ANOVAs, the authors should describe how the assumption of sphericity was assessed and handled. Where sphericity is an issue, the Greenhouse-Geisser correction should be applied when GG epsilon (ϵ) < 0.75, and the Huynh-Feldt

correction was used otherwise. Failure to address violations of sphericity can lead to unreliable results, so these adjustments must be made explicit. The authors should also state how normality was assessed (e.g., Shapiro-Wilk test?).

RESPONSE: We performed a two-way mixed ANOVA (within and between factors). However, we arranged our tables in the software package with each row containing data from each mouse across the experimental protocol (time). This is why we originally stated repeated measures as the same parameter can be tracked across each of the behaviours that raised respiratory drive from baseline to occlusion. A two-way mixed ANOVA was used for relevant data sets with no missing values. When occasional values within a set were missing for technical reasons (which violates an assumption for ANOVA), the software (Prism) ran a mixed-effects model with genotype as the fixed factor and each mouse as the random factor (as the inference was for all mice and not just the mice studied).

Normality was assessed for all data sets using the Kolmogorov-Smirnov test and visual inspection of Q-Q plots. Considering your comments, for data sets not normally distributed we decided to \log_{10} transform the data to better approximate normality before running statistical tests. The original data sets appear in graphs for consistency between muscles and groups and for consistency with previously published data from our research group (O'Halloran et al., 2023).

We did not assume sphericity and so the Greenhouse Geisser correction was applied to all data sets as recommended by Prism.

We have revised the Methods and legends to clarify these points. We have removed 'repeated measures' from the description of the two-way ANOVA.

Bonferroni correction is generally more conservative and widely recommended when controlling for family-wise error rates in within-subject designs.

RESPONSE: The software Prism recommended Sidak for post hoc comparisons, but we have performed Bonferroni as suggested by the editor. Revisions have been made to the text, figures, tables and legends.

Genotype (EIC EMG during occlusion) and post-hoc effects in EMG data (EIC, sternomastoid, cleidomastoid and scalene during occlusion) are now not significant in 4-month-old mdx. In essence, this strengthens the argument previously made that progressive impairment in extra-diaphragmatic muscles contributes to the emergence of respiratory system morbidity (decrease in peak inspiratory pressure) in advanced disease.

P-values must be stated to three significant figures (not decimal places) even when 'no statistical significance' is being reported (i.e. for anything >0.001 , please report to 3 significant figures, e.g. 0.00236 or 0.523, etc.). These should be stated in the main text, figures and their legends and tables. The only exception to this is if p is less than 0.001, in which case '<' is permitted.

RESPONSE: Apologies, we have now reported all P values to three significant figures.

Lastly, the quality of the figures should be improved, and p-values should be included directly on the figures in accordance with the journal's statistics guidelines (https://jp.msubmit.net/cgi-bin/main.plex?form_type=display_requirements#statistics). This will enhance the clarity and accessibility of the results.

RESPONSE: We apologise for the quality of the Figures, which appears to be an issue in the conversion of the high quality original images to tiff format. A pdf of high-quality Figures is provided to accompany files re-submitted. We have revised the manuscript to incorporate the cytokine data into Tables. P values are clearly displayed in Figures and Tables. The legends contain the necessary details.

END OF COMMENTS

Dear Ken,

Re: JP-RP-2025-288709R1 "**Obligatory and accessory respiratory muscle structure, function and control in early and advanced disease in the *mdx* mouse model of DMD**" by Aoife D Slyne, David P Burns, Karina Woeller, Amandine May, Roisin Dowd, Sarah E Drummond, Grzegorz Jasioneck, and Ken D O'Halloran

Thank you for submitting your manuscript to The Journal of Physiology. It has been assessed by a Reviewing Editor and by 3 expert referees and we are pleased to tell you that it is acceptable for publication following satisfactory revision.

REVISION CHECKLIST:

We look forward to receiving your revised submission.

Best wishes,

Harold Schultz
Senior Editor
The Journal of Physiology

REQUIRED ITEMS

- Papers must comply with the Statistics Policy: https://jp.msubmit.net/cgi-bin/main.plex?form_type=display_requirements#statistics.

In summary:

- If n {less than or equal to} 30, all data points must be plotted in the figure in a way that reveals their range and distribution. A bar graph with data points overlaid, a box and whisker plot or a violin plot (preferably with data points included) are acceptable formats.
- If $n > 30$, then the entire raw dataset must be made available either as supporting information, or hosted on a not-for-profit repository, e.g. FigShare, with access details provided in the manuscript.
- 'n' clearly defined (e.g. x cells from y slices in z animals) in the Methods. Authors should be mindful of pseudoreplication.
- All relevant 'n' values must be clearly stated in the main text, figures and tables.
- The most appropriate summary statistic (e.g. mean or median and standard deviation) must be used. Standard Error of the Mean (SEM) alone is not permitted.
- Exact p values must be stated. Authors must not use 'greater than' or 'less than'. Exact p values must be stated to three significant figures even when 'no statistical significance' is claimed.

EDITOR COMMENTS

Reviewing Editor:

Comments for Authors to ensure the paper complies with the Statistics Policy (Required):

Reviewer #2 and the statistics consulting editor had some concerns that need to be addressed before a final decision can be made.

Comments to the Author:

Thank you for the thoughtful responses to the reviewers. Overall, the revision is well-received, but there are some remaining concerns with respect to statistics and how the data are presented. We consulted with our in-house statistics editor how has some remaining requests, as does reviewer 2. Please respond to these requests. Thank you.

Senior Editor:

Comments for Authors to ensure the paper complies with the Statistics Policy (Required):

The manuscript complies with statistical reporting. However, we suggest improving the format for reporting statistical testing (see comment from the statistical editor). Report P values to three significant digits. (e.g., $P = 0.002$ rather than $P = 0.00235$). If the level is below 0.001, then report the P value as $P < 0.001$. In addition, the font size of p-values in most of the figures is below a readable resolution.

Comments to the Author:

The revised version of your research article has been considered for acceptance by The Journal of Physiology. Although the revision is potentially acceptable for publication, some concerns remain from referee #2, and the statistical, reviewing, and senior editors. Please address all comments and ensure the list of requirements or publication in the journal are met.

Comments from Senior Editor:

1. The resolution of text in some of the figures [e.g., Abstract Figure (the text in graphs on the right) and Figs. 2 and 4 (X-axis labels)] are not sufficient to be clear when enlarged.

In addition, the font size of the X-axis labels in Figs. 2 and 4 likely will not meet the Journal's standards for readability. We suggest placing the labels on a diagonal as in Fig. 10, to allow a larger font size.

The issue with font size and readability also applies to the P values reported in Figs. 2,4, and 10.

2. Although we commend the authors for accuracy, the p-values are reported to 5 significant digits (e.g., $P = 0.00235$) in most cases in text, tables, and figures. TJP requires p-values to be reported only to 3 significant digits ($P = 0.002$). If the level is below 0.001, report the P value as $P < 0.001$. This will save space for P values in the figures and allow the font size to increase. It will also enhance comprehension of the numbers in text, tables and figures.

We recommend that all figures be checked to ensure font size for all text meets the publication requirements for figures.

3. As a suggestion, the interactive statistics in the figures could be placed in the legend to reduce clutter in the figure and circumvent the problem with the very small font size.

4. Asterisks are not needed in the figures because the actual p-values are reported. The asterisks serve only to confuse the reader for their significance.

REFeree COMMENTS

Referee #1:

The authors have made substantial improvement to the resubmitted manuscript, namely the clarifications on some of the figure legends and statistical methodology. The manuscript would be of keen interest to the DMD disease modeling field and especial in the context of pulmonary dysfunction that occurs in animal models (specifically mdx strains). It does provide significant reporting on the pulmonary muscle function and cytokine release. The study is original and builds off of previous work. The design and animal size justifications are appropriate. The validity for the conclusions drawn is also appropriate. I have no additional concerns.

Referee #2:

The reviewers addressed some of the comments in the previous review with outstanding issues remaining.

Statistical analyses presented primarily reflect genotype comparisons which were conducted differently depending on the need for correction for some data sets and not for others. However, this approach makes the presentation challenging given the need to provide specific notation for the exact statistical tool used each time. It seems that a conservative approach (e.g., one using the least number of assumptions) may allow the use of a much more limited number of statistical tools, simplifying the presentation (e.g., always use Welch correction for the t-test analyses).

Section 2.7. Lines 299 and 347. Specify whether and how muscles were fixed (e.g., at resting or at optimal length). This point is important in comparing the number of fibers examined across muscles and differences in fiber number despite using a fixed frame (400x400 or 600x600 microns), as well as Feret diameters.

Section 2.7. Lines 396 & 399. Please provide lot # for each antibody.

Table 4 does not report mechanical power or work for the scalene muscle.

For the most part, morphological analyses are presented comparing genotypes for each age group, muscle and measurement without inclusion of main effects either in the text or figure 10 (e.g., note that main effects are in other figures). The data supporting statements about differences across ages are not presented (e.g., lines 727-9, 744-5, 759-61).

Line 759. Statement likely refers to a comparison to 16-months of age (not 4) given that the corresponding figure is 10H.

Figure 12. There is no data presented in the figure (data are in Tables 5-10) for IL-1beta, yet the legends indicate that it is reported within the figure. The fold-changes can be confusing given the very low levels at baseline for many of the cytokines and the fact that comparisons between the 4- and 16-month-old groups are obviated. Since the raw data is already in Tables 5-10, it is not clear that Fig. 12 provides any additional information and indeed may confound interpretation.

The cytokine analyses presented (including the figures) reflect comparisons across genotypes within each age group. The statistics for comparisons across age groups are not presented, thus corresponding statements (lines 815, 829-30) lack support.

Many of the figures are still of low quality in every uploaded version, precluding reading the very small font text used in most of them.

Referee #3 (statistics review):

The original comments provided in my initial review have been well addressed. My only remaining suggestion is to more clearly outline which effects were paired (within-subjects) versus independent (between-subjects) in the two-way mixed ANOVA description. The current text, "All data were statistically compared by two-way mixed analysis of variance (ANOVA) (gas × genotype (plethysmography); time × genotype (EMG); stimulus frequency × genotype (muscle function)) with Bonferroni post hoc test," could be improved for clarity by adopting a revision such as: "All data were statistically compared by two-way mixed analysis of variance (ANOVA) with genotype as the between-subjects factor and gas (plethysmography),

time (EMG), or stimulus frequency (muscle function) as the within-subjects factors, followed by Bonferroni post hoc tests."

END OF COMMENTS

EDITOR COMMENTS

Reviewing Editor:

Comments for Authors to ensure the paper complies with the Statistics Policy (Required):
Reviewer #2 and the statistics consulting editor had some concerns that need to be addressed before a final decision can be made.

Comments to the Author:

Thank you for the thoughtful responses to the reviewers. Overall, the revision is well-received, but there are some remaining concerns with respect to statistics and how the data are presented. We consulted with our in-house statistics editor who has some remaining requests, as does reviewer 2. Please respond to these requests. Thank you.

RESPONSE: Thank you for your positive appraisal. We have addressed the outstanding issues.

Senior Editor:

Comments for Authors to ensure the paper complies with the Statistics Policy (Required):
The manuscript complies with statistical reporting. However, we suggest improving the format for reporting statistical testing (see comment from the statistical editor). Report P values to three significant digits. (e.g., $P = 0.002$ rather than $P = 0.00235$). If the level is below 0.001, then report the P value as $P < 0.001$. In addition, the font size of p-values in most of the figures is below a readable resolution.

RESPONSE: The statistics policy states that p values should be reported to 3 significant digits (not decimal places). There is a contradiction in what is being advised comparing the comments of the statistics editor and the senior editor. This arises over interpretation of what constitutes a significant digit, specifically whether non-zero digits are significant. The statistical policy cited by the statistics editor gave the example of a requirement for $P = 0.00236$ as an example, not $P = 0.002$. This interprets zeros after the decimal place as non-significant. The SE advises restriction to $P = 0.002$, which interprets the zeros as significant.

The SE advises that reporting to 3 decimal places is adequate, and this helps our Figures given how many P values are reported. All P-values are now reported to 3 decimal places. The font size of P-values in all graphs has been increased. We have also revised the graphs to increase their size by expanding Fig 4 to two panels and likewise with Fig 10.

Comments to the Author:

The revised version of your research article has been considered for acceptance by The Journal of Physiology. Although the revision is potentially acceptable for publication, some concerns remain from referee #2, and the statistical, reviewing, and senior editors. Please address all comments and ensure the list of requirements or publication in the journal are met.

RESPONSE: Thank you. We have responded to all comments.

Comments from Senior Editor:

1. The resolution of text in some of the figures [e.g., Abstract Figure (the text in graphs on the right) and Figs. 2 and 4 (X-axis labels)] are not sufficient to be clear when enlarged.

In addition, the font size of the X-axis labels in Figs. 2 and 4 likely will not meet the Journal's standards for readability. We suggest placing the labels on a diagonal as in Fig. 10, to allow a larger font size.

The issue with font size and readability also applies to the P values reported in Figs. 2,4, and 10.

RESPONSE: We have revised the graphs and increased the font size of labels and P values. We have improved the resolution of the graphs.

2. Although we commend the authors for accuracy, the p-values are reported to 5 significant digits (e.g., $P = 0.00235$) in most cases in text, tables, and figures. TJP requires p-values to be reported only to 3 significant digits ($P = 0.002$). If the level is below 0.001, report the P value as $P < 0.001$. This will save space for P values in the figures and allow the font size to increase. It will also enhance comprehension of the numbers in text, tables and figures.

We recommend that all figures be checked to ensure font size for all text meets the publication requirements for figures.

RESPONSE: The reporting to 3 significant digits was in line with the advice from the statistics editor. We have reverted to 3 significant digits as 3 decimal places as advised. Thank you.

3. As a suggestion, the interactive statistics in the figures could be placed in the legend to reduce clutter in the figure and circumvent the problem with the very small font size.

RESPONSE: We have revised the Figures.

4. Asterisks are not needed in the figures because the actual p-values are reported. The asterisks serve only to confuse the reader for their significance.

RESPONSE: We were encouraged to add asterisks to the graphs in our JP 2023 paper in addition to P values to add further clarity by highlighting the significant differences, given that the convention is still commonly used. Inclusion of P values for all comparisons can obscure significant changes. We have removed the asterisks as requested, but significant findings are highlighted by P values in **bold.**

REFEREE COMMENTS

Referee #1:

The authors have made substantial improvement to the resubmitted manuscript, namely the clarifications on some of the figure legends and statistical methodology. The manuscript would be of keen interest to the DMD disease modeling field and especial in the context of pulmonary dysfunction that occurs in animal models (specifically mdx strains). It does provide significant reporting on the pulmonary muscle function and cytokine release. The study is original and builds off of previous work. The design and animal size justifications are appropriate. The validity for the conclusions drawn is also appropriate. I have no additional concerns.

RESPONSE: Thank you for your positive appraisal of the manuscript and for your helpful comments, which are much appreciated.

Referee #2:

The reviewers addressed some of the comments in the previous review with outstanding issues remaining.

RESPONSE: Thank you for the series of new comments and suggestions that follow. We appreciate your careful critique of our manuscript and your helpful comments.

Statistical analyses presented primarily reflect genotype comparisons which were conducted differently depending on the need for correction for some data sets and not for others. However, this approach makes the presentation challenging given the need to provide specific notation for the exact statistical tool used each time. It seems that a conservative approach (e.g., one using the least number of assumptions) may allow the use of a much more limited number of statistical tools, simplifying the presentation (e.g., always use Welch correction for the t-test analyses).

RESPONSE: Although we recognize that this would refine the text section of the Results, we nevertheless prefer to use the test most appropriate for the data. The Welch correction is applied to data sets with unequal variances.

Section 2.7. Lines 299 and 347. Specify whether and how muscles were fixed (e.g., at resting or at optimal length). This point is important in comparing the number of fibers examined across muscles and differences in fiber number despite using a fixed frame (400x400 or 600x600 microns), as well as Feret diameters.

RESPONSE: Separate muscles or muscle bundles were used for histology and therefore optimum length was unknown. Muscles were fixed at resting length in a manner described in the text. This additional detail has been added to the revised text.

Section 2.7. Lines 396 & 399. Please provide lot # for each antibody.

RESPONSE: Primary and secondary antibody lot numbers have been added to the revised text.

Table 4 does not report mechanical power or work for the scalene muscle.

RESPONSE: In the parasternal and accessory muscles, the results for shortening at 50% load were inconsistent and we had no confidence in the data and discontinued these measurements. We have revised the text to highlight that work and power measures for these muscles were excluded.

For the most part, morphological analyses are presented comparing genotypes for each age group, muscle and measurement without inclusion of main effects either in the text or figure 10 (e.g., note that main effects are in other figures). The data supporting statements about differences across ages are not presented (e.g., lines 727-9, 744-5, 759-61).

RESPONSE: These data were compared with Student's *t* tests or Mann-Whitney tests. Supporting details on statistical comparisons for the difference between 4-month-old and 16-month-old *mdx* for centralized myonuclei, area of inflammation and area of collagen are now included in the revised text. Thank you for pointing this out.

Line 759. Statement likely refers to a comparison to 16-months of age (not 4) given that the corresponding figure is 10H.

RESPONSE: Thank you for spotting this error. The text is now revised.

Figure 12. There is no data presented in the figure (data are in Tables 5-10) for IL-1beta, yet the legends indicate that it is reported within the figure. The fold-changes can be confusing given the very low levels at baseline for many of the cytokines and the fact that comparisons between the 4- and 16-month-old groups are obviated. Since the raw data is already in Tables 5-10, it is not clear that Fig. 12 provides any additional information and indeed may confound interpretation.

RESPONSE: Thank you, corrected in the revised manuscript. We appreciate the referee's comment but nevertheless suggest that the Figure offers a helpful summary of the main findings, with quantitative data available to the reader in the Tables. We prefer to keep the Figure, which captures the main findings.

The cytokine analyses presented (including the figures) reflect comparisons across genotypes within each age group. The statistics for comparisons across age groups are not presented, thus corresponding statements (lines 815, 829-30) lack support.

RESPONSE: Since there are so many observations that would require a summary of the statistical support for the statements, these two sentences have been deleted.

Many of the figures are still of low quality in every uploaded version, precluding reading the very small font text used in most of them.

RESPONSE: We have revised the Figures and increased the font size to improve readability. We have improved the resolution of the Figures. We apologise for the inconvenience.

Referee #3 (statistics review):

The original comments provided in my initial review have been well addressed. My only remaining suggestion is to more clearly outline which effects were paired (within-subjects) versus independent (between-subjects) in the two-way mixed ANOVA description. The current text, "All data were statistically compared by two-way mixed analysis of variance (ANOVA) (gas × genotype (plethysmography); time × genotype (EMG); stimulus frequency × genotype (muscle function)) with Bonferroni post hoc test," could be improved for clarity by adopting a revision such as: "All data were statistically compared by two-way mixed analysis of variance (ANOVA) with genotype as the between-subjects factor and gas (plethysmography), time (EMG), or stimulus frequency (muscle function) as the within-subjects factors, followed by Bonferroni post hoc tests."

RESPONSE: Thank you for this suggestion. The text is revised as suggested. We appreciate your guidance in reviewing our manuscript.

END OF COMMENTS

Dear Ken,

Re: JP-RP-2025-288709R2 "**Obligatory and accessory respiratory muscle structure, function and control in early and advanced disease in the *mdx* mouse model of DMD**" by Aoife D Slyne, David P Burns, Karina Woeller, Amandine May, Roisin Dowd, Sarah E Drummond, Grzegorz Jasioneck, and Ken D O'Halloran

We are pleased to tell you that your paper has been accepted for publication in The Journal of Physiology.

Best wishes,

Harold Schultz
Senior Editor
The Journal of Physiology

If you would like to receive our 'Research Roundup', a monthly newsletter highlighting the cutting-edge research published in The Physiological Society's family of journals (The Journal of Physiology, Experimental Physiology, Physiological Reports, The Journal of Nutritional Physiology and The Journal of Precision Medicine: Health and Disease), please click this link, fill in your name and email address and select 'Research Roundup':

<https://www.physoc.org/journals-and-media/membernews>

- You can help your research get the attention it deserves! Check out Wiley's free Promotion Guide for best-practice recommendations for promoting your work at: www.wileyauthors.com/eoo/guide. You can learn more about Wiley Editing Services which offers professional video, design, and writing services to create shareable video abstracts, infographics, conference posters, lay summaries, and research news stories for your research at: www.wileyauthors.com/eoo/promotion.

EDITOR COMMENTS

Reviewing Editor:

Thank you for your responsiveness to the comments. I have no further requests.

Senior Editor:

The editors wish to thank the authors for these final adjustments to the manuscript. The article is now accepted for publication. Congratulations for an interesting and insightful study. Please consider the Journal of Physiology for your future studies.

REFEREE COMMENTS

Referee #2:

Thank you for considering the prior comments. There are no additional comments.

Referee #3:

The comment provided in my last review has been well addressed. I'm satisfied with the p-value reporting as is.